# The role of hydration in the removal of glyphosate (GLY) and aminomethylphosphonic acid (AMPA) by nanofiltration membranes

Phuong B. Trinh[1], Minh N. Nguyen [1], Zdenek Futera[2], Babak Minofar [3], Marco Personeni [4], Poul B. Petersen [4] & Andrea I. Schäfer [1]✉

Nanofiltration can remove glyphosate (GLY) and aminomethylphosphonic acid (AMPA) from water via steric, Donnan, and dielectric exclusions, although the significance of dielectric exclusion resulting from hydration has not been elucidated. This study investigates the properties of hydration and its role in GLY/AMPA removal. Results show that charge and dielectric exclusions are dominant in membranes with molecular weight cut-off (MWCO) > 150 Da. The contribution of dielectric exclusion is evident when GLY and AMPA in neutral forms are partially removed (50–80%) with >150 Da membranes at pH 2. When GLY/AMPA are negatively charged (pH from 4 to 12), GLY/AMPA removal increased from 50–80 to 90%, indicating the growing contribution of both charge and dielectric exclusions. The hydration layer can be shredded at higher applied pressures, decreasing removal from 86 to 28% (GLY) and 27 to 7% (AMPA). Both molecular dynamics and Fourier-transform infrared spectroscopy (FTIR) agree on the strong hydration of GLY/AMPA especially at pH 4–6. Understanding the role of hydration in the removal of small and charged organic micropollutants is important for tuning NF membranes for water purification.

Pollution of water resources by pesticides and herbicides is an emerging concern worldwide[1–3] and reportedly causes an annual economic loss of 78 million euros in Europe[4]. Organophosphate herbicides are widely used in agriculture[5–7], despite their well-established toxicity to aquatic organisms and human health[8]. Exposure to elevated concentrations of glyphosate (N-(phosphonomethyl) glycine) (GLY) and aminomethylphosphonic acid (AMPA) are linked to increased risks of kidney, dermatological, and respiratory issues, mental disorders, miscarriages, and cancer[9–14]. In some countries, these pesticides are not regulated and sold in commodity shops (e.g., Roundup) without any regulations[15]. GLY accounts for 30% of total herbicide use worldwide[16–18] and its market is still expanding, projected to reach 17 billion Euro by 2031[7]. In 2023, the European Commission authorized the GLY market for 10 more years until 2033[19]. GLY can be broken down by microbial organisms into shorter and potentially harmful degradation products[20,21]. AMPA is formed from GLY via an N–C bond breaking[20] and exhibits similar toxicity to humans[22,23], although AMPA is more persistent in soil than GLY[24]. Another source of AMPA is antiscalants for RO membranes, as AMPA leaching from RO can be detected in the water treatment plant effluents[25]. In ground- and surface water, GLY and AMPA generally occur at concentrations between 0.2 and 370 $\mu g\,L^{-1\,[17,26–28]}$, although in some locations, GLY has also been reported in milligrams-per-litter concentrations[29]. In Germany, GLY and AMPA concentrations were detected at around $1\,\mu g\,L^{-1\,[30,31]}$.

[1]Institute for Advanced Membrane Technology (IAMT), Karlsruhe Institute of Technology (KIT), Eggenstein-Leopoldshafen, Germany. [2]Department of Physics, Faculty of Science, University of South Bohemia in České Budějovice, České Budějovice, Czech Republic. [3]Department of Physical Chemistry, Faculty of Chemistry, University of Lodz, Łódź, Poland. [4]Physikalische Chemie II, Ruhr-Universität Bochum, Bochum, Germany. ✉e-mail: Andrea.Iris.Schaefer@kit.edu

In 2022, the European Commission proposes to regulate the herbicide concentrations in groundwater and inland surface water to be below 100 ng L$^{-1}$ for each type of herbicide, and 500 ng L$^{-1}$ for total herbicides[32]. The same thresholds are regulated for drinking water[33]. Balancing economic benefit and water environmental protection is the key to sustainable development[34], which emphasizes the importance of advanced water treatment technologies to remove GLY and its metabolites (such as AMPA) from water sources[35,36].

Nanofiltration (NF) and reverse osmosis (RO) membranes with nominal molecular weight cut-offs (MWCOs, which are the molecular weight of solutes of which at least 90% is retained by the membranes) of 100–300 Da are semi-permeable barriers for GLY and AMPA[37–39]. In a pilot system, RO membranes (DOW XLE and Toray THM) with a low nominal MWCO of 100 Da[40,41] remove 97% AMPA (111 Da) from an initial concentration of 0.7 μg L$^{-1}$ with an applied pressure of 5.8 bar[39]. With the NF 300, which is a more permeable membrane with a nominal MWCO of 180 Da, 95% GLY (169 Da) are removed from an elevated initial concentration of 48 mg L$^{-1}$ at an applied pressure of 10 bar[37]. Looser NF membranes such as the NFX (nominal MWCO of 150–300 Da) removes only 80% of GLY and 70% AMPA from an initial concentration of 50 μg L$^{-1}$ at 25 bar[38]. High pressures (up to 25 bar) for operation scaled with energy requirements and operational costs[38,42], which is a challenge for GLY/AMPA removal by RO membranes (0.4–1.7 kW h m$^{-3}$ for brackish water desalination and water reuse[43]) compared to NF membranes (0.2–0.5 kW h m$^{-3}$ [44,45]).

The incomplete removal of GLY/AMPA can be explained by the structural imperfections of NF/RO membranes, which contain sub-nanometer pores (or sometimes referred to as regions of lower polymer densities) forming transport channels of water and certain solutes[46,47]. Micropollutants such as GLY/AMPA are excluded via i) size, ii) Donnan, and iii) dielectric exclusions[48] (see Fig. 1). Size exclusion is based on the relative sizes of the membrane pores and target molecules[49–51]; removal by size exclusion thus strongly depends on NF/RO membrane pores[52], which are non-uniform in sizes[53,54], and the molecular characteristics such as shape and orientation[55]. When micropollutants penetrate the pores, adsorption on membrane surface materials (such as polyamide) occurs and subsequently, the micropollutants are transported through the membrane by a combination of desorption, diffusion, and convection in the pores[48,56] (Fig. 1D). Charged and hydrophilic micropollutants such as GLY/AMPA have a weaker interaction with polyamide than uncharged and hydrophobic micropollutants[57]; hence, adsorption of these to membrane material is low[38].

The weak attraction of charged micropollutants to the membrane polymer is also related to two other exclusion mechanisms: Donnan and dielectric exclusions[58–60]. Charge (Donnan) exclusion indicates the retention due to electrostatic repulsion between the (negatively) charged ions or micropollutants and the charged membrane surface[61]. The charge at the membrane surface (or the pore section close to the membrane surface) is characterized by the zeta potential ($\zeta$)[62] and the double layer thickness at the membrane surface (the Debye length, $\kappa^{-1}$)[63,64]. The relative scale of pore radius and the Debye length controls the transport of ions and charged species that are smaller than membrane pores[63,64]. The surface charge (or zeta potential) will determine the electrostatic interaction 'strength', which will decrease significantly (by around 2.8 times, or 1 e at the Debye length distance[63–65]. If the Debye length is greater than the pore radius, charged species at the pore entrance likely repel the charged membrane surface or pore walls and encounter resistance depending on the (pore) surface electrostatic potential[66], increasing the likelihood of removal (Supplementary Fig. 1). Naturally, pores are tortuous, which complicates the application of this conceptual model. The strength of electrostatic interactions depends on the charges of the membrane surface and GLY/AMPA[37]. Both micropollutants are polar and hydrophilic with amine, phosphate, and carboxylate groups[67,68], which makes the charge strongly pH-dependent (Supplementary Fig. 2).

Dielectric exclusion (i.e., retention of the hydrated solutes) depends on the combined size of the molecule (or ion) plus the hydration layer being larger than the pore[60,69], and the resistance of ions/micropollutants entering the pores due to an energy barrier associated with dehydration[70,71]. Dielectric exclusion always occur and is directly independent of the charges of micropollutants and membrane (although the hydration properties may be influenced by the charges)[72]. Many studies have looked into dielectric exclusion for ion transport through NF/RO pores[59,73], but research on this mechanism for charged organic compounds (that are larger than ions but may possess multiple charged sites) is sparse. Molecular dynamics (MD)[74], density functional theory (DFT) simulations[75], and quantum mechanics (ab initio) calculations[76] suggest that zwitterionic GLY adopts a linearized structure in water and dominantly forms strong hydrogen bonds with water at the phosphate and carboxylate ends. Larger hydration shell sizes can be induced with the increase of molecular

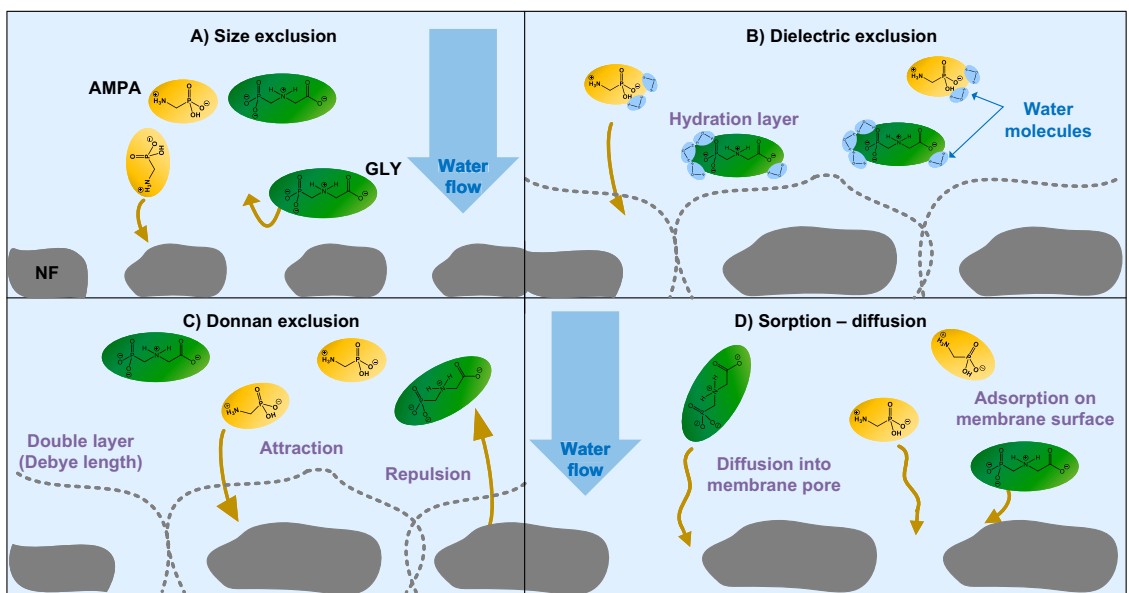

**Fig. 1 | GLY/AMPA removal mechanisms by NF membranes.** Size exclusion (**A**), dielectric exclusion (**B**), Donnan exclusion (**C**), and sorption–diffusion (**D**).

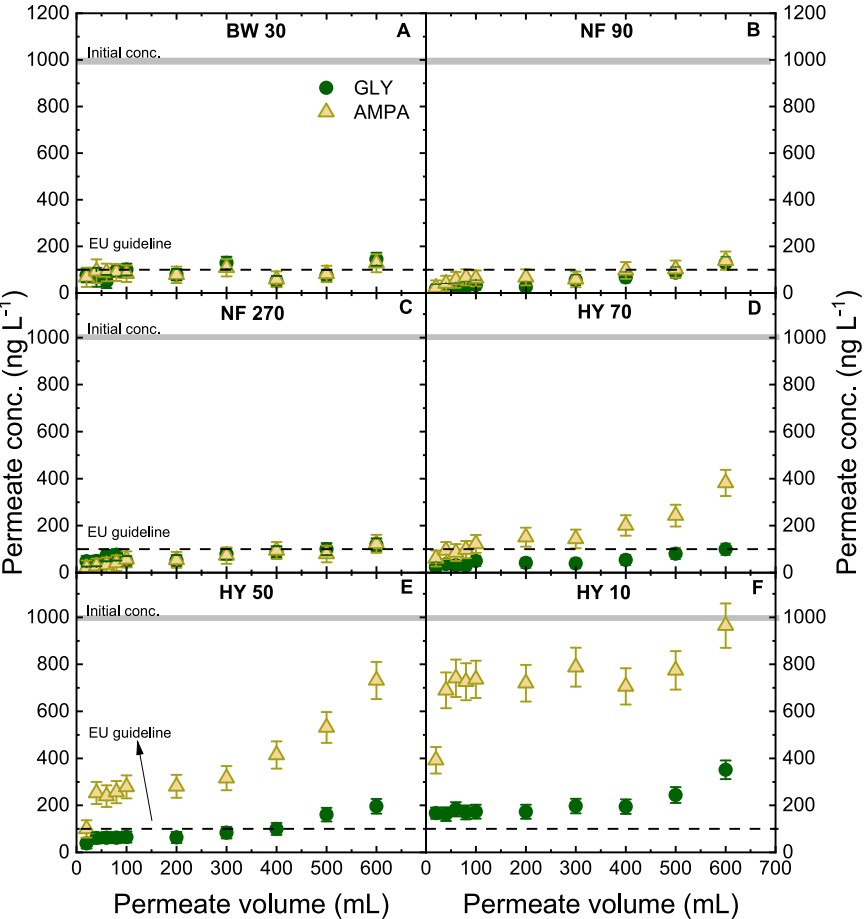

**Fig. 2 | GLY/AMPA retention property of NF membranes.** Permeate concentration as a function of permeate volume with removal by BW 30 (**A**), NF 90 (**B**), NF 270 (**C**), HY 70 (**D**), HY 50 (**E**), and HY 10 (**F**) with respective membrane nominal MWCO (flux 50 L m$^{-2}$ h$^{-1}$, initial GLY/AMPA concentration 1 µg L$^{-1}$ (each, mixed), 1 mM NaHCO$_3$, 10 mM NaCl, pH 8.2 ± 0.1, 20 °C). Flux for BW 30 and HY 70 was 30 L m$^{-2}$ h$^{-1}$ instead of L m$^{-2}$ h$^{-1}$ because the maximum pressure was reached. The nominal MWCO values for the membranes can be found in Supplementary Table 2. Error bars represent propagated error from operational parameter variations and analytical error.

charge corresponding to the change in phosphate and carboxylate group charges[74]. Ions or molecules with high charge density mean more energy is required to shed the hydration layer[77,78]. Such shredding is possible if the kinetic energy (for example, by applied pressure in pressure-driven membrane filtration) surpasses the activation energy of partial hydration, allowing charged ions and molecules to enter the NF/RO pores[79,80]. This dehydration effect has been reported for solvated ions (and potentially charged molecules) from computational[81,82] and experimental studies[79,83].

It is possible to characterize the strength of the hydration layer of charged pollutants. Infrared solvation shell spectroscopy[84] and Raman spectroscopy, paired with multivariate-curve-resolution[85,86] (IR-SSS and Raman-MCR) can extract the vibrational spectrum of a probe molecule with its solvation shell from the total spectrum. The total spectrum of a solution is considered to contain two contributions: the spectrum of (i) the bulk background solvent and that of (ii) the solute molecule with its solvation shell, i.e., the part of the solvent that is perturbed by the solute due to the interactions between the solute and solvent[84,87]. The spectra of the probe molecule solution and that of the pure background solvent are measured individually[88], and their difference (following a mathematical MCR routine[89]) results in the solute-correlated spectrum[90,91]. The obtained solute-correlated spectrum is analyzed for the spectral shifts of the OH stretch in the solvation shell relative to that of the bulk. The OH stretch vibration is very sensitive to the local hydrogen-bonded network and thus an excellent probe of the changes in the hydrogen-bonding strengths between the solute and

solvent compared to that of the solvent-solvent interactions[92]. Here, stronger hydrogen bonding weakens the OH bond, resulting in a red-shift of the OH stretch vibration. The OH stretch spectrum thus provides a map of the hydrogen-bond strengths within the sample[84].

Combining spectroscopy, molecular dynamics simulation, and filtration experiments, this study investigates the hydration properties of GLY and AMPA, which are small and charged micropollutants with multiple charged sites, and how it potentially influences GLY/AMPA removal by NF. The role of hydration in combination with dielectric exclusion in the removal will be elucidated by addressing the following research questions: 1) Which membrane shows the strongest evidence of exclusion due to the hydration layer?; 2) How does the hydration layer of GLY/AMPA vary with pH?; 3) Does the hydration strength affect the interaction of GLY/AMPA with membrane material and transport?

## Results and discussions

The filtration experiment of GLY and AMPA with NF membranes proceeded with variable nominal MWCO, pH, and flux to elucidate the importance of varied removal mechanisms (steric, Donnan, and dielectric exclusions). Experimental filtration observations will be correlated with the Fourier-transform infrared spectroscopy (FTIR) spectra as well as the radial distribution functions (RFD) of GLY/AMPA at different pH. In the first instance, the removal of GLY and AMPA for the chosen membranes is established. Filtration parameters (pressure, flux, temperature) are reported in Supplementary Figs. 9–14.

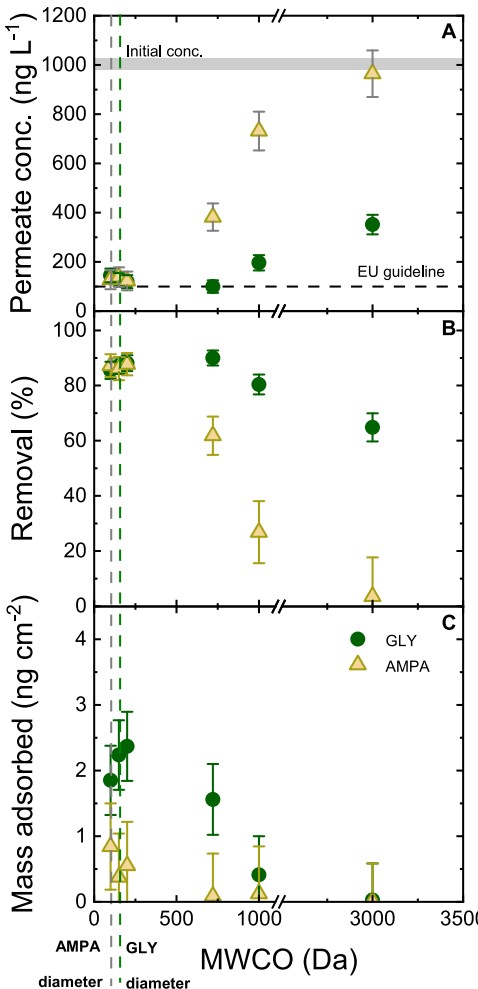

**Fig. 3 | Evaluation of size exclusion.** Permeate concentration (**A**), removal (**B**), and specific mass adsorbed (**C**) as a function of membrane nominal MWCO (flux 50 L m$^{-2}$ h$^{-1}$, initial GLY/AMPA concentration 1 μg L$^{-1}$, 1 mM NaHCO$_3$, 10 mM NaCl, pH 8.2 ± 0.1, 20 °C). Flux for BW 30 and HY 70 was 30 L m$^{-2}$ h$^{-1}$ instead of 50 L m$^{-2}$ h$^{-1}$ because the maximum pressure was reached. The nominal MWCO values for the membranes can be found in Supplementary Table 2. Error bars represent propagated error from operational parameter variations and analytical error.

## Removal of GLY and AMPA by NF membranes

The filtration of NF membranes (BW 30, NF 90, NF 270, HY 70, HY 50, HY 10) was performed with a feed concentration of 1000 ng L$^{-1}$ that was relevant in German surface waters[31]. The change in permeate concentration with filtrated (permeate) volume is shown in Fig. 2.

For denser membranes (BW 30, NF 90, NF 270), the permeate concentration evolution patterns are relatively stable and identical (steady state from 70 ± 10 ng L$^{-1}$ at permeate volume of 20 mL to 120 ± 20 ng L$^{-1}$ at permeate volume 600 mL) (Fig. 2A–C). The increase is related to the increase in concentration in the stirred cell over the course of an experiment, while removal can be explained with both adsorption and steric exclusion[93]. GLY concentrations remained low for HY 70 and HY 50 (100 ± 16 ng L$^{-1}$ L and, 196 ± 25 ng L$^{-1}$, respectively) because GLY (0.62 nm[94]) is larger than AMPA (0.49 nm[94]), hence retention may be better (Fig. 2D, E). Note that the sizes stated above are the equivalent sphere hydrodynamic diameters of GLY and AMPA that do not account for the hydration layer thickness or shape. For the most open membrane, HY 10 (nominal MWCO 3000 Da), the permeate concentration of GLY was 351 ± 33 ng L$^{-1}$ (Fig. 2F) due to the larger pore diameter (2.83 nm[94]).

The AMPA permeate concentration was higher for the looser membranes and increased more significantly with permeate volume as

the concentration in the cell increased (Fig. 2D–F). The permeate concentration of AMPA increased from 60 ± 25 to 382 ± 55 ng L$^{-1}$ for HY 70, from 100 ± 26 to 731 ± 75 ng L$^{-1}$ for HY 50, and from 392 ± 36 to 964 ± 94 ng L$^{-1}$ for HY 10. The initial low AMPA concentration at the first permeate (20 mL filtration) in HY 50 and HY 10 was likely due to the initial adsorption of AMPA on the membrane active layer with sulfonated polyether sulfone material[95]. The removal of GLY/AMPA by HY 70, HY 50, and HY 10 membranes was initially determined by adsorption (first 300–400 mL) and later on declined due to increasing diffusion, dependent on size and charge exclusion. Further investigation on the retention and adsorption of GLY and AMPA will be carried out as a function of nominal MWCO.

## Retention and adsorption of GLY and AMPA by NF membranes

To ascertain to what extent adsorption and size exclusion contribute to removal mechanisms (Fig. 3), filtration of GLY and AMPA at pH 8 was performed with six different membranes (BW 30, NF 90, NF 270, HY 70, HY 50, and HY 10). The active layers of BW 30, NF 90, and NF 270 are made of polyamide, and those of HY 70, HY 50, and HY 10 are made of sulphonated polyethersulfone. The corresponding nominal MWCOs are in the range of 100–3000 Da and are summarized in Fig. 2.

BW 30 (average pore diameter 0.41–0.51 nm[94]), NF 90 (0.44–0.63 nm[94]), and NF 270 (0.57–0.89 nm[94]) membrane with nominal MWCO < 200 Da[96,97] can remove 85–88% GLY and AMPA (Fig. 3). The removal of GLY/AMPA by BW 30 membrane can be attributed to a combination of adsorption and size exclusion. The incomplete removal (<100%) was assumed to be due to the inhomogeneity of pore sizes of the commercial membranes[53,54], and possibly concerted processes of adsorption-desorption and gradual transport in the sorbed phase through the larger pores[56]. Even though the pore diameter of BW 30 (0.41–0.51 nm) is smaller than the hydraulic diameters of GLY and AMPA (0.62 nm and 0.49 nm, respectively[94]) for effective size exclusion, it must be noted that the NF membranes contain larger pores and defects, which allows the breakthrough of GLY and AMPA[53,54]. Additionally, the transport of micropollutants through the membrane pores also depends on the molecular shape and orientation[55]. While size exclusion is the decisive factor leading to the retention of GLY/AMPA by BW 30, Donnan and dielectric exclusions cannot be neglected in the pollutant transport through the dense membranes and hence their partial removals. Meanwhile, the removal by NF 90 and NF 270 membranes was attributed by all transport mechanisms (adsorption, size, charge, and dielectric exclusion). GLY adsorption by polyamide membranes was quantifiable, from 2 to 2.5 ng cm$^{-2}$ with uncertainty of 1.0 ng cm$^{-2}$. However, AMPA adsorption to the polyamide membranes was very low (0.5–0.8 ng cm$^{-2}$) and insignificant (see error bars). GLY and AMPA can be adsorbed on polyamide membranes to a low extent via hydrogen bonding and electrostatic interaction[38,98]. However, due to the weak interaction between GLY/AMPA and the membrane polymer, GLY and AMPA can be desorbed, diffuse through, and be released from the membrane structure into the permeate. This results in the partial removal by membranes, even with low nominal MWCOs.

For membranes with nominal MWCO > 200 Da (HY 70, HY 50, and HY 10), removal decreased from 87 to 64% (GLY) and from 87 to 3% (AMPA) (Fig. 3). It was expected that HY series membranes could not reject GLY and AMPA by size exclusion due to the big pore diameter (0.67–2.83 nm[94]). However, the partial removal where the pore diameter is higher than the hydrodynamic diameter of GLY/AMPA indicates that size exclusion is not the only removal mechanism. The mass adsorption of GLY and AMPA by HY membrane was low (<0.4 ng cm$^{-2}$ for GLY, and <0.1 ng cm$^{-2}$ for AMPA with an uncertainty of 1 ng cm$^{-2}$), indicating a low interaction and adsorption capability of GLY and AMPA to the membrane polymer. Sulfonated polyether sulfone is the active layer material of HY membranes has a lower affinity for GLY/AMPA than polyamide.

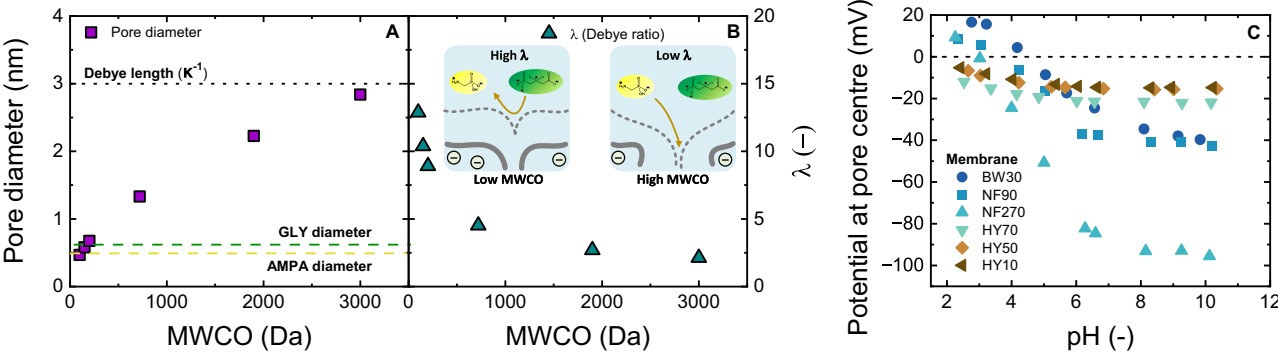

**Fig. 4 | Charge property of NF membranes.** Membrane pore diameter (**A**) and Debye ratio (λ) (**B**) for NF membranes as a function of nominal MWCO ($\kappa^{-1} = 3$ nm at $C_{NaCl} = 10$ mM and 20 °C), and schematic illustration of the interplay between the Debye length of a charged molecule and an idealized membrane pore: relatively high retention at low MCWO and lower retention at high nominal MWCO. **C** Potential at the center of the NF pores.

**Fig. 5 | GLY/AMPA retention at different pH.** Permeate concentration, removal, and mass loss as a function of pH with GLY/AMPA removal by NF 90, NF 270, and HY 50 (flux 50 L m$^{-2}$ h$^{-1}$, initial GLY/AMPA concentration 1 µg L$^{-1}$, 1 mM NaHCO$_3$, 10 mM NaCl, 20 °C). Error bars represent propagated error from operational parameter variations and analytical error.

In summary, the removal of GLY and AMPA by membranes with pore diameters larger than the equivalent sphere molecular diameters (all except BW 30) was due to other mechanisms. The role of Donnan and dielectric exclusion will be investigated next.

**Charge exclusion of GLY and AMPA with loose membrane**
Prior to the Donnan exclusion examination, the membrane double layer was investigated by the Debye length and charge at the pore center. Debye length ($\kappa^{-1}$) and Debye ratio (λ) is reported as a function of membrane nominal MWCO/average pore diameter (Fig. 4A, B). The potential at the pore center would be calculated based on the zeta potential of the NF membrane, the Debye length, and the average pore radius (Fig. 4C). If the diameter of GLY/AMPA is larger than the Debye length, Donnan exclusion can be a relevant mechanism, although the strength of electrostatic interactions between GLY/AMPA and membrane surface will depend on the electrostatic potential in the pore

space, which is the electrostatic repulsive interaction between the micropollutants and the membrane surface (notably the pore entrance), as this repulsion prevents the micropollutants from entering the pores (Donnan exclusion).

The Debye length ($\kappa^{-1} = 3$ nm) is larger than the pore dimensions of all membranes (0.46–2.83 nm[94], Fig. 4A), and the Debye ratio ($\lambda$) decreases from 12.8 to 2.1 with the increase of nominal MWCO from 100 to 3000 Da, but remains greater than 1 (Fig. 4B). The large Debye length indicates the constant electrostatic interaction between GLY/AMPA and membrane pores at different pH (corresponding to different charges, Supplementary Fig. 2). BW 30, NF 90, and NF 270 are more negatively charged than the HY 70, HY 50, and HY 10 membranes, indicating more charge interactions with the GLY/AMPA molecules (Fig. 4C).

The electrostatic potentials at the pore centers were low (in the order of −40 mV (BW30, NF 90), and −95 mV (NF 270) at neutral pH 8. The charge of the membrane originates from the membrane fixed charged groups (carboxyl and amine)[99]. The potential at the pore center of HY membranes were in the order of −20 mV at neutral pH 8. GLY and AMPA are negatively charged at pH 8 (valence of −2 and −1, respectively), which means strong electrostatic repulsion can be expected with the polyamide membranes (BW 30, NF 270 and NF 90), while a weaker electrostatic repulsion would occur with the sulfonated polyether sulfone membranes (HY series). As a result, the HY membranes are ideal for investigations of dielectric exclusion.

For Donnan exclusion mechanism investigations, the membrane that is dominated by size exclusion (BW 30) will be omitted from investigations. HY 70 and HY 10 membranes are similar to HY 50 in that their nominal MWCOs are larger than the MW of GLY and AMPA. Therefore, NF90, NF270, and HY 50 membranes were chosen for the evaluation of Donnan exclusion where filtration of GLY and AMPA at different pH was performed (Fig. 5). Filtration results of BW 30, HY 70, and HY 10 at different pH are shown in Supplementary Figs. 3–5.

The mass adsorbed of GLY and AMPA was measurable and significantly lower for HY 50, compared to NF 90 and NF 270 (Fig. 5). At pH 2 no adsorption was detected, while at pH 4–12 the polyamide membranes were in the order of 2–3.5 ng cm$^{-2}$ (Supplementary Fig. 4). This highlights the role of functional group charge in the adsorption of GLY/AMPA to membrane polymers. The adsorption of GLY and AMPA onto polyamide (NF 90 and NF 270 membrane material) is likely to be due to hydrogen bonding, van der Waals interactions, and electrostatic interaction via membrane surface functional groups such as amine[100]. Meanwhile, HY 50 exhibited low adsorption of GLY and AMPA (<0.5 ng cm$^{-2}$) due to lower adsorption of hydrophilic compounds like GLY and AMPA onto hydrophobic materials like sulphonated polyethersulfone as well as the low electrostatic attraction[101].

At pH 2, the charges of GLY and AMPA are 0, while the charges of NF 90, and NF 270 are positive (8–9 mV) and the charge of HY 50 is negative (−15 mV) (Fig. 4). The removal of GLY was 50–90 ± 5% while the removal of AMPA was 9–62 ± 12% (Fig. 5). The removal of GLY and AMPA was also low (5–50 %) for HY 70 and HY 10 membranes (Supplementary Fig. 5). At this pH, GLY and AMPA have a neutral net charge (Supplementary Fig. 2), and the removal of GLY was higher than that of AMPA in all membranes. Because of the neutral charges, removal at pH 2 was less likely to be contributed by Donnan exclusion. The contribution of dielectric exclusion is strong in the case of HY 50 at pH 2, as the membrane pore size is larger than the hydrodynamic diameters of GLY and AMPA, and the adsorption of the micropollutants was insignificant.

At pH > 2, the removal of GLY and AMPA could be attributed to all removal mechanisms (size exclusion with the denser NF 90, Donnan exclusion, and dielectric exclusion). GLY and AMPA removal by NF 90 remained higher at 90 ± 5% at all pH values. By increasing the pH from 4 to 12, GLY removal increased from 84 ± 3 to 94 ± 5% for NF 270, and from 66 ± 7 to 83 ± 6% for HY 50. AMPA removal

increased from 71 ± 6 to 92 ± 5% for NF 270, and from 12 ± 5 to 83 ± 11% for HY 50. The same increase in removal of GLY/AMPA was observed with HY 70 and HY 10 (Supplementary Fig. 5), while BW 30 did not exhibit any change in removal with pH because size exclusion dominates (Supplementary Fig. 3). The increase in removal by pH observed with HY 50 membrane can be due to two phenomena: i) stronger electrostatic repulsion between GLY/AMPA and the membrane surface at increasing pH, and ii) denser hydration layer with higher charge density of GLY/AMPA, resulting in larger effective hydrated diameters. Size exclusion, if any, is not a decisive factor as the nominal MWCO of HY 50 membrane (1000 Da) is much higher than the molecular weights of GLY (169 Da) and AMPA (111 Da). The removal of GLY was always higher than that of AMPA for all pH values, which could be attributed to the higher negative charge of GLY than that of AMPA at the same pH (Supplementary Fig. 2). The contribution of dielectric exclusion is not clear. In an attempt to indicate a clearer contribution of hydration to the overall GLY/AMPA removal by NF membranes, the driving force was varied in experiments with different fluxes.

## Removal of GLY and AMPA at different fluxes

Molecules in solution are surrounded by a hydration shell, and partial dehydration is expected when molecules enter the membrane pores. This has been reported for inorganic ions[59]. In pressure-driven membranes like NF/RO membranes, the hydration layer is expected to be shredded more at higher pressures (and thus higher fluxes), and hence lower removal is expected due to less effective dielectric exclusion. To evaluate the contribution of flux to the removal of charged micropollutants, the transmembrane flux was varied (25, 35, 50, 60, 75, and 100 L m$^{-2}$ h$^{-1}$ for NF 90, NF 270, and 25, 35, 50, 60, and 75 L m$^{-2}$ h$^{-1}$ for HY 50 due to the maximum system pressure being reached by membranes with different permeabilities, Fig. 6).

Results indicate that the removal of GLY and AMPA by NF 90 and NF 270 was not affected by increasing flux from 25 to 100 L m$^{-2}$ h$^{-1}$ (2.5 to 10 bar (NF 90), and 1.5 to 6 bar (NF 270)), which remained at 80–90% for both GLY and AMPA due to the strong Donnan and dielectric exclusion (Fig. 6). Consequently, no clear change in dielectric exclusion by flux could be observed. The mass adsorbed was significant at the lowest flux (25 L m$^{-2}$ h$^{-1}$, 2.5 bar) where mass adsorbed on NF 90 was 4 ± 1 and 3.6 ± 1 ng cm$^{-2}$ for GLY and AMPA, respectively (Fig. 6C). NF 270 could also adsorb 3 ± 1 and 1.4 ± 1 ng cm$^{-2}$ for GLY and AMPA, respectively, at the flux 25 L m$^{-2}$ h$^{-1}$ (1.5 bar) (Fig. 6F). The mass adsorbed decreased by 70% with the increase of flux and reached 1.2 ± 0.7 ng cm$^{-2}$ and around zero for GLY and AMPA, respectively, at the flux of 100 L m$^{-2}$ h$^{-1}$ (6 bar) due to shorter hydraulic residence times that will limit adsorption (Fig. 6C, F).

For HY 50, the removal decreased with increasing flux, from 86 ± 3 % at flux 25 L m$^{-2}$ h$^{-1}$ (3 bar) to 28 ± 8% at flux 75 L m$^{-2}$ h$^{-1}$ (9 bar) for GLY, and from 27 ± 11 to 7 ± 5 % for AMPA (Fig. 6H). This is an indicator of dielectric exclusion associated with hydration. If electrostatic exclusion was a controlling retention mechanism, GLY and AMPA would not vary with increasing pressure. Higher pressure (higher flux) applied to the membrane provides molecules with kinetic energy to overcome the energy barrier and shred hydrogen-bonded water molecules[81]. The break in GLY removal by the HY 50 membrane at a flux of 50 L m$^{-2}$ h$^{-1}$ (6 bar) indicates the pressure threshold where the hydration layer shreds to a large extent; the removal of AMPA gradually decreased with increasing pressure and flux and without a significant break in performance. Therefore, more molecules could pass through the membrane pores, and lower removal was observed.

The rejection of GLY and AMPA is also controlled by concentration polarization[102]. Therefore, the concentration polarization is evaluated via solute flux, concentration at the membrane surface, and mass adsorbed of the NF membranes (Fig. 7). The calculation is mentioned in supporting information.

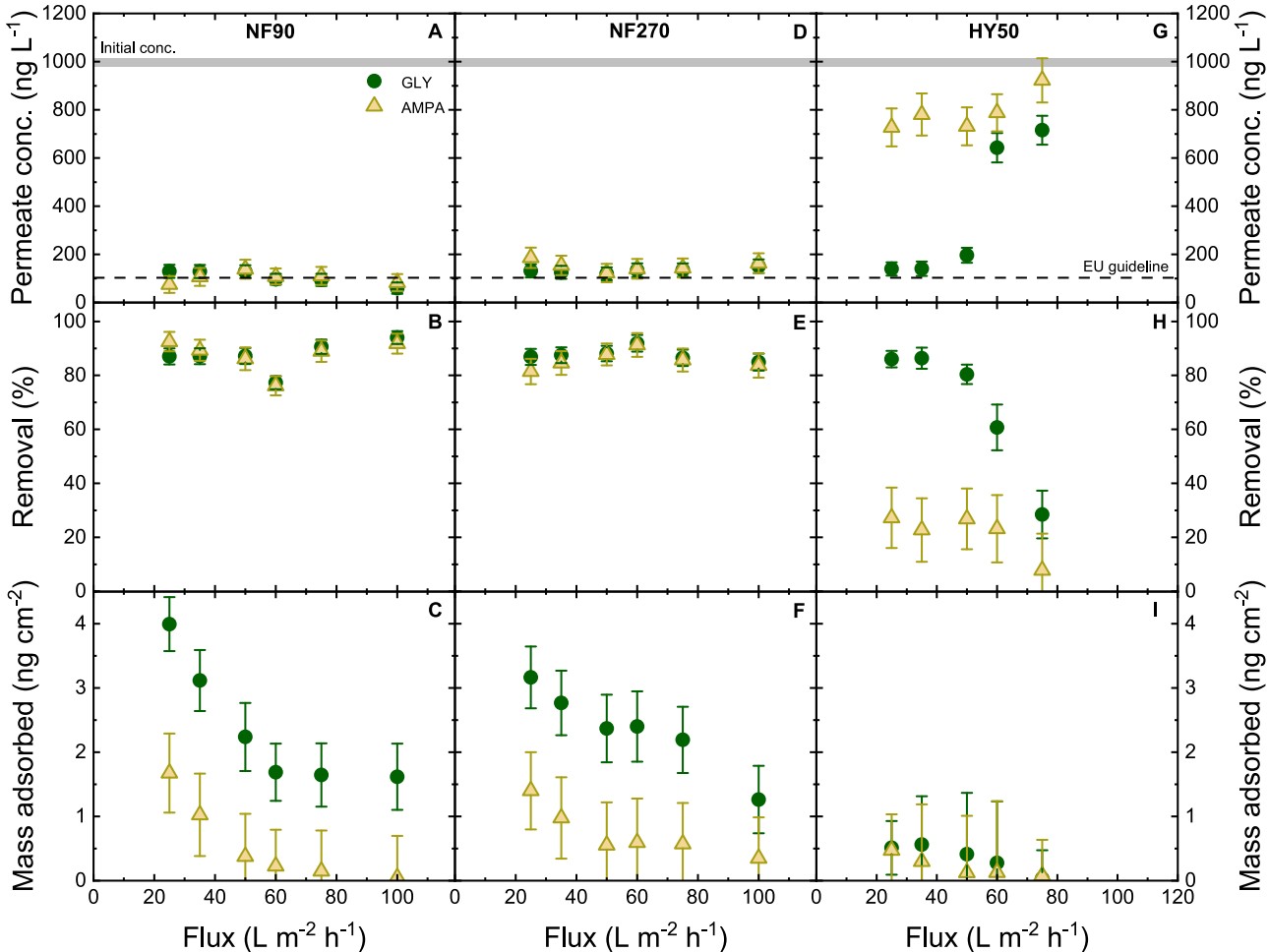

**Fig. 6 | GLY/AMPA retention at different fluxes.** Permeate concentration, removal, and mass loss as a function of flux with GLY/AMPA removal by NF 90, NF 270, HY 50 membrane (pH 8, initial GLY/AMPA concentration $1\,\mu g\,L^{-1}$, 1 mM NaHCO$_3$, 10 mM NaCl, 20 °C). Error bars represent propagated error from operational parameter variations and analytical error.

As the pure water flux increases from 25 to $100\,L\,m^{-2}\,h^{-1}$, the theoretical concentration of GLY/AMPA at the membrane surface increases by 4–20 times for NF 270, and 2–20 times for NF 90, from the feed concentration ($100\,ng\,L^{-1}$). This means there was strong concentration polarization linked to the high retention by NF 270 and NF 90 (80–90%, Fig. 6). However, for HY 50, the theoretical concentration at the membrane surface varied between 1470 and $4080\,ng\,L^{-1}$, and does not deviate strongly from the feed concentration. This corresponds to lower retention by this membrane (from $86 \pm 3$ % to $28 \pm 8$% for GLY, and from $27 \pm 11$ to $7 \pm 5$% for AMPA, with clear decreasing removal trend at increasing flux). The permeate GLY/AMPA flux shows a slight increase trend with pure water flux for NF 270 and NF 90, but there was stronger increase for HY 50 membrane. While retention by HY 50 membrane can be attributed to an interplay of Donnan and dielectric exclusions (and a degree of steric exclusion by the small pores of the pore size distribution), the trend in solute flux is likely attributed to dielectric exclusion and the increased hydration shell shredding at higher pressures. The higher pressure contributes to the molecular kinetics and speeds up the hydrogen bond network reordering (Supplementary Figs. 6, 7). For NF 270 and NF 90, the contribution of dielectric exclusion is unclear. The real removals (calculated from the difference between the permeate concentration and the concentration at the membrane surface) of NF 90 and NF 270 were 92–99% for both GLY and AMPA, while the real removal of HY 50 was 82–90% for GLY and 52–80% for AMPA.

## Characterization of GLY and AMPA hydration

FTIR in the attenuated total reflection (ATR) was used to characterize the structuring of water molecules in the hydration layer and to quantify the strength of hydrogen bonding between GLY/AMPA molecules and water molecules. The simplified MCR routine applied to FTIR-ATR spectra was used to characterize the hydrogen bonding between GLY/AMPA molecules and the bulk of water molecules.

The measurement was taken in Milli-Q water at a concentration of $100\,mg\,L^{-1}$ (100,000 times higher than the GLY/AMPA concentration in the filtration experiments of $1\,\mu g\,L^{-1}$) as a function of pH (4, 6, 8, 10, 12) (Fig. 8A, B). In addition, the distribution of hydrogen bonding distance between water molecules was simulated by MD at different GLY and AMPA charges to be compared with the results from FTIR-ATR measurements (Fig. 8C, D). The distributions were collected from 5000 samples equally spaced along the 10 ns MD trajectories. The hydration shell distribution of water around GLY and AMPA molecules is visualized at different charges, as shown in Fig. 8E.

In a closer look, the broad O–H vibrational band covering ~3000–3700 cm$^{-1}$ experienced a notable peak splitting and shift to both higher and lower frequencies was observed for both GLY and AMPA when decreasing the pH from 8–12 to 4–6. These shifts reflect changes in the hydrogen-bonded network around the solutes, where at the higher pH values, some water molecules are involved in a more structured hydrogen-bonded complex, and the remaining water molecules become less structured. This difference is driven by the charge state of GLY and AMPA going from −1 to −3 for GLY and 0 to −2

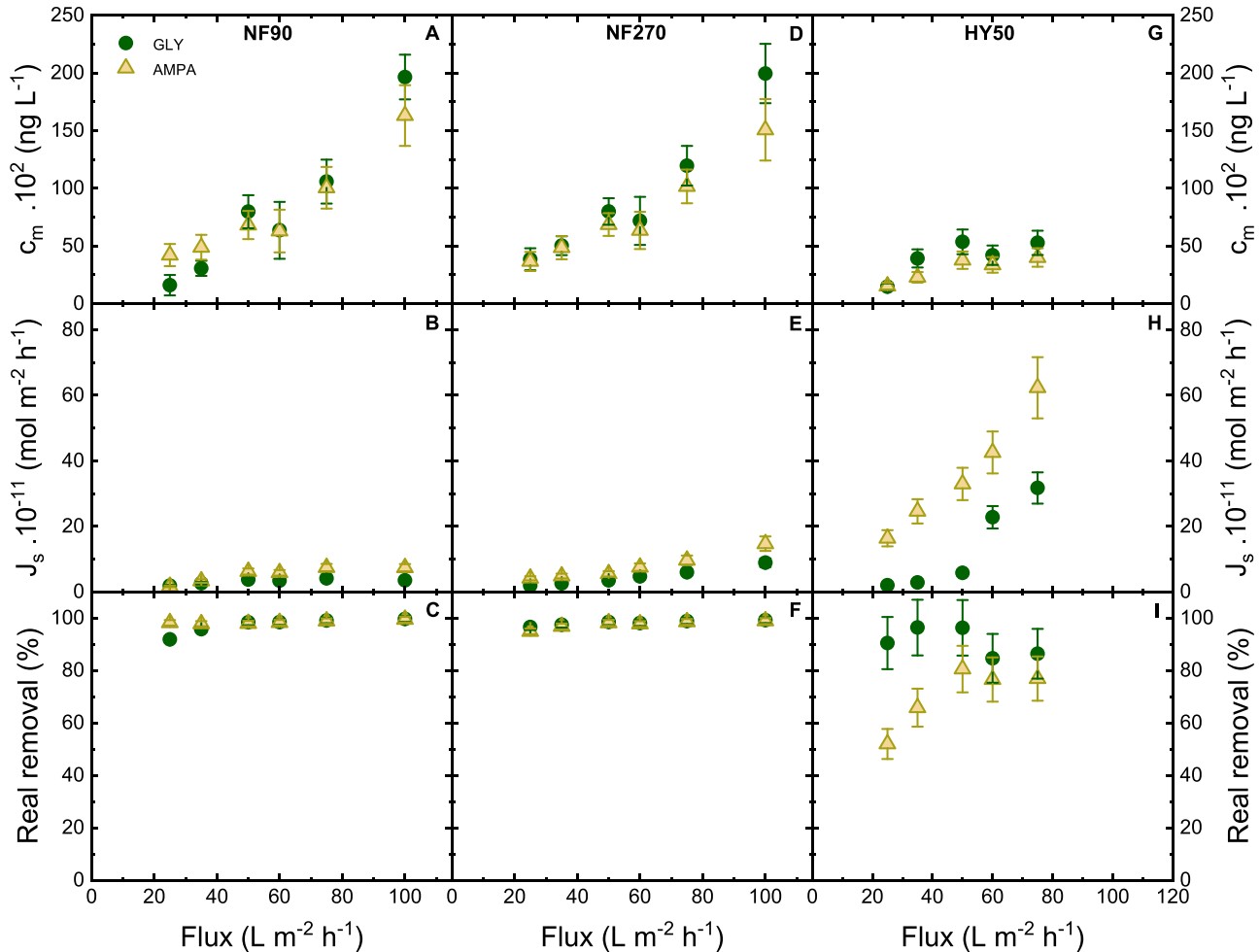

**Fig. 7 | Concentration polarization in NF membranes.** Concentration at membrane surface, solute flux, and real removal as a function of flux with GLY/AMPA removal by NF 90, NF 270, HY 50 membrane (initial GLY/AMPA concentration 1 μg L$^{-1}$, 1 mM NaHCO$_3$, 10 mM NaCl, pH 8, 20 °C). Error bars represent propagated error from operational parameter variations and analytical error.

for AMPA, further affirmed and further clarified by the MD simulations (Fig. 8C, D). The distance between oxygen in water molecules was 0.28 nm at charge +1, 0, and −1 for GLY (corresponding to pH 2–4), and the same (0.28 nm) at charge +1 and 0 for AMPA (corresponding to pH 2–4). For GLY at charge −2 and −3 (corresponding to pH 6–12) and AMPA at charge −1, and −2 (corresponding to pH 6–12), the distance between oxygen in water molecules was smaller (0.26 nm). The longer hydrogen-bond distances between water (at low pH) indicate a larger separation between water in the hydration shell and bulk water. Therefore, the stronger/more structured hydration shell of GLY and AMPA (stronger interaction between GLY/AMPA with water molecules) is observed at lower pH, which is comparable with the FTIR spectrum. The overall weakening of hydrogen bonding partly results in lower removal of GLY at lower pH, which was observed in NF 270 and NF 90, and most evidently in HY 50 membranes. At higher pH, the hydration shell is potentially easier to shred when the micropollutants enter the NF pores.

The hydration shell distribution of water around GLY and AMPA molecules as a function of charge was simulated by MD (Fig. 8 E). More water was found to be in interaction with GLY/AMPA, forming a bigger hydration layer at higher pH than at lower pH, which can be linked to better removal of GLY/AMPA at higher pH in NF membrane filtration (Fig. 5). However, the diameter of the micropollutant plus hydration shell appear to vary little at different pH and in the order of 1.6 nm, which means dielectric exclusion does not contribute to any retention

if the membrane pores are larger than the hydration shell sizes (i.e., for the case of HY 10, see Supplementary Fig. 5). In contrast, there is clear variation in the number of water molecules involved in hydration, which is an indicator of hydration shell strength. In the case of GLY, an increase in water density around the PO$_3$ group was observed with the deprotonated in charge state from −1 to −3 (corresponding to the increase of pH from 4 to 12). A similar effect was observable in AMPA, where the solvation shell around PO$_3$ is also denser at high pH, with the charge changing from 0 to −1 with pH from 4 to 12. The hydration layer around the PO$_3$ group was more structured than the hydration of the carboxylate group (COO$^-$) in GLY. The hydration shell around the COO$^-$ group remained the same with pH as the COO$^-$ group was deprotonated at all pH levels (Supplementary Fig. 2). At pH <10, the central NH$_2$ group of GLY interacts with water molecules, which can be seen on the blue rings around the central part of the molecule, indicating the oxygen positions that are hydrogen-bonded to NH$_2$. This may correspond to the spectroscopic shift towards low frequencies in Fig. 8A, B where water molecules around these rings are more structured, while water molecules outside these rings are less structured. This water ring was not observed at pH >10 due to the protonation of the amine group. The difference in the density in the hydration shell between GLY and AMPA explains the higher removal of GLY compared to AMPA in NF at the higher pH, which is complementary to the results of hydrogen bond distances (Fig. 8C, D). This finding also agrees with experimental results that the removal of GLY and AMPA decreases with increasing

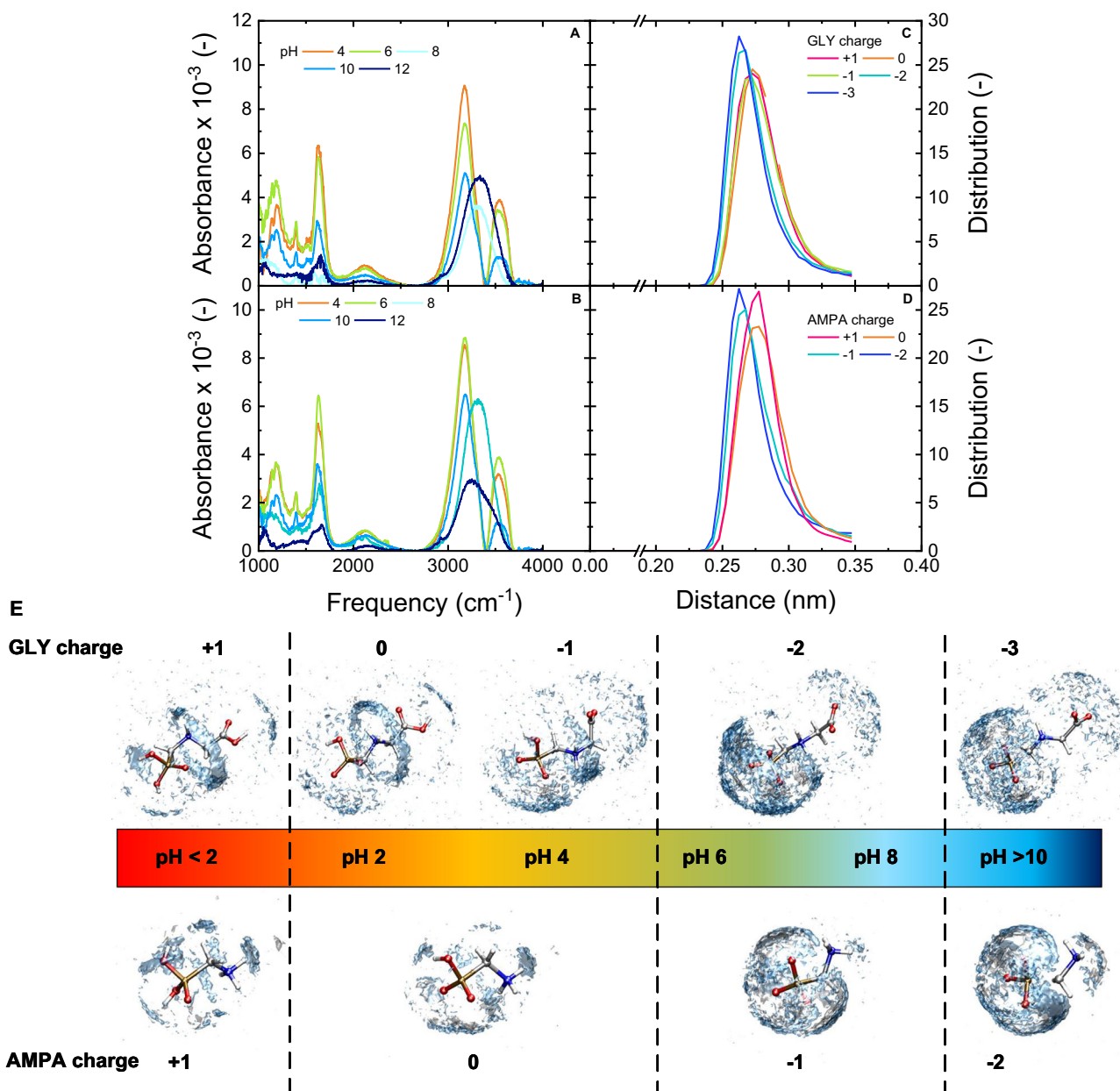

**Fig. 8 | Hydration of GLY/AMPA.** IR-solvation shell spectra of GLY (**A**) and AMPA (**B**) at pH 4–12 (GLY/AMPA concentration 100 mg L$^{-1}$). Distributions of hydrogen-bond distances between water (**C** GLY in charge states +1, 0, −1, −2, −3; **D** AMPA in charge states +1, 0, −1, −2); **E** Hydration shell distribution functions of water around GLY and AMPA. The blue areas indicate the positions of the oxygens, while the hydrogen position distributions are shown in silver. The distributions were collected from 5000 samples equally spaced along the 10 ns MD trajectories.

pH (Fig. 5), which could be partly attributed to the strengthening of the hydration shell at NF (especially HY 50) pores.

To give insight into the interactions of GLY/AMPA molecules with water, the radial distribution functions (RDF) were calculated from the classical MD trajectories. The RDF of GLY and AMPA were computed for water oxygen and hydrogen interactions with the phosphate group (Fig. 9), the carboxyl group of GLY, and the amino group of AMPA (Supplementary Fig. 8).

The phosphate group exhibits similar interactions with GLY and AMPA, where RDF results changed due to the protonation of the phosphate moiety of GLY and AMPA. The water oxygens are hydrogen-bonded to the two hydrogens in the PO$_3$H$_2$ groups in the protonated +1 form, indicating the oxygen of water pointed to the phosphate group of GLY and AMPA, resulting in a strong P-OH...OH$_2$ hydrogen bonding

(Fig. 9). For both GLY and AMPA, at the higher charge states, there was a peak below 0.2 nm in the RDF of phosphate groups with water oxy-gen (Fig. 9A, C) and a peak at longer distances in the RDF of phosphate groups with water hydrogen (Fig. 9B, D). These result indeed from the P-OH...OH2 hydrogen-bond for both GLY and AMPA, which in turn result in the low-frequency peak in the solvation shell spectra at low pH (more structured hydration shell) (Fig. 8A, B). At the lower charge state found at higher pH, the phosphate is deprotonated and there is no such strong hydrogen bond, resulting in the high-frequency peak in the solvation shell spectra at low pH (Fig. 8A, B). The opposite water arrangement (hydrogen pointing to the phosphate oxygens) is typical for the deprotonated −2 and −3 forms, resulting in a P...HOH hydrogen bonding. The partly deprotonated phosphate in the 0 and −1 forms exhibits both kinds of interactions with water from the first hydration

layer. Similar trends can be seen on the RDF of the carboxyl group, which interacts with the water oxygen via hydrogen molecule at the +1 and 0 charges (corresponding to low pH <4 when the carboxyl group is protonated). At higher pH, the carboxylate group is deprotonated showing the charge −1 to −3, forming hydrogen bonding between the carboxylate oxygens and water hydrogens. In the protonated +1 charge of AMPA (pH <2), the hydration layer around the $PO_3H_2$ interacts mostly via hydrogen with the water oxygens (Fig. 9), and the same arrangement is around the $NH_3^+$ group (Supplementary Fig. 8). As the AMPA deprotonates and gets charges 0, −1, and −2, respectively, the water molecules around phosphates are rotated where hydrogen molecules point to $PO_3$ oxygens (each of the oxygen in the $PO_3$ group interacting with one water molecule). Meanwhile, the water interaction with $NH_3$ is weaker, unless the group gets deprotonated in charge −2 (pH >10), where the amine group interacts with nearby water hydrogen. Overall, the graph shows stronger interaction between the phosphate group and water molecules at higher pH, linking to the increased hydration shell strength; this, together with any Donnan exclusion at the NF pore surface, partly results in better removal of both GLY and AMPA at higher pH.

In summary, the removal mechanism of GLY and AMPA by NF membrane includes size exclusion, Donnan exclusion, and dielectric exclusion. Size exclusion was dominant in the filtration of the membrane at low nominal MWCO (<150 Da) with 90% GLY/AMPA removed. Donnan exclusion and dielectric exclusion are more dominant for membranes with a nominal MWCO > 200 Da. By comparing the Debye length and pore radii of the membranes, it was revealed that Donnan exclusion would always occur with membrane pore sizes (200–3000 Da), where GLY/AMPA interact with the membrane wall which carries the negative charge. At low pH (i.e., pH 2), the contribution of Donnan exclusion was less likely because GLY and AMPA are neutrally charged, and partial removal by HY 50 membranes implies an increased effective size of GLY/AMPA due to a hydration layer. At higher pH, Donnan exclusion by the membrane became more important, although the extent of Donnan exclusion may vary between the membranes as the HY series induces lower electrical potential (and charge) at all pH. Both Donnan and dielectric exclusions enhanced the removal of GLY/AMPA with increasing pH. Removal of GLY and AMPA reached >90% for NF 90 and NF 270, and 83% for HY 50 membrane. The higher flux would provide more kinetic energy to overcome the energy barrier for dehydration before the molecules enter the pores, although the higher concentrations in the permeate can also be attributed to a stronger concentration polarization effect at higher fluxes (i.e., higher concentration gradient across the membrane, which drives diffusion).

A denser hydration layer of GLY and AMPA was observed at higher pH by MD simulation, which explains the higher removal at higher pH. FTIR-ATR measurement showed that the hydrogen bonding of GLY/AMPA hydration layer was stronger at lower pH (4–6) than at higher pH (8–12), indicating water was held more strongly/more structured in the hydration shell at low pH, and this means the effective hydrated sizes of GLY and AMPA are smaller at low pH compared to high pH. This phenomenon was confirmed by MD simulation, which determined that the distribution of distances between water molecules was larger at low pH, indicating a larger separation between water in the hydration shell and bulk water. RDF study illustrated that water molecules interact strongly with phosphate groups in GLY and AMPA, which significantly contributed to the hydration layer of the GLY/AMPA molecules. The orientation of water molecules in the hydration layer was changed due to pH, where oxygen water points to the phosphate group at low pH and reverses at higher pH (hydrogen points to the phosphate group). The lower removal of AMPA compared to GLY may be attributed to the smaller effective hydrated sizes of AMPA.

## Methods
### Filtration system and protocol
Filtration was performed in a dead-end stainless steel stirred cell system at a fixed stirring speed of 400 rpm[97]. The mass transfer of this system was characterized by Imbrogno et al. and showed similarities to other lab-scale cross-flow systems and the spiral-wound module[97]. The dead-end system was selected for this study to elevate the shear

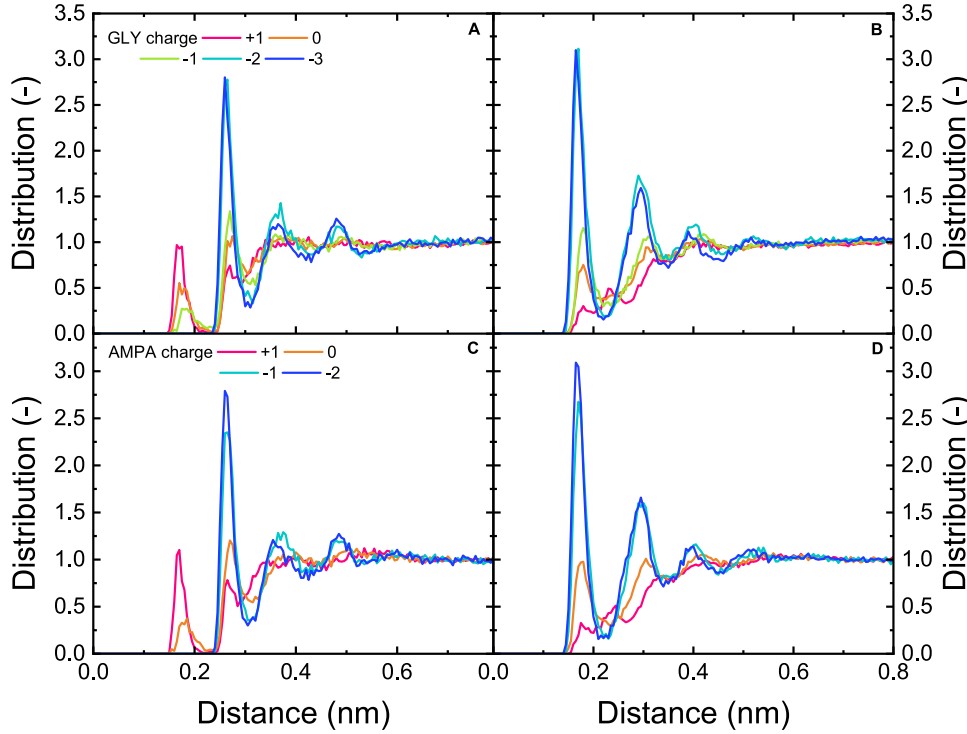

**Fig. 9 | Interaction distances between GLY/AMPA and water molecules.** Radial distribution functions (RDF) for interactions between GLY phosphate groups with water oxygen (**A**) and hydrogen (**B**) atoms; and AMPA phosphate groups with water oxygen (**C**) and hydrogen (**D**) atoms.

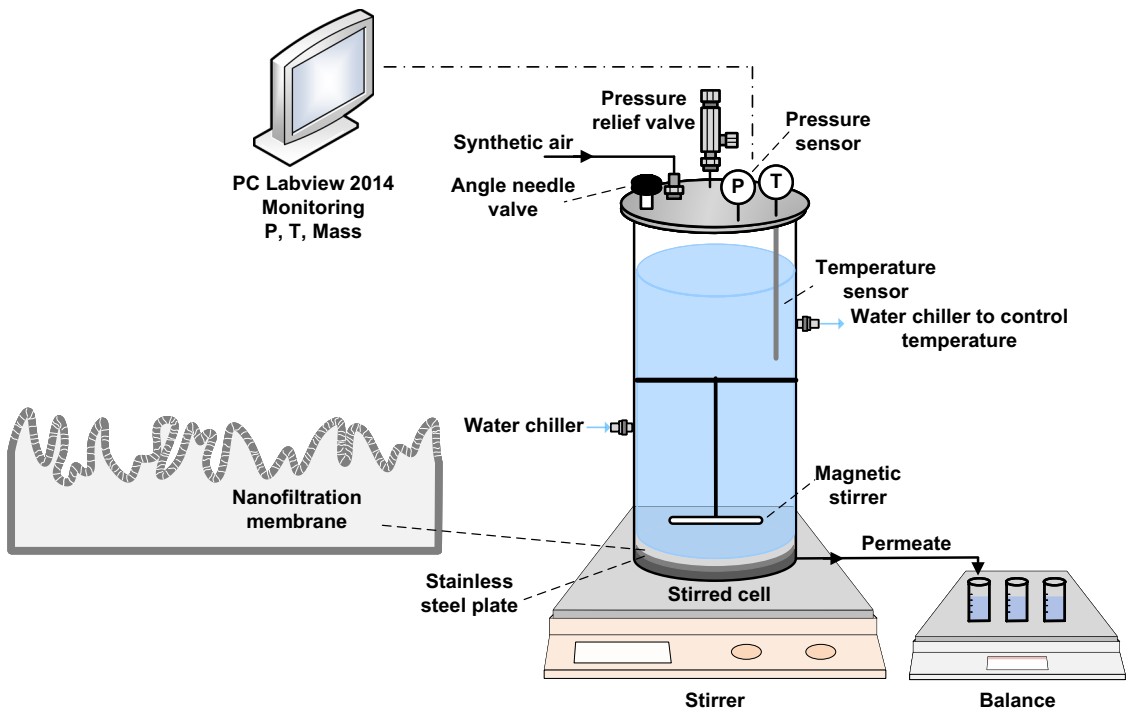

**Fig. 10 | Schematic of the stirred cell filtration system.** P and T indicate the pressure and temperature data recording, respectively.

force[103,104] and intensify concentration polarization. In this set-up, the concentration polarization reassembles the conditions in spiral-wound modules, whereas it is inadequately captured in lab-scale crossflow systems due to the inherently small membrane areas/lengths[97]. Both the shear force and concentration polarization act together in governing retention. The stirred cell which was designed and built in-house contains three parts: base, membrane cell, and top with the stirrer. The internal membrane diameter is 7 cm with an effective exposed area of 38.5 cm² and an internal volume of 990 mL (Fig. 10). The bottom part includes a stainless-steel porous support (SIKA-5AX, GKN Powder Metallurgy, Germany, thickness 2 mm, average pore diameter 10 μm, porosity 31%) to support NF membranes.

Three parts of the cell were assembled, sealed, and connected to a synthetic air tank (Alpha Gas, Air Liquid, Germany), which consisted of 20% oxygen and 80% nitrogen, which was used to pressurise the cell. Pressure and temperature were measured by a pressure transducer (PX219−30V85G5V, Omega Engineering, Germany) and a thermocouple (TJ2-CPSS-M60U-250-SB, Omega Engineering, Germany) located in the top part of the cell. The temperature of the stirred cell was controlled and regulated by a thermostatic circulator system composed of a chiller (LKB 2219 MultiTemp II, Bromma Germany) and a stainless-steel flexible serpentine (Water Way Engineering GmbH, Germany) wrapped around the stainless steel stirred cell (See Supplementary Fig. 15, UF-PBSAC filtration system). Permeate mass was measured by electronic balance (Adventurer ProAV 2102, Ohaus, Germany). All data were collected with a tailored LabView 2014 program (National Instruments, Germany). The filtration protocol is adapted from Imbrogno and Schäfer[97], shown in Supplementary Table 1.

## Membrane characteristics

Six types of NF membranes were used in this study including BW 30 (MWCO 80−120), NF 90 (MWCO 90−180), NF 270 (MWCO 150−340), HY 70 (MWCO 600−720), HY 50 (MWCO 1000−1500), and HY 10 (MWCO 3000−3600)[96,97,105,106]. Membrane properties are shown in Supplementary Table 2.

## Solution chemistry

The stock GLY and AMPA solution (Stock 1) was prepared as a mixture of both pollutants by dissolving both GLY (98%, Sigma Aldrich, USA) and AMPA powder (99%, Sigma Aldrich, USA) in Milli-Q water (Reference A+, Merck Millipore, USA) with a concentration of 10 mg L⁻¹ for each compound. Stock 2 containing GLY/AMPA at a concentration of 100 μg L⁻¹ for each compound was prepared by diluting Stock 1 in Milli-Q water. Feed solutions of environmentally relevant GLY/AMPA concentrations (1000 ng L⁻¹ each) were prepared by diluting Stock 2 in background electrolyte solutions which was a mixture of 1 mM NaHCO₃ and 10 mM NaCl. The background electrolyte stock solution was 5 mM NaHCO₃ (dissolved from analytical-grade, >99.7% powder, Merck, Germany) and 50 mM NaCl (dissolved from analytical-grade ≥99.5% powder, Honeywell Fluka, Germany). The background electrolyte had the conductivity and pH at ~1400 μS cm⁻¹ and 8.2 ± 0.1 unless adjusted with 1 M HCl (diluted from 37% HCl, Roth, Germany) and 1 M NaOH (dissolved from pellets, Merck, 99%) for pH experiments. Analytical standards (analytical-grade, 100 mg L⁻¹ in Milli-Q water) of GLY and AMPA were obtained from Dr. Ehrenstorfer (analytical grade, 99%, Germany) for analysis and quality control of the analysis. ¹³C GLY (analytical grade, 98%, Sigma Aldrich, USA) was used as the internal standard for analysis and ammonium formate powder (analytical-grade, 99%, VWR, Germany) was used as a buffer for analysis.

## Analytical method

GLY and AMPA were analyzed by liquid chromatography with tandem mass spectrometry (LC-MS/MS), which consists of an ultra-high performance liquid chromatographic system (LX50) coupled to a QSight 420 triple-quadrupole mass spectrometer (PerkinElmer, USA)[107]. The hydrophilic interaction liquid chromatography column (HILIC) Obelisc N 2.1·150 mm, 5 μm (SIELC Technologies, USA) was used with isocratic mobile phase 85/15 v/v (H₂O: acetonitrile (99.9%, VWR, Germany) + 0.05% CH₂O₂ (98%, VWR, Germany)) at the flow rate of 0.6 mL min⁻¹ and was conditioned daily before the measurement. Electrospray ionization was applied in negative mode at the source temperature of 450 °C and the surface-induces desolvation temperature at 320 °C. The elution time was 6 min and the injection volume was 100 μL.

For sample preparation, 10 μL of internal standard ($^{13}C$ GLY 100 μg L$^{-1}$) and 10 μL of ammonium formate (NH$_4$COOH 200 mM) buffer was added to each sample (at a volume of 1000 μL) as such the concentration of internal standard and buffer in each sample remained at 1000 ng L$^{-1}$ and 2 mM respectively to stabilize sample pH at 3.5. Samples with pH <4 and >10 were neutralized to pH 8 with 1 M HCl and 1 M NaOH. The GLY and AMPA concentrations were calculated based on the calibration solutions of GLY and AMPA at concentrations of 0, 2, 5, 10, 20, 50, 100, 500, and 1000 ng L$^{-1}$ for each compound. The limit of detection is 10 ng L$^{-1}$ (Perkin Elmer, USA). Data analysis was carried out using the Simplicity 3Q$^{TM}$ software platform. Details about analytical methods and calibration curves are shown in Supplementary Table 3 and Supplementary Fig. 16.

### Hydration layer characterization

IR spectra of the solutions were measured by FTIR in the attenuated total reflection (ATR) configuration. IR spectra were processed as described above (maybe put a reference/link to the section here) to obtain the solvation shell spectra. GLY and AMPA were prepared separately by dissolving GLY or AMPA powder in Milli-Q with a starting concentration of 10 g L$^{-1}$. The concentrations are still below solubility limits (12 g L$^{-1}$ for GLY, and 50 g L$^{-1}$ for AMPA[108]). Note that the concentrations had to be 10 million times higher than in the feed due to the lower sensitivity limit of the instrument. The FTIR spectrum of GLY and AMPA at different pH was measured at a concentration of 100 mg L$^{-1}$. The pH of the solutions was adjusted from 4 to 12 using 10 mM HCl and 10 mM NaOH. In addition to the spectra of the GLY and AMPA solutions, spectra of the solutions with identical composition without GLY and AMPA (solvents-only) were measured for the solvation-shell subtraction. The IR spectra were measured in the spectral range from 1000 to 4000 cm$^{-1}$ at a resolution of 4 cm$^{-1}$. The shifts in the OH stretch vibrations of water in the range of 1800–3700 cm$^{-1}$ in the solvation shell of GLY and AMPA as a function of pH were critically examined to deduce information on the hydrogen-bond strength and hydration. The shift of OH stretch vibration from the higher frequency to the lower frequency would indicate stronger hydrogen bonding, hence, a more structured hydration shell.

### Molecular dynamics simulation for the hydration of GLY/AMPA

The molecular mechanics calculations were performed by the Gromacs 2024.1 program[109]. The parameters of the molecules in different protonation states were derived from the General Amber Force Field (GAFF2)[110,111] using the structure geometries previously optimized at the QM level in vacuo. Atomic point charges were determined by the RESP method[112] from HF/6-31 G(d) wavefunctions, consistently with GAFF2. The molecules were placed in cubic simulation boxes of ~40 Å side lengths and surrounded by explicit water described by the TIP3P model[113]. The appropriate number of potassium or chloride ions was added to the box to neutralize the whole system. The resulting models were first preoptimized by 5000 steepest descent steps to avoid possible inter-molecular close constants. The system temperature was equilibrated by 50 ps MD run with 1 fs timestep under the influence of the stochastic velocity-rescaling thermostat[114]. Then, the solution density was equilibrated by 500 ps MD run with 2 fs timestep, applying the stochastic barostat[115] with the relaxation time 1 ps and compressibility 4.5 e$^{-5}$ bar$^{-1}$. Finally, the free-energy perturbation (FEP) method was applied to evaluate the hydration energy by sequential turning on of the molecule interaction with the solvent. This was done in 21 stages where both electrostatic and van der Waals interactions were scaled linearly. Beutler-type soft-core function[116] was applied to avoid close contacts. The final hydration-free energies were evaluated from the obtained distributions using Bennet's acceptance ratio method[117]. In all simulations, the electrostatic interactions were evaluated by the smooth particle mesh Ewald (PME) summation[118] with the 12 Å cutoff and Fourier spacing of 1.2 Å. The cutoff was applied to van der Waals interactions, which were smoothly attenuated to zero between 11 and 12 Å. Inter-atomic bonds involving hydrogen atoms were constrained by the LINCS algorithm during MD simulations[119].

### Experimental filtration data analysis

The permeate concentration, removal, and mass adsorbed are the main parameters to evaluate the performance of the NF membrane. Debye length was calculated to determine the double layer thickness. All calculations are summarized in Supplementary Table 4. Error analysis is explained in Supplementary Tables 5, 6.

## Data availability

The data that supports the findings of the study are included in the main text and supplementary information files. Raw and source data can be obtained from the corresponding author upon request. Source data are provided with this paper.

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

## Acknowledgements

The Helmholtz Recruitment Initiative is thanked for the IAMT laboratory and project funding. Deutscher Akademischer Austauschdienst (DAAD) provided a PhD scholarship for P.B.T. (program ID 57440921). P.B.P. acknowledges the Research Training Group "Confinement-Controlled Chemistry", which is funded by the Deutsche Forschungsgemeinschaft (DFG) – GRK2376/331085229 as well as the DFG under Germany's Excellence Strategy, Grant No. EXC 2033-390677874 RESOLV. Z.F. and B.M. are grateful for computational resources provided by the e-INFRA CZ project (ID:90254), supported by the Ministry of Education, Youth and Sports of the Czech Republic. DuPont Water Solutions is thanked for supplying NF270 and NF90 membranes. Nitto Hydranautics supplied Hydracore (HY 10, HY 50 and HY 70) membranes. Ben Corry is thanked for the discussion on the hydration layer. Youssef-Amine Boussouga and Alessandra Imbrogno discussed membrane and separation mechanisms.

## Author contributions

A.I.S., P.P., Z.F. and B.M. conceived the project and provided expertise in membranes and micropollutant removal (A.I.S.), spectroscopic characterization of the hydration shells (P.P.), and molecular dynamics (Z.F. and B.M.). P.B.T., M.N.N. and A.I.S. developed the concept of this work. P.B.T. designed and conducted all experiments with the NF membranes and analysis of micropollutants concentration. M.N.N. performed advanced analyses of experimental data and checked all calculations. M.P. and P.B.P. both characterized the hydration layer using Fourier-transform infrared spectroscopy. Z.F. and B.M. constructs models that yield hydration energy results. P.B.T. wrote the manuscript. All authors have revised the manuscript.

## Funding

## Competing interests

The authors declare no competing interests.
