## [Transparent Peer Review file · Nature Communications]

The role of hydration in the removal of glyphosate (GLY) and aminomethylphosphonic acid (AMPA) by nanofiltration membranes

Corresponding Author: Professor Andrea Schaefer

Version 0:

Reviewer comments:

Reviewer #1

(Remarks to the Author)

The authors explored the removal from water of glyphosate (GLY) and aminomethylphosphonic acid (AMPA), which are two micropollutants that tend to hydrate due to their charged and hydrophilic nature, using nanofiltration (NF) membranes of different density. They found that NF membranes with MWCO < 150 Da can efficiently remove the two micropollutants through steric effects, while the removal by membranes with MWCO > 150 Da was achieved through combination of Donnan and dielectric effects, depending on the pH. They also showed that pressure can assist in partial dehydration of the ions, resulting in their reduced rejection. Last, the authors explored the hydration properties of the two molecules in solution using FTIR and molecular dynamics (MD) to highlight the role of hydration in the removal of small and charged micropollutants by NF.

While the study addresses a relevant and contemporary challenge to understand the removal of micropollutants by NF membranes, I cannot recommend its publication in Nature Communications due to its limited novelty, partially improper research design and discussion, and cumbersome structure that makes the reading tiring and unsuitable for the broader readership of Nature Communications. Most notably, while the study combines reasonable experimental work with molecular simulations, it does not provide a clear and novel-enough scientific merit that justifies publication in Nature Communications. Below are several examples that exemplify these downsides. I encourage the authors to improve their manuscript and submit to a more specialized journal.

Specific examples:

1. Title: it is better not to use abbreviations. Instead, and since the names of the specific pollutants is not so important here, the authors can just replace the names with 'small and charged micropollutants...'
2. The structure of the introduction is cumbersome and inappropriate for a research article (especially not a communication). For example, the authors dedicate a full section (1.1) to talk about the exposure risk of GLY and AMPA, which is irrelevant to this study and more appropriate to be elaborated in theses (or papers exploring exposure risks). Also, the specific examples of removal by RO and NF can be shortened to one or two general sentences highlighting the insufficient removal of GLY and AMPA by RO and NF, which invites further research. This is true also for the next sections in the introduction, which are too long for a research paper (e.g., the authors explain the theory of Donnan exclusion in detail).
3. Discussing NF membranes in terms of MWCO is misleading since the removal is based not only on size effects. Manufacturers tend to indicate this parameter, but this is scientifically inaccurate. Also, the numbers indicated by the authors are not clear. For example, NaCl has a MW of 58 Da and its removal efficiency by RO membranes is > 99%, so how come in line 90 the authors state that the MWCO ranges from 100 to 300 Da (perhaps this example demonstrates the problematic use of MWCO when discussing NF and RO membranes).
4. Line 106-108: this is not a full sentence. Also, it is more acceptable to describe transport in RO membranes using the solution diffusion model and not the pore flow model.
5. In lines 113-115, the authors mention Donnan and dielectric effects and then in the next sentences (lines 116-123), the authors discuss different phenomena and get back to talk about Donnan exclusion on line 127. This is another example of

an unfocused and unconcise text.

6. Line 165: the sentence is not clear. What does it mean that "The establishment of the hydration of charged molecules is less evident". In what context? This is a too general statement.
7. Figure 2: why did the authors decide to show the permeate concentration as a function of the permeate volume? What is it scientifically so interesting in understanding the effect of permeate volume?
8. Why did the authors use a dead-end filtration and not a crossflow filtration, which reflects better real NF and RO processes? It seems that this comment/question is related to the previous one. If the authors used a crossflow filtration, their permeate concentration wouldn't change over time (or permeate volume), which makes the measurement much more robust.
9. Lines 260-262: the sentence is not good.
10. Lines 268-273: why didn't charge and dielectric exclusion play a role in the rejection by dense membranes? How can the authors exclude it? In general, the authors try to decouple steric, Donnan, and dielectric effects, which is, perhaps, impossible because they are all dependent on each other (e.g., it is impossible to say that if the rejection is 90%, so the steric is 60% and Donnan is 30%)
11. Following the previous comment, in section 2.3 the authors try to "isolate" charge effects from steric effects by using "loose NF membranes". This is impossible also due to the pore size distribution of the membranes. That is, the NF90 and NF270 membranes possess enough small pores that induce steric exclusion.
12. Section 2.4: the authors claim that pressure can contribute to partial dehydration of ions. However, it is unlikely that the pressure used in this study can affect the hydration shells of ions. The authors will need to show theoretical calculations that pressure can substantially decrease the $P\Delta V$ energy, given that ΔV of dehydration is ultrasmall (i.e. volume of a few water molecules).
13. The authors did not account for concentration polarization in their study. This analysis is critical to understand the REAL performance of NF membranes, since they exhibit relatively high fluxes.
14. Section 2.5 (hydration characterization) is not smoothly connected to the experimental membrane work and may give the feeling that this part was only done to combine simulations with experimental work to increase the impact of the study. Most notably, reading the last paragraph of the abstract on GLY and AMPA hydration does not provide any direct connection to the findings in the filtration experiments.
15. Figures are not of high quality and with compromised quality in several cases (e.g., legend of figure 4c).

Reviewer #2

(Remarks to the Author)

This is a systematic investigation of transport and separation of glyphosate and aminomethylphosphonic acid through a series of commercial membranes. The focus is on performance. I recommend the publication with minor comments:

The full name of GLY and AMPA should be mentioned in beginning of the introduction (line 54 and 55), before line 58.

Line 56-57: The expression "Third world countries" is outdated and should be avoided.

Line 454: should it be $\text{pH} < 10$ for protonation of NH_2 and not $\text{pH} > 10$, correct?

Table 1: It would be more appropriate to right simply "Polyamide" for BW 30 and NF 90 or "Fully aromatic polyamide" for both. "Sulfonated polyethersulfone" is ok for the others without the word "polymer".

For membranes with cut-off < 150 Da, is size the only mechanism? Are all interactions with the polymer excluded?

Could the morphology of the membrane surface be more explicitly commented, for instance summarizing the differences based on the literature or even presenting a collection of SEM images in the supplementary information, comparison all?

Reviewer #3

(Remarks to the Author)

The experimental results are worth publishing as the data provides new insights on the retention of GLY and AMPA by a range of NF membranes.

The conclusions are supported by the work, however, I would like to see a more in depth analysis and different approach of the data and some additional experiments.

More detailed information can be found in the evaluation report.

Version 1:

Reviewer comments:

Reviewer #1

(Remarks to the Author)

I have carefully read the authors' response and appreciate their efforts to address my comments. However, the essence of my original criticism remains. While the study is interesting and relevant, with some novel aspects, I do not find it sufficiently strong in scientific merit, conceptual depth, or overall impact to meet the standards of Nature Communications.

Although the authors provided detailed replies, the actual revisions to the manuscript are relatively minor and do not meaningfully enhance its scientific strength or clarity. Therefore, even with improved writing (which is still needed), the manuscript would still not reach the level of originality and significance expected for this journal.

Below are specific concerns regarding the authors' responses (numbers correspond to those in their rebuttal letter):

- General response: The fact that hydration of micropollutants has not been widely studied (which is debatable) does not, by itself, make the work sufficiently novel or significant for Nature Communications.
- Response 1: The title remains overly general and non-informative, resembling that of a review article rather than an original research paper. It does not convey the specific insight or contribution of the study. While this is partly stylistic, I still recommend making it more focused and descriptive.
- Response 2: The Nature Communications guidelines explicitly prohibit subheadings in the Introduction and emphasize brevity. The current Introduction is long, subdivided, and includes excessive background (e.g., two full paragraphs on contaminant exposure and abundance). Such information could be summarized in one or two sentences. Similar verbosity appears throughout the manuscript, resulting in a presentation more suited to a thesis than to a concise, high-impact research paper.
- Response 8: Although dead-end filtration is experimentally simpler, it is less representative of real membrane systems and more prone to concentration polarization—an important factor in transport studies. The claim that dead-end configuration promotes ion dehydration more than crossflow remains weak and insufficiently supported, even if previously mentioned in ref. 49.
- Response 12: I do not dispute the occurrence of dehydration; rather, I question whether the pressures typical of RO/NF systems (5–70 bar) are sufficient to induce it. The cited studies (refs. 55 and 59) assert this effect but do not demonstrate it quantitatively. The authors were asked to perform simple estimations to evaluate whether such pressures could realistically compress hydration shells, but this was not done. The distinction they draw between pressure effects in bulk solution and within the membrane also remains unclear.
- Response 13: The statement “discern concentration polarization from dehydration” is conceptually unclear—these are distinct and non-comparable phenomena. In addition, describing concentration polarization as “evaluated via concentration at the membrane surface” is incorrect: elevated surface concentration is a symptom, not the cause, of polarization.

Reviewer #2

(Remarks to the Author)
My comments were addressed.

Reviewer #3

(Remarks to the Author)
After re-evaluation, I have the following additional comments

The title is rather narrow focusing on GLY and AMPA, whereas the topic of the paper is more generic. However, there is a lot of attention on GLY and AMPA in the public domain, so I appreciate that authors mention these compounds explicitly in their title.

The readability of the paper has been improved compared to the earlier version. Also the introduction is clearer and somewhat more concise than the previous version, although still somewhat lengthy, e.g. L88-101 provides a lot of detail on performances of different/specific membranes, which can be summarized in a couple of sentences.

In addition, the description of figure 1 can be more concise, e.g. the explanation on the role of zeta potential (L127-152) can be found in standard text books, and does not require such extensive explanation in this paper.

L178-207, can also be more compact focusing less on the details of the methodology and more on what information can be obtained with these techniques and how this relates to the interpretation of dielectric exclusion.

The authors have chosen a dead-end filtration set up to study the retention of GLY/AMP, because more shear force can be generated. Authors claim that the hydrodynamic conditions in the dead-end module are similar to those in cross-flow with reference to previous work that they have done. No further details are given in this paper. Nevertheless, it is well known that the conditions in a dead-end filtration system whereby the shear force is generated by a magnetic stirrer on top of the membrane are significantly different from cross flow filtration, which is used in practice. Therefore, there remains uncertainty in how far the results from one system (dead-end) can be transferred to that of another system (cross-flow). Authors claim that the higher shear force in a dead-end filtration system can lead to more shredding of the hydration shell and therefore less retention. However, the higher shear force will also lead to less concentration polarization and hence higher permeation of ions. I wonder how, with the current experimental setup, these two mechanisms can be decoupled. Also, Donnan and Dielectric effects cannot be fully decoupled up to an extent that one can quantify what part of the rejection is due to steric and what part due to Donnan and dielectric exclusion.

I believe this paper is worth publishing, despite the many remaining questions and uncertainties, as it provides new insights, although not conclusive, on the importance of Donnan and dielectric exclusion in addition to size exclusion of relatively small molecules at different (nominal) MWCO's. Also the chosen molecules (GLY/AMPA) are highly relevant, e.g. for aquatic systems toxicity as well as (drinking) water quality.

However, if the paper doesn't meet the requirements of Nature Communications, then I would recommend publication in e.g.

J. Membrane Science or J. Water Process Engineering.

Version 2:

Reviewer comments:

Reviewer #1

(Remarks to the Author)

The authors have made additional (and important) changes to the manuscript. As my core concerns regarding the novelty remained, I'm leaving the decision on publication to the editors and wishing the authors best of luck.

Reviewer #3

(Remarks to the Author)

I have once more read the submitted paper and I believe the quality has improved compared quite a bit compared to the two previous versions. Also, the introduction is more concise and focused. This paper contributes in obtaining a better understanding (although not conclusive) of the transport mechanisms in nanofiltration membranes and are worth publishing.

REVIEWER COMMENTS

Reviewer #1 (Remarks to the Author):

The authors explored the removal from water of glyphosate (GLY) and aminomethylphosphonic acid (AMPA), which are two micropollutants that tend to hydrate due to their charged and hydrophilic nature, using nanofiltration (NF) membranes of different density. They found that NF membranes with MWCO < 150 Da can efficiently remove the two micropollutants through steric effects, while the removal by membranes with MWCO > 150 Da was achieved through combination of Donnan and dielectric effects, depending on the pH. They also showed that pressure can assist in partial dehydration of the ions, resulting in their reduced rejection. Last, the authors explored the hydration properties of the two molecules in solution using FTIR and molecular dynamics (MD) to highlight the role of hydration in the removal of small and charged micropollutants by NF.

While the study addresses a relevant and contemporary challenge to understand the removal of micropollutants by NF membranes, I cannot recommend its publication in Nature Communications due to its limited novelty, partially improper research design and discussion, and cumbersome structure that makes the reading tiring and unsuitable for the broader readership of Nature Communications. Most notably, while the study combines reasonable experimental work with molecular simulations, it does not provide a clear and novel-enough scientific merit that justifies publication in Nature Communications. Below are several examples that exemplify these downsides. I encourage the authors to improve their manuscript and submit to a more specialized journal.

We appreciate very much your comments, especially your concerns related to the work's novelty, quality of research design and discussion, and the structure of our writing. While the writing will be revised to make it more exciting for diverse readers, we respectfully disagree that the work lacks novelty (although this has been further emphasized) or that the research design and discussion were improper. We have addressed the specific concerns with great care in the responses below and trust that this will enable removing such doubt. The work is exciting and novel indeed.

Regarding the novelty, this study reveals the role of the hydration layer on the transport and removal of small and charged micropollutants by nanofiltration (NF) membranes via meticulous experimentation, spectroscopic characterization, and MD modelling. In recent years, the topic of hydration in ion transport has been much discussed by the community (please note that Schäfer is one of the pioneers in this topic [1]), however, the hydration effect has never been investigated for small and charged micropollutants with multiple charged sites. This is of course very challenging, compared to ions. The innovation of this study has been stressed on page 8, lines 208–216, which now reads:

“The novelty of this study lies in the elucidate of how the hydration of small and charged micropollutants with multiple charged sites (such as GLY and AMPA) affect their transport through NF membranes via spectroscopy, molecular dynamics simulation, and experimental removal results. The role of hydration in combination with dielectric exclusion in the removal will be elucidated by addressing the following research questions: 1) Which membrane shows the strongest evidence of exclusion due to the hydration layer?; 2) How does the hydration layer of GLY/AMPA vary with pH?; 3) Does the hydration strength affect the interaction of GLY/AMPA with membrane material and transport?”

Specific examples:

1. Title: it is better not to use abbreviations. Instead, and since the names of the specific pollutants is not so important here, the authors can just replace the names with ‘small and charged micropollutants...’

Thank you very much for your comment. We refrain from changing the title of the paper, because this paper will appeal to two sets of readers: those who are interested in the removal performance of specific pesticides (glyphosate and AMPA) – and there is a lot of controversy surrounding these specific pollutants, and those who are interested in the mechanistic evaluation of nanofiltration membrane processes (the hydration of charged and hydrophilic molecules). Hence, referring to only “small and charged micropollutants” will make it unclear for the first set of readers as a lot of our work is about micropollutants (which would be the better term, but it is too general). Additionally, some readers (likely from the environmental field) may know of the term AMPA and are less familiar with the full name (aminomethylphosphonic acid), hence we decided to keep both the full name and abbreviation in the title for these readers. Hopefully you can agree with this reasoning.

2. The structure of the introduction is cumbersome and inappropriate for a research article (especially not a communication). For example, the authors dedicate a full section (1.1) to talk about the exposure risk of GLY and AMPA, which is irrelevant to this study and more appropriate to be elaborated in theses (or papers exploring exposure risks). Also, the specific examples of removal by RO and NF can be shortened to one or two general sentences highlighting the insufficient removal of GLY and AMPA by RO and NF, which invites further research. This is true also for the next sections in the introduction, which are too long for a research paper (e.g., the authors explain the theory of Donnan exclusion in detail).

Thank you very much for your opinion. GLY and AMPA are toxic and highly relevant micropollutants, and existing technologies are struggling to remove these, especially AMPA, to the required levels. So, we dedicated a section to discuss the general information about these micropollutants and justify the importance of our work in removing these micropollutants. Indeed, we agreed with the comment that the introduction was long, although the background information is necessary for readers from various fields. We have shortened a few parts of the introduction and removed texts that appeared redundant following further review - we have also reduced the number of background references. The detailed modifications are given in page 4 lines 88–101 which now reads as follows;

“Nanofiltration (NF) and reverse osmosis (RO) membranes with nominal molecular weight cut-offs (nominal MWCOs, which are the molecular weight of solutes of which at least 90% can be retained by the membranes) of 100–300 Da are semi-permeable barriers for GLY and AMPA [2-4]. RO membranes (DOW XLE and Toray THM) with a low nominal MWCO of 100 Da [5, 6] removed 97% AMPA (111 Da) from an initial concentration of 0.7 µg/L and with an applied pressure of 5.8 bar in a pilot system [4]. The removal was high, yet incomplete, and this has been confirmed with NF 300, which is a more permeable membrane with a nominal MWCO of 180 Da. 95% GLY (169 Da) could be removed in a pilot scale system from an elevated initial concentration of 48 mg/L at an applied pressure of 10 bar [2]. Looser NF membranes such as the NFX (nominal MWCO of 150–300 Da) were reported to remove only 80% of GLY and 70% AMPA from an initial concentration of 50 µg/L at 25 bar [3]. High pressures (up to 25 bar) for operation scaled with energy requirements and operational costs [3, 7], which is a challenge for GLY/AMPA removal by RO membranes (0.4–1.7 kW.h/m³ for brackish water desalination and water reuse [8]) compared to NF membranes (0.2–0.5 kW.h/m³ [9, 10]). ”

3. Discussing NF membranes in terms of MWCO is misleading since the removal is based not only on size effects. Manufacturers tend to indicate this parameter, but this is scientifically inaccurate. Also, the numbers indicated by the authors are not clear. For example, NaCl has a MW of 58 Da and its removal efficiency by RO membranes is > 99%, so how come in line 90 the authors state that the MWCO ranges from 100 to 300 Da (perhaps this example demonstrates the problematic use of MWCO when discussing NF and RO membranes).

We appreciate your comments and yes we are quite aware of NF characterization, however, MWCO is relevant when one deals with organic molecules. Indeed, the MWCOs provided by membrane suppliers are only nominal values, based on complicated and error prone characterization. The polyamide-based NF/RO membranes contain pores that are naturally non-uniform in size and even defects [11-13]; several works from our group (IAMT-KIT) have evaluated the effects of this pore size distribution on the retention of hydrophobic and uncharged micropollutants, namely steroid hormones [14-16]. The nominal MWCO values are used in this paper only for the assessment of whether steric exclusion is a governing retention mechanism. If the MW of target pollutants is several times lower than the MWCO (which means the hydrodynamic size of the pollutants is less than the average pore size), steric exclusion will not likely be a governing factor, although the pollutants may still be retained due to the Donnan and dielectric effects at play [17]. The same reason applies to NaCl in your example – salt removal efficiency by NF/RO membranes is contributed by both charge screening and hydration [18]. To enter a membrane ‘pore’, the ions and molecules need to overcome the (Donnan) potential related to electrostatic repulsion [19, 20] and the energy requirement for dehydration [1, 18].

Although the MWCO and pore size distribution were not precisely determined, our main message in this work is that GLY and AMPA are vastly smaller than the MWCOs of most NF membranes evaluated (NF 270 and all the HydraCoRe membranes), and the partial removal of these pollutants should be attributed to a combination of Donnan and dielectric exclusions.

We have revised the texts and modified all the occurrences of “MWCO” to “nominal MWCO” for clarity.

4. Line 106-108: this is not a full sentence. Also, it is more acceptable to describe transport in RO membranes using the solution diffusion model and not the pore flow model.

Thank you very much for pointing out this grammar error and your opinion about transport models, which is subject to a rather heated debate in the field at present. We are, of course, fully aware of those schools of thought.

It is indeed more acceptable to describe transport in RO membranes using the solution-diffusion model, which has been used for decades. While the model is useful to explain phenomena, there are many debates on the extent to which it is based on reality. In the solution-diffusion model, water first partitions into the membrane and then diffuses down a concentration gradient of water within the RO membrane, under the assumption that the membrane is non-porous and instead treated as a medium in which the diffusion of pollutants is hindered. However, the recent research on experimental and computational studies has questioned these assumptions [21, 22]. Firstly, RO membranes are a structurally non-homogeneous layer of crosslinked polymers with areas of lower polymer density as well as holes (*i.e.*, intrinsic pores). Spectroscopic analyses reveal that RO membranes are supported by the sub-nanometer pore structures with the pore size in the range of 0.4–0.5 nm and 0.7–0.9 nm [23], whereas advanced microscopic techniques reveal definite water transport pathways in RO membranes [24]. Secondly, in the solution-diffusion model, the pressure is assumed to be

uniform and is equal to the hydrostatic pressure in the feed water. It has been argued that the hydrostatic pressure across RO membranes should not be uniform but decreases with further distance from the pore surface [21]. The simulation studies with small sections of membrane polymers reveal that there is a pressure drop from the feed side to the permeate side across the membrane [25] (this is coupled with the expectation that the membrane contains pores). Moreover, the solution-diffusion model assumes that water and solute transport independently [26]. However, this coupling between water and solute (pollutant) needs to be further evaluated, and studies on the pollutant hydration (and the shredding of this layer when the pollutant enters the 'pore') are important.

As the pore transport models have their own limitations, no model is able to describe the complex transport phenomena perfectly [27, 28]. As this paper does not need to take side with one of the models (pore flow or solution diffusion), we have decided to removed the controversial information. The sentences in page 4, lines 103–106 now reads as follows.

“NF/RO membranes comprise a composite structure comprising sub-nanometer pores (or sometimes referred to as regions of lower polymer densities), where water and certain solutes pass through [24, 29]. Micropollutants are excluded by NF/RO membranes *via* i) size exclusion, ii) Donnan exclusion, and iii) dielectric exclusion [17]...”

5. In lines 113-115, the authors mention Donnan and dielectric effects and then in the next sentences (lines 116-123), the authors discuss different phenomena and get back to talk about Donnan exclusion on line 127. This is another example of an unfocused and unconcise text.

Thank you very much. The introduction is now rewritten in page 4–5 lines 106–136 to read as follows;

“Size exclusion is based on the relative sizes of the membrane pores and target molecules [30-32]. Removal by size exclusion thus strongly depends on NF/RO membrane pores [11], which are non-uniform in sizes [12, 13], and the molecular characteristics such as shape and orientation [33]. Besides steric exclusion, when the micropollutants like GLY and AMPA enter pores, adsorption on membrane surface materials (such as polyamide) *via* van der Waals interaction [3, 34] and hydrogen bonding [3] may occur. Subsequently, partial transport through membranes pores is driven by a combination of desorption, diffusion, and convection in the pores [17, 35] (Figure 1 D). Charged and hydrophilic micropollutants such as GLY/AMPA have weaker interaction with polyamide than uncharged and hydrophobic micropollutants [36], resulting in low adsorption on membrane material [3]. Hence, GLY/AMPA is less likely to transport through membrane pores by sorption–diffusion, compared to uncharged and hydrophobic micropollutants.

Figure 1. Schematic of the removal mechanisms of GLY and AMPA by NF membrane. Size exclusion (A), dielectric exclusion (B), Donnan exclusion (C), and sorption–diffusion (D).

Charged molecule removal can be governed by other mechanisms, namely charge (Donnan) and dielectric exclusions [37-39], implying the hindered transport through NF/RO compared to uncharged molecules [32, 40]. Charge (Donnan) exclusion indicates the retention due to electrostatic repulsion between the charged ions or micropollutants and the charged membrane surface [41]. The charge at the membrane surface (or the pore section close to the membrane surface) is characterized by the zeta potential (ζ) [42] and the double layer thickness of charge solutes on the membrane surface (Debye length, κ^{-1}) [43, 44]. The zeta potential ζ , is the electrical potential at the slipping plane, the distance from the membrane surface where the electrolyte ions become mobile [45]. The Debye length indicates the reach of electrical potential from the charged (pore) surface [46]. The relative scale of pore radius and the Debye length controls the transport of ions and charged species that are smaller than membrane pores [43, 44]. If the Debye length is greater than the pore radius, charged species at the pore entrance likely repel the charged membrane surface or pore walls and encounter resistance depending on the (pore) surface electrostatic potential [47], increasing the likelihood of removal. If the Debye length is smaller than the pore radius, species in the pore center will be able to pass through the pores with much weaker electrostatic interaction with the pore walls (Figure S1). Naturally, pores are tortuous, which complicates the application of this conceptual model.”

6. Line 165: the sentence is not clear. What does it mean that “The establishment of the hydration of charged molecules is less evident”. In what context? This is a too general statement.

Thank you for your comment. Indeed, this sentence is unclear. The sentence is now rewritten in page 6 line 162–163 to read.

“Compared to the extensive research on the hydration of ions, studies on the hydration of charged molecules are relatively sparse.”

7. Figure 2: why did the authors decide to show the permeate concentration as a function of the permeate volume? What is it scientifically so interesting in understanding the effect of permeate volume?

We appreciate your comment. The permeate concentration was reported as a function of volume because the permeate concentration indeed varies during the course of an experiment, as evident in Figure 2. At first (300–400 mL of permeate volume), the removal was determined by adsorption, and then, as more pollutants in the sorbed phase transport through the membrane via diffusion, the removal was controlled by exclusion. This ‘breakthrough’ phenomenon was more evident and has been reported in great detail in our previous work with steroid hormones – hydrophobic and uncharged molecules [16]. By collecting samples at various permeate volumes, we can identify when removal is no longer likely controlled by adsorption. Additionally, by using fixed permeate volumes in all experiments, we could perform mass balance and compare the adsorbed masses between different membranes and across operating conditions (see Figures 3, 5 and 6). This type of work is very well established in our field and the ‘breakthrough’ phenomenon is really important for micropollutant retention.

8. Why did the authors use a dead-end filtration and not a crossflow filtration, which reflects better real NF and RO processes? It seems that this comment/question is related to the previous one. If the authors used a crossflow filtration, their permeate concentration wouldn’t change over time (or permeate volume), which makes the measurement much more robust.

Thank you very much for your thoughtful suggestions. We selected dead-end filtration due to simplicity of experimental set-up. Dead-end filtration would generate more shear force than crossflow filtration [48, 49], which has been associated to higher permeation of ions due to the shredding of the hydration shell [49]. As we would like to intensify the effect of the hydration, the dead-end mode was selected. Note that we had done comparative studies between our dead-end stirred cell and two crossflow systems with different configurations, and reported that the hydrodynamic parameters and salt retention are comparable to each other and deviate little from the spiral wound module [50]. Hopefully this indicates that our methods are indeed very solid.

The clarification is added in page 22 lines 566–569 as follow.

“The dead-end filtration was chosen as more shear force could be generated than crossflow filtration [48, 49], associated to higher permeation of ions due to the shredding of the hydration shell [49].”

About your comment that the permeate concentration would not change over time in crossflow – we disagree, as we have evidence of the breakthrough behavior of micropollutants (*i.e.* pollutants at very low, sub-micrograms to several micrograms per liter concentrations) in both dead-end and cross-flow filtration, and this depends on the adsorption strength of the membrane towards the micropollutants. Please refer to our previous work with steroid hormones for your reference [16]. The difference in the stirred cell is the concentration that occurs in the cell with time, but this can be accounted for and is not so dissimilar of the concentration that occurs along a spiral wound membrane module.

Additionally, the dead-end filtration allows us to minimize the error in mass balance. In crossflow, the mass loss of micropollutants is expectedly higher as a significant amount of micropollutants are lost in the tubing required in the circulation. The use of dead-end filtration will result in more accurate mass balance calculation and we can identify adsorbed mass, if any, for elucidating the retention mechanisms. Note that the IAMT lab has an array of different set-ups that allow to choose the best system for a given task.

9. Lines 260-262: the sentence is not good.

Thank you. The sentence is now revised in page 10 lines 256–259.

“To ascertain to what extent adsorption and size exclusion contribute to removal mechanisms (Figure 3), filtration of GLY and AMPA at pH 8 was performed with six different membranes (BW 30, NF 90, NF 270, HY 70, HY 50 and HY 10). The active layers of BW 30, NF 90, and NF 270 are made of polyamide, and those of HY 70, HY 50 and HY 10 are made of sulphonated polyethersulfone.”

10. Lines 268-273: why didn't charge and dielectric exclusion play a role in the rejection by dense membranes? How can the authors exclude it? In general, the authors try to decouple steric, Donnan, and dielectric effects, which is, perhaps, impossible because they are all dependent on each other (e.g., it is impossible to say that if the rejection is 90%, so the steric is 60% and Donnan is 30%)

Thank you very much for your comments. Indeed, Donnan and dielectric effects are coupled, which increases the effective size of the pollutants, resulting in their retention by NF/RO membranes, even when the pore sizes are larger than the intrinsic sizes of the pollutants [37]. For highly dense membranes (BW 30) due to the smaller average pore diameter of BW 30 (0.41–0.51 nm) compared to the intrinsic hydraulic diameter of GLY and AMPA (0.62 nm and 0.49 nm, respectively [51], size exclusion is likely the decisive factor. However, because the membrane contains larger pores and potentially defects, charge and dielectric exclusions cannot be neglected, but due to a lack of characterization, we cannot ascertain whether they play a significant role. The discussion has now been revised to reflect the above discussion on pages 10–11, lines 267–279, to read.

“BW 30 (average pore diameter 0.41–0.51 nm [51]), NF 90 (0.44–0.63 nm [51]), and NF 270 (0.57–0.89 nm [51]) membrane with nominal MWCO < 200 Da [50, 52] can remove 85–88% GLY and AMPA (Figure 3). The removal of GLY/AMPA by BW 30 membrane can be attributed to a combination of adsorption and size exclusion. The incomplete removal (< 100%) was assumed to be due to the inhomogeneity of pore sizes of the commercial membranes [12, 13], and concerted processes of adsorption-desorption and gradual transport in the sorbed phase through the larger pores [35]. Even though the pore diameter of BW 30 (0.41–0.51 nm) is smaller than the hydraulic diameters of GLY and AMPA (0.62 nm and 0.49 nm, respectively [51]) for effective size exclusion, it must be noted that the NF membranes contain larger pores and defects, which allow the breakthrough of GLY and AMPA [12, 13]. Additionally, the transport of micropollutants through the membrane pores also depends on the molecular shape and orientation [33]. While size exclusion is the decisive factor leading to the retention of GLY/AMPA by BW 30, Donnan and dielectric exclusions cannot be neglected in the pollutant transport through the dense membranes and hence their partial removals.”

11. Following the previous comment, in section 2.3 the authors try to “isolate” charge effects from steric effects by using “loose NF membranes”. This is impossible also due to the pore size distribution of the membranes. That is, the NF90 and NF270 membranes possess enough small pores that induce steric exclusion.

The pore size distribution is indeed an intriguing aspect of membrane technology, and interesting new findings can be found day after day! Recently, it has been reported that water and solutes selectively transport through the bigger pores and defects (or, in RO terminology, areas with lower polymer density [13, 24]) – which explains our bias towards the pore-flow mechanism. It is noted that, while the average pore size of NF 90 is in the range of 0.44–0.63 nm, and that of NF 270 is 0.57–0.89 nm[51], GLY (0.62 nm) and AMPA (0.49 nm) tend

to transport through pores at the higher end of the pore size distribution. Hence, we agree that one cannot fully exclude size exclusion, following the previous answer. Although we selected NF 90, NF 270, HY 50 membranes for the study, Donnan and dielectric exclusions, only the HY 50 with a higher nominal MWCO of 1000 Da is expected to have little contribution from steric exclusion because of the relatively larger average pore size (1.58 nm) than the hydrodynamic diameters of GLY and AMPA.

12. Section 2.4: the authors claim that pressure can contribute to partial dehydration of ions. However, it is unlikely that the pressure used in this study can affect the hydration shells of ions. The authors will need to show theoretical calculations that pressure can substantially decrease the $P\Delta V$ energy, given that ΔV of dehydration is ultrasmall (i.e. volume of a few water molecules).

Of course, the pressure in the bulk phase has a small effect on the hydration layer value, and the related enthalpic contribution is negligible, as mentioned by the Reviewer. However, the hydration shell size of the charged molecular species is relatively large due to the dipolar orientation of surrounding water molecules and increased hydrogen bonding. The effective radius of these shells is then comparable to the membrane pore sizes and hinders the pore entering and flow inside them. Often, the molecules need to be partly dehydrated to pass through membrane pores [53, 54]. The dehydration is a result of the competitive interactions of molecules with liquid water and polar groups of polymeric NF membranes. These interactions are energetically of the same order of magnitude; however, the hydration-layer disruption or even partial removal and substitution by membrane residues is not a barrierless process and could be slow. The kinetics could be enhanced by applying pressure in pressure-driven membrane filtrations, which provides the needed activation energy for the partial hydration [55, 56]. This phenomenon has been reported for solvated ions (and potentially charged molecules) from computational [57, 58] as well as experimental studies [55, 59]. Our note on pressure effects is thus related to these transport phenomena specific to NF membranes, not to bulk-liquid pressure.

To investigate these phenomena in detail, we built a computational model of a pore in piperazine polyamide NF membrane and performed MD simulations of GLY and AMPA molecules movement there. Snapshots of these simulations, as an example, are shown in Figure 2 A, B. While this information is added here, this will not be incorporated in the manuscript, but we will discuss these data in another study. However, here we show the models to demonstrate the typical size and complexity of these systems. The molecules (shown in orange) are surrounded by water (blue) and interact strongly with the counterions present in the solution. Their hydration layers have a radius of a similar size to the pores (\sim nm), and the molecules interact with dipolar groups of the polymers. This is shown on radial distribution functions (RDF) in Figure 2 C, D, where the distance between GLY/AMPA and oxygen in water and membrane polymer, respectively, are detected. From the peaks between 0.15 and 0.2 nm, it is clear that the water molecules in the hydration layer are partially removed and replaced by carbonyl oxygens of the polyamide chains. All these interactions, besides the steric effects, hinder the molecular flow, which could be promoted by applying the external pressure.

Figure 2. MD snapshots of computational models representing a pore of piperazine polyamide NF membrane (grey lines) filled with water (blue) and containing GLY in -2 charge (a) and AMPA in +1 charge (b), shown in orange, while the green spheres indicate the counterions. Interactions of these molecules with water (c) and polyamide (d) oxygens were detected by radial distribution functions (RDF) at different charges (i.e., pH).

13. The authors did not account for concentration polarization in their study. This analysis is critical to understand the REAL performance of NF membranes, since they exhibit relatively high fluxes.

Thank you very much. This is an important comment that has inspired a lot of additional analysis to attempt to discern concentration polarization from dehydration. This is not an easy task and the following discussion is added in page 16–17 lines 412–434, to now read (an additional graph was added to the manuscript to include this analysis);

“The rejection of GLY and AMPA is also controlled by concentration polarization [16]. Therefore, the concentration polarization is evaluated via solute flux, concentration at the membrane surface, and mass adsorbed of the NF membranes (Figure 3).

Figure 3. Concentration at membrane surface, solute flux, and real removal as a function of flux with GLY/AMPA removal by NF 90, NF 270, HY50 membrane (initial GLY/AMPA concentration 1 $\mu\text{g/L}$, 1mM NaHCO_3 , 10 mM NaCl, pH 8, 20 $^\circ\text{C}$).

As the pure water flux increases from 25 to 100 $\text{L/m}^2\cdot\text{h}$, the theoretical concentration at the membrane surface increases by 4 to 20 times for NF 270, and 2 to 20 times for NF 90, from the feed concentration (100 ng/L). This means there was strong concentration polarization linked to the high retention by NF 270 and NF 90 (80–90%, Figure 6). However, for HY 50, the theoretical concentration at the membrane surface varied between 1470 and 4077 ng/L , and does not deviate strongly from the feed concentration. This corresponds to lower retention by this membrane (from $86 \pm 3\%$ to $28 \pm 8\%$ for GLY, and from 27 ± 11 to $7 \pm 5\%$ for AMPA, with clear decreasing removal trend at increasing flux). The permeate GLY/AMPA flux shows a slight increase trend with pure water flux for NF 270 and NF 90, but there was stronger increase for HY 50 membrane. While retention by HY 50 membrane can be attributed to an interplay of Donnan and dielectric exclusions (and a degree of steric exclusion by the small pores of the pore size distribution), the trend in solute flux is likely attributed to dielectric exclusion and the increased hydration shell shredding at higher pressures. For NF 270 and NF 90, the contribution of dielectric exclusion is unclear. The real removals (calculated from the difference between the permeate concentration and the concentration at the membrane surface) of NF 90 and NF 270 were 92–99% for both GLY and AMPA, while the real removal of HY 50 was 82–90% for GLY and 52–80% for AMPA.”

14. Section 2.5 (hydration characterization) is not smoothly connected to the experimental membrane work and may give the feeling that this part was only done to combine simulations with experimental work to increase the impact of the study. Most notably, reading the last paragraph of the abstract on GLY and AMPA hydration does not provide any direct connection to the findings in the filtration experiments.

Thank you very much for your insightful comment. You are correct that the spectroscopic analysis and MD serve to reveal the hydration effects in GLY and AMPA where the charged sites vary with pH. We have planned for further works to characterize the hydration layer of GLY/AMPA at the membrane surface via FTIR-ATR, as well as performing non-equilibrium MD modelling for GLY/AMPA transport in membrane polymers. These tasks will require significant method development efforts and we plan to carry on these in our next collaborative publication. The spectroscopy and MD characterizations indicate varied extents of hydration at different pH, not the sizes of the hydration layer but the strength of the hydrogen bonding network. This may correspond to the different removals by the looser NF membranes (HY 50), although we acknowledge the interplay of Donnan and dielectric exclusions in real scenarios.

The discussion on hydration characterization was added with the experimental results in page 17–21 lines 436–532.

“FTIR-ATR was used to characterize the structuring of water molecules in the hydration layer, and quantify the strength of hydrogen bonding between GLY/AMPA molecules and water molecules. The simplified MCR routine applied to FTIR-ATR spectra was used to characterize the hydrogen bonding between GLY/AMPA molecules and the bulk of water molecules.

Figure 4. IR-solvation shell spectra of GLY (A) and AMPA (B) at pH 4-12 (GLY/AMPA concentration 100 mg/L). Distributions of hydrogen-bond distances between water ((C) GLY in charge states +1, 0, -1, -2, -3; (D) AMPA in charge states +1, 0, -1, -2,); (E) Hydration shell distribution functions of water around GLY and AMPA. The blue areas indicate the positions of the oxygens, while the hydrogen position distributions are shown in silver. The distributions were collected from 5000 samples equally spaced along the 10 ns MD trajectories.

The measurement was taken in Milli-Q water at a concentration of 100 mg/L (100000 times higher than the GLY/AMPA concentration in the filtration experiments of 1 $\mu\text{g/L}$) as a function of pH (4, 6, 8, 10, 12) (Figure 4 A, B). In addition, the distribution of hydrogen bonding distance between water molecules was simulated by MD at different GLY and AMPA charges to be compared with the results from FTIR-ATR measurements (Figure 4 C, D). The distributions were collected from 5000 samples equally spaced along the 10 ns MD trajectories. The hydration shell distribution of water around GLY and AMPA molecules is visualized at different charges, as shown in Figure 4 E.

In a closer look, the broad O–H vibrational band covering ~ 3000 to 3700 cm^{-1} experienced a notable peak splitting and shift to both higher and lower frequencies was observed for both GLY and AMPA when decreasing the pH from 8–12 to 4–6. These shifts reflect changes in the hydrogen-bonded network around the solutes, where at the higher pH values, some water molecules are involved in a more structured hydrogen-bonded complex, and the remaining water molecules become less structured. This difference is driven by the charge state of GLY and AMPA going from -1 to -3 for GLY and 0 to -2 for AMPA, further affirmed and further clarified by the MD simulations (Figure 4 C, D). The distance between oxygen in water molecules was 0.28 nm at charge $+1$, 0 , and -1 for GLY (corresponding to pH 2–4), and the same (0.28 nm) at charge $+1$ and 0 for AMPA (corresponding to pH 2–4). For GLY at charge -2 and -3 (corresponding to pH 6–12) and AMPA at charge -1 , and -2 (corresponding to pH 6–12), the distance between oxygen in water molecules was smaller (0.26 nm). The longer hydrogen-bond distances between water (at low pH) indicate a larger separation between water in the hydration shell and bulk water. Therefore, the stronger/more structured hydration shell of GLY and AMPA (stronger interaction between GLY/AMPA with water molecules) is observed at lower pH, which is comparable with the FTIR spectrum. The overall weakening of hydrogen bonding partly results in lower removal of GLY at lower pH, which was observed in NF 270 and NF 90, and most evidently in HY 50 membranes. At higher pH, the hydration shell is potentially easier to shred when the micropollutants enter the NF pores. The hydration shell distribution of water around GLY and AMPA molecules as a function of charge was simulated by MD (Figure 4 E). More water was found to be in interaction with GLY/AMPA, forming a bigger hydration layer at higher pH than at lower pH, which can be linked to better removal of GLY/AMPA at higher pH in NF membrane filtration (Figure 5). However, the diameter of the micropollutant plus hydration shell appear to vary little at different pH and in the order of 1.6 nm , which means dielectric exclusion does not contribute to any retention if the membrane pores are larger than the hydration shell sizes (e.g., for the case of HY 10, see Figure S5). In contrast, there is clear variation in the number of water molecules involved in hydration, which is an indicator of hydration shell strength. In the case of GLY, an increase in water density around the PO_3 group was observed with the deprotonated in charge state from -1 to -3 (corresponding to the increase of pH from 4 to 12). A similar effect was observable in AMPA, where the solvation shell around PO_3 is also denser at high pH, with the charge changing from 0 to -1 with pH from 4 to 12. The hydration layer around the PO_3 group was more structured than the hydration of the carboxylate group (COO^-) in GLY. The hydration shell around the COO^- group remained the same with pH as the COO^- group was deprotonated at all pH levels (Figure S2). At $\text{pH} < 10$, the central NH_2 group of GLY interacts with water molecules, which can be seen on the blue rings around the central part of the molecule, indicating the oxygen positions that are hydrogen-bonded to NH_2 . This may correspond to the spectroscopic shift towards low frequencies in Figure 4 A, B where water molecules around these rings are more structured, while water molecules outside these rings are less structured. This water ring was not observed at $\text{pH} > 10$ due to the protonation of the amine group (Figure S1). The difference in the density in the hydration shell between GLY and AMPA explains the higher removal of GLY compared to AMPA in NF at the higher pH, which is complementary to the results of hydrogen bond distances (Figure 4 C, D). This finding also agrees with experimental results that the removal of GLY and AMPA decreases with increasing pH (Figure 5), which could be partly attributed to the strengthening of the hydration shell at NF (especially HY 50) pores. To give insight into the interactions of GLY/AMPA molecules with water, the radial distribution functions (RDF) were calculated from the classical MD trajectories. The RDF of GLY and AMPA were computed for water oxygen and hydrogen interactions with the phosphate group (Figure 5), the carboxyl group of GLY, and the amino group of AMPA (Figure S6).

Figure 5. Radial distribution functions (RDF) detect interactions between GLY phosphate groups with water oxygen (A) and hydrogen (B) atoms; and AMPA phosphate groups with water oxygen (C) and hydrogen (D) atoms.

The phosphate group exhibits similar interactions with GLY and AMPA, where RDF results changed due to the protonation of the phosphate moiety of GLY and AMPA. The water oxygens are hydrogen-bonded to the two hydrogens in the PO_3H_2 groups in the protonated +1 form, indicating the oxygen of water pointed to the phosphate group of GLY and AMPA, resulting in a strong $\text{P-OH}\cdots\text{OH}_2$ hydrogen bonding (Figure 5). For both GLY and AMPA, at the higher charge states, there was a peak below 0.2 nm in the RDF of phosphate groups with water oxygen (Figure 5 A and C) and a peak at longer distances in the RDF of phosphate groups with water hydrogen (Figure 5 B and D). These result indeed from the $\text{P-OH}\cdots\text{OH}_2$ hydrogen-bond for both GLY and AMPA, which in turn result in the low-frequency peak in the solvation shell spectra at low pH (more structured hydration shell) (Figure 4 A and B). At the lower charge state found at higher pH, the phosphate is deprotonated and there is no such strong hydrogen bond, resulting in the high-frequency peak in the solvation shell spectra at low pH (Figure 4 A and B). The opposite water arrangement (hydrogen pointing to the phosphate oxygens) is typical for the deprotonated -2 and -3 forms, resulting in a $\text{P}\cdots\text{HOH}$ hydrogen bonding. The partly deprotonated phosphate in the 0 and -1 forms exhibits both kinds of interactions with water from the first hydration layer. Similar trends can be seen on the RDF of the carboxyl group, which interacts with the water oxygen *via* hydrogen molecule at the +1 and 0 charges (corresponding to low pH < 4 when the carboxyl group is protonated). At higher pH, the carboxylate group is deprotonated showing the charge -1 to -3 , forming hydrogen bonding between the carboxylate oxygens and water hydrogens. In the protonated +1 charge of AMPA (pH < 2), the hydration layer around the PO_3H_2 interacts mostly *via* hydrogen with the water oxygens (Figure 5), and the same arrangement is around the NH_3^+ group (Figure S6). As the AMPA deprotonates and gets charges 0 , -1 , and -2 , respectively, the water molecules around phosphates are rotated where hydrogen molecules point to PO_3 oxygens (each of oxygen in the PO_3 group interacting with one water

molecule). Meanwhile, the water interaction with NH_3 is weaker, unless the group gets deprotonated in charge -2 ($\text{pH} > 10$), where the amine group interacts with nearby water hydrogen. Overall, the graph shown stronger interaction between the phosphate group and water molecules at higher pH linking to the increased hydration shell strength; this, together with any Donnan exclusion at the NF pore surface, partly results in better removal of both GLY and AMPA at higher pH.”

15. Figures are not of high quality and with compromised quality in several cases (e.g., legend of figure 4c). Thank you. The figures are now fixed in page 12 lines 320–325.

Figure 6. Membrane pore diameter (A) and Debye ratio (λ) (B) for NF membranes as a function of nominal MWCO ($\kappa^{-1} = 3 \text{ nm}$ at $C_{\text{NaCl}} = 10 \text{ mM}$ and $20 \text{ }^\circ\text{C}$), and schematic illustration of the interplay between the Debye length of a charged molecule and an idealized membrane pore: relatively high retention at low MCWO and lower retention at high MWCO. (C) Potential at the center of the NF pores.

Reviewer #2 (Remarks to the Author):

This is a systematic investigation of transport and separation of glyphosate and aminomethylphosphonic acid through a series of commercial membranes. The focus is on performance. I recommend the publication with minor comments:

The full name of GLY and AMPA should be mentioned in beginning of the introduction (line 54 and 55), before line 58.

Thank you. The full names have been added in the Introduction, page 3.

Line 56-57: The expression “Third world countries” is outdated and should be avoided.

We have removed this term in the introduction and changed to “some countries”.

Line 454: should it be $\text{pH} < 10$ for protonation of NH_2 and not $\text{pH} > 10$, correct?

Thank you very much for pointing out this typing error. We have fixed this.

Table 1: It would be more appropriate to right simply “Polyamide” for BW 30 and NF 90 or “Fully aromatic polyamide” for both. “Sulfonated polyethersulfone” is ok for the others without the word “polymer”.

Thank you for your suggestion. Table 1 is now updated in page 23 lines 590–595.

“Table 1. NF membrane types and properties (MWCO, permeability, pore diameter and isoelectric point (IEP) [50, 52, 60, 61]. NF 270 and NF 90 surfaces have a net positive net charge at IEP < 4 and a negative net charge at pH > 4. HY membrane surface has a negative net charge at all pH 2–12.

No	Membrane type	Company	Nominal MWCO (Da)	Permeability (L/m ² .h.bar)	Pore diameter (nm)	Zeta potential pH 8	Active layer materials
1	BW 30	DuPont, USA	80–120 ^a	4 ± 1 ^a	0.41–0.51	-40 ^e	Polyamide
2	FilmTec NF 90		90–180 ^b	8 ± 2 ^b	0.44–0.63 ^b	-49 ^b	Polyamide
3	FilmTec NF 270		150–340 ^b	14 ± 2 ^a	0.57–0.89 ^b	-116 ^b	Semi-aromatic piperazine-based polyamide
4	HydraCoRe 70 (HY 70)	Nitro – Hydranautics, USA	600–720 ^c	3.1 ± 0.3 ^d	1.21–1.31	-33 ^d	Sulfonated polyether sulfone
5	HydraCoRe 50 (HY 50)		1000 ^c	7.8 ± 0.5 ^d	1.58	-33 ^d	Sulfonated polyether sulfone
6	HydraCoRe 10 (HY 10)		3000 ^c	58 ± 8 ^d	2.83	-38 ^d	Sulfonated polyether sulfone

^a Cai *et al.* [52], ^b Imbrogno and Schäfer [50], ^c Nominal value provided by supplier (Nitro – Hydranautics), ^d Bousouga *et al.*, ^e Idil Mouhoumed *et al.* [61], ^f not determined as IEP is below 2.

“

For membranes with cut-off <150 Da, is size the only mechanism? Are all interactions with the polymer excluded?

For membrane with nominal MWCO < 150 Da, size exclusion is the governing mechanism, although we had addressed the concerns from reviewer 1 about using this nominal expression in the discussion of mechanisms. Indeed, the NF membranes show pore size distribution, and the presence of larger pores and defects resulted in lower-than-expected removal of molecules that are larger than the (average) pore sizes. We do not omit polymer interactions and state the adsorbed mass (which is the mass of micropollutants adsorbed to the membrane polymers, which are negligible to low. Please refer to our response to Reviewer #1, Query #10 for more detail.

Could the morphology of the membrane surface be more explicitly commented, for instance summarizing the differences based on the literature or even presenting a collection of SEM images in the supplementary information, comparison all?

The morphology of the NF membranes is well studied in previous papers as these membranes are commercial membranes [62-65], so we do not include these characterizations and SEM images in this paper.

Reviewer #3 (Remarks to the Author):

The experimental results are worth publishing as the data provides new insights on the retention of GLY and AMPA by a range of NF membranes. The conclusions are supported by the work, however, I would like to see a more in depth analysis and different approach of the data and some additional experiments.

We appreciate very much your comment.

More detailed information can be found in the evaluation report.

Removal of glyphosate and aminomethylphosphonic acid (AMPA) by nanofiltration membranes: the role of hydration

In this paper Trinh et al. study the retention of glyphosate of AMPA by a range of nanofiltration membranes with different sizes and different separate layer (chemistries) for the membranes. The main message of the paper is that besides size exclusion, Donnan exclusion and dielectric exclusion are both important parameters for membranes with larger pore sizes. Regarding the paper, I have the following comments.

1. L 76: it is strange to base the amount of tonnage of organophosphate chemicals in agriculture in 2025 on a prediction from 2020. Is there no current data available of how much organophosphates have been used more recently, e.g. in 2024 or even better the first half of 2025.

Thank you very much for your suggestion. Indeed, in 2025 organophosphate chemicals are still produced in parge quantity, which raises environmental and health risk concerns. The recent data on GLY has been updated in page 3 lines 74–78 to read.

“Nevertheless, with the extensive annual usage of 600–750 thousand tones in agriculture and estimated to increase to 740–920 thousand tones by 2025 [66], the concentrations of organophosphate chemicals, particularly GLY, in the water environment (surface water, ground water, and sea water) are expected to further increase [67]. In 2024, 1.2 million tons glyphosate was produced for the market [68].”

2. L 132: It mentions that the slipping plane is the distance from the membrane where the electrolyte becomes mobile. This is not correct, beyond the slipping plane the water molecules become mobile. Electrolyte ions can also be mobile behind the slipping plane due to surface conductivity. Instead of characterizing the surface charge by zeta potential it would be better to determine the (total) charge of the membrane. Also the zetapotential is a function of the electrolyte concentration, the lower the concentration the higher the potential. It remains unclear at what concentrations the zeta potential has been measured and also with what method the surface charge has been calculated/derived from the zeta potential.

Thank you very much for your insightful comments. The zeta potential measurments of the membranes were not a part of this work but in previous research [60]. The zeta potential (ZP) of the membranes was calculated from the streaming potential measurements using an electrokinetic analyser (SurPASS™ 3, Anton Paar, Austria) and following the method of Luxbacher [69]. The solution of 10 mM NaCl (VWR chemicals, purity ≥99.9 %, Germany) was used as the electrolyte, with varying HA concentration from 1 to 20 mgC/L at a temperature of 25 ± 2 °C. The pH of the electrolyte solution was adjusted by an integrated dosing unit using 50 mM HCl or NaOH. In the adjustable gap cell, the membrane samples were placed in between two rectangular holders (2

cm by 1 cm), facing each other with a gap height of $100 \pm 2 \mu\text{m}$. The pressure was decreasing in the range of 50–300 mbar between the inlet and the outlet of the cell, corresponding to an average flow rate of 50 mL/min.

3. L135 etc.: I would rather prefer to see an explanation in terms of Donnan potential, instead of electrical resistance due to the Debye length. The main parameter is the relative concentration of the ions in the membrane relative to that in the bulk.

Thank you very much. The Donnan potential on the membrane surface can be calculated. It is impossible to experimentally calculate the Donnan potential in the membrane pores. The contribution of Donnan exclusion in NF membrane was simulated in other study [70] which is outside the scope of this research.

4. L147: Equalizing Ψ_d with the ξ potential is not correct. Normally the ξ potential is lower than the Stern potential, especially for polymeric surfaces as membranes or rough surfaces. It is important to note that even at distances larger than the Debye length, the potential is still not zero. At the Debye length the potential has dropped by $1/\kappa$. I would recommend to make reference to standard books about double layer theory. There is no need to explain this in detail in this paper and in any the case the given explanation is too simplistic.

This is indeed correct. We have corrected the texts according to your comments, that we cannot equalize Ψ_d (Stern potential) with the zeta (ξ) potential. It is noted that we could not measure the Stern potential directly at the membrane surface where ions are immobile, but only the ξ potential. The measurable ξ potential would be more useful in ion transport characterization because the ions at this plane are mobile. To ascertain the effect of pore size on the contribution of Donnan exclusion, we have assumed that the ξ potential at the membrane surface is applicable to the pore entrance, and ii) the pore size is uniform. The latter is indeed false because NF pores are non-uniform by nature, and it is challenging to directly measure the pore sizes (note that the reported average pore sizes are based on models). Hence, we have also assumed that iii) the average pore diameter is referred to the distance between the two slipping planes characterizable with the ξ potential.

Additionally, we have restated that the potential is not zero but decreases significantly (by around 2.8 times, or $1 e$ at the Debye length distance, to further emphasize that Donnan exclusion is relevant anywhere in the pore space.

5. Fig. 2 and L 249-253: The adsorbed amount of GLY and AMPA is calculated from the mass balance (aa given in table 2), under dynamic conditions. However, I would rather prefer to see independent adsorption isotherms, whereby the adsorbed amount is measured for a range of equilibrium concentrations.

Thank you for your comment. In this paper, the adsorption isotherm was not studied as it was not the focus of this paper, partly because the adsorption of GLY and AMPA by the membrane is very low. The mass balance shows such minimal adsorption, and changing the concentrations to attain adsorption isotherms (where experiment protocols, including much longer experiment time, will need to be revised to be able to ascertain the adsorption equilibrium) will make this paper unfocused.

6. In figure 3 and also fig 4 it is puzzling to relate the datapoints to the actual membranes. The text mentions membrane (names) and the graphs shows MWCO. This makes it all very puzzling. For example in fig 3 at the 3 low MWCO, which datapoint corresponds with which membrane? For the tighter membranes it was apparently not possible to obtain fluxes of 50 L/m².h., the flux was only 30 L/m².h. This shortcoming makes it difficult to

compare the results for the different membranes, especially as the authors want to study the effect of the membrane flux on the extension of the hydration layer, and hence retention. The pore diameter of the tighter membranes (BW 30, NF 90 and NF 270) varies from 0.46nm to 0.68nm, which is more than 45% increase. Nevertheless, the results in fig. 3 hardly show any 2 van 3 difference in performance. This is highly puzzling. Also the hydraulic diameter of GLY and AMPA (resp. 0.62nm and 0.49nm) is larger than the pore diameter of BW30. I therefore would expect the highest retention for this membrane, which should be close to 100%.

The rejection of GLY and AMPA is contributed by size, charge, and dielectric exclusion. There is no clear separation between the three very dense membranes (BW 30, NF 90 and NF 270) and it is only possible to reveal dielectric effects in looser membranes (HY series). Please refer to Reviewer #1, Query #10 for more detailed answer.

7. L277: why would the molecular orientation have an effect on the retention? I assume that if an omp is small enough it will pass the membrane anyway independent of orientation.

Thank you very much for your comments. Indeed, molecular dynamics (MD) [71], density functional theory (DFT) simulations [72], and quantum mechanics (*ab initio*) calculations [73] have suggested that zwitterionic GLY adopted a linearized structure in water solution, but the MD in our work reveal largely spherical hydration shells. Although molecular orientation in theory affects the rejection by NF membranes, taking into account the hydration effect, the molecular orientation plays a less pronounced role.

8. L297-303: see previous comment, I would rather prefer to explain this in terms of Donnan potential, which determines the ion concentrations in the pores. The way it is described now remains very qualitative. Also L302 states that 'the strength of electrostatic interactions will depend on the potential in the pore space'. Which interactions are meant here. Is it the interaction between the omp and the membranes, or is this all about the ion concentration in the pore which is determined by the Donnan potential?

Thank you very much for your comment. We meant the electrostatic repulsive interaction between the MPs and the membrane surface (notably the pore entrance), as this repulsion prevents the MPs from entering the pores (Donnan exclusion). The theoretical Donnan potential when the ion is in the middle of a pore with an average diameter stated in Table 1 has been present in Fig 4 B.

9. Fig 4: B is hard to read because of the size of the figure. Also I am puzzled by what the insert should demonstrate.

We have fixed this figure, please refer to Reviewer #1, Query #15.

10. L317: the presented zeta potentials are meaningless, as long as it is not clear how they are measured and at what salt concentrations.

Please refer to query #2.

11. L337: states that 'No adsorption is measured at pH 2', however, fig. 5 I, does show low adsorption at pH 2.

Thank you for your comment. Although there seems to be some adsorption shown in Figure 5, the error bars are large and extend to below the zero value. This means adsorption was extremely low (below 0.5 ng/cm²),

and we could not ascertain if there was indeed mass loss or not. However, we have revised the sentence to clarify that “adsorption was low and appeared insignificant at pH 2, as the error bars are larger than the adsorbed mass values”.

12. Fig 5 shows a strong effect of membrane surface chemistry (polyamide vs polysulfone) on adsorption and removal. In l338-344 an attempt is made to explain this observation, but this remains rather opportunistic and superficial. I miss a more in depth analysis, e.g. what is the difference in surface groups, what is the density of these groups and how can they interact with the AMPA/GLY, how strong are these interactions, etc.

Thank you very much for your comment. Because the polyamide membranes and the sulphonated polyethersulfone membranes have different average pore sizes as well as charge density on the surface (lower charge density corresponds to lower zeta potential measured at the same pH), the effect of chemistry on observed adsorption and removal is not clear. From the response to Reviewer #1's Comment #12, we have learnt from MD simulation that the water molecules surrounding the GLY and AMPA are replaced by the oxygen of polyamide carboxylic groups – implying that GLY and AMPA are adsorbed to polyamide materials via hydrogen bonding. Polyamide membranes contain the repeat unit R-NH-CO-R', where the N-H and C=O bonds are available for hydrogen-bonding. The N-H group is a strong hydrogen-bond donating group and the C=O is a good hydrogen-bond accepting group. Such groups are known to form medium and strong hydrogen-bonds to carboxylic acid groups [74, 75] such as those on GLY and phosphate groups of both GLY and AMPA. In the SPES structure, the sulphonate group can also induce strong hydrogen bonding typically as H-acceptor [76]. The ether oxygen is a weak hydrogen-bond acceptor and the sulphonate group are medium hydrogen-bond acceptors comparable to the C=O. Unlike carboxylates, sulphonates are not capable of donating hydrogen-bonds to GLY and AMPA. Additionally, the density of this group may vary from the density of carboxylic group in polyamide membranes. Because of the complexity of size, charge and functional group density effects, the individual functional group chemistry cannot be linked to the difference in adsorbed masses between the two types of membranes.

13. L365: Why is the higher removal of GLY (compared to AMPA) due to the higher negative charge of GLY. Would the removal not be higher because of the higher Mw of GLY?

Thank you very much for your comments. We had taken the charges of GLY and AMPA into account, and naturally the lower (more negative) charges repel the negative surface of the NF membranes more strongly. In the context of this sentence (only for HY 50), while size exclusion cannot be ruled out, it is not decisive factor as the nominal MWCO of the membranes is much higher than the MW of GLY and AMPA. We have addressed the relevance of size exclusion in response to Reviewer #1's comment #3.

Rewrite in page 14 lines 370–378:

“The increase in removal by pH observed with HY 50 membrane can be due to two phenomena: i) stronger electrostatic repulsion between GLY/AMPA and the membrane surface at increasing pH, and ii) denser hydration layer with higher charge density of GLY/AMPA, resulting in larger effective hydrated diameters. Size exclusion, if any, is not a decisive factor as the nominal MWCO of HY 50 membrane (1000 Da) is much higher than the molecular weights of GLY (169 Da) and AMPA (111 Da). The removal of GLY was always higher than AMPA for all pH values, which could be attributed to the higher negative charge of GLY than AMPA at the same

pH (Figure S2). The contribution of dielectric exclusion is not clear. In an attempt to discern a potential role of hydration, the driving force was varied in experiments with different fluxes.”

14. L399-401: Here it is stated that more molecules could pass through the membrane pores because of shredding of the hydration layer. I wonder if this is true and without quantifying concentration polarization this statement is hard to substantiate.

Thank you very much for your comment. We have addressed the same comments from Reviewer #1. The concentration polarization was added and mentioned in Reviewer #1, Query #12, #13.

15. L423: states that ‘the different removal between GLY and AMPA cannot be explained by hydrogen bonding dielectric exclusion’. However, L 425-426 states that both Donnan exclusion and dielectric exclusion play a role. So what part can be explained by Donnan exclusion and what part by dielectric exclusion?

Thank you for your question. As we had written “This means the different removal between GLY and AMPA cannot be explained only by hydrogen bonding derived from dielectric exclusion. Instead, the retention of GLY and AMPA by NF membrane is found to be a combination of Donnan and dielectric exclusion.”, we confirm that the two mechanisms go hand-in-hand and it is hard to differentiate. We had discussed this extensively with Reviewer #1. From MD, we have reported that more water molecules are involved in the hydration shells of GLY than AMPA, this may contribute to the observation that removal of GLY is higher. However, and in pores of which radii are similar to or smaller than the Debye length (3 nm), there is always Donnan effect, and stronger repulsion (i.e. higher removal) can be expected for GLY owing to its higher charge density than AMPA.

16. L439-443: My understanding is that at lower pH the hydration shell is more structured and hence smaller. If this is indeed the analysis, then this should be described clearer.

Thank you very much for your comments. Indeed the hydration shell is more structured and there are more involving water molecules, but it is not evident that the hydration shell is smaller. We have provided further clarifications in response to Reviewer #1, Query #14.

17. L457: apart from the hydration shell also the Mw of GLY is higher than that of AMPA, which should have an effect on the retention.

Thank you for your comment. We have mentioned the effect of MW on the rejection of GLY and AMPA in Figure 2. The discussion on hydration layer characterization should take into account that this layer makes the water-pollutant complex much bulkier than the hydrodynamic size of the pollutant (derivable from the MW). Therefore, MW would not be mentioned in this section here to avoid redundancy in writing.

18. L506: In the conclusion paragraph I find it confusing to read that the flux was expected to play an important role in the transport ... The conclusion paragraph should be conclusive what the role of the flux is, in other words plays the flux an important role or not.

The conclusion was clarified in page 21 lines 534–550, to read.

“The removal mechanism of GLY and AMPA by NF membrane includes size exclusion, Donnan exclusion, and dielectric exclusion. Size exclusion was dominant in the filtration of the membrane at low nominal MWCO (<

150 Da) with 90% GLY/AMPA removed. Donnan exclusion and dielectric exclusion are more dominant for membranes with nominal MWCO > 200 Da. By comparing the Debye length and pore radii of the membranes, it was revealed that Donnan exclusion would always occur with membrane pore sizes (200–3000 Da), where GLY/AMPA interact with the membrane wall which carries the negative charge. At low pH (i.e. pH 2), the contribution of Donnan exclusion was less likely because GLY and AMPA are neutrally charged, and partial removal by HY 50 membranes implies an increased effective size of GLY/AMPA due to a hydration layer. At higher pH, Donnan exclusion by the membrane became more important, although the extent of Donnan exclusion may vary between the membranes as the HY series induces lower electrical potential (and charge) at all pH. Both Donnan and dielectric exclusions enhanced the removal of GLY/AMPA with increasing pH. Removal of GLY and AMPA reached > 90% for NF 90 and NF 270, and 83% for HY 50 membrane. The higher flux would provide more kinetic energy to overcome the energy barrier for dehydration before the molecules enter the pores, although the higher concentrations in the permeate can also be attributed to stronger concentration polarization effect at higher fluxes (i.e. higher concentration gradient across the membrane which drives diffusion).”

Some smaller comments

L 107: remove ‘and’

L141: remove ‘the’

L148: remove ‘then’

L 183: replace difference with ‘different’.

L260: remove ‘to the’ and add ‘and’

L 292: add ‘and’

L344: Should read ‘sulphonated’

L 376: replace ‘differing’ with ‘different’

Thank you very much for pointing out these errors. We have fixed these mistakes. Thank you very much.

** See Nature Portfolio’s author and referees’ website at www.nature.com/authors for information about policies, services and author benefits.

This email has been sent through the Springer Nature Tracking System NY-610A-NPG&MTS

Confidentiality Statement:

This e-mail is confidential and subject to copyright. Any unauthorised use or disclosure of its contents is

prohibited. If you have received this email in error please notify our Manuscript Tracking System Helpdesk team at <http://platformsupport.nature.com>.

Details of the confidentiality and pre-publicity policy may be found here <http://www.nature.com/authors/policies/confidentiality.html>

Privacy Policy | Update Profile

- [1] L.A. Richards, A.I. Schäfer, B.S. Richards, B. Corry, The importance of dehydration in determining ion transport in narrow pores, *Small*, 8 (2012) 1701-1709.
- [2] H. Saitúa, F. Giannini, A.P. Padilla, Drinking water obtaining by nanofiltration from waters contaminated with glyphosate formulations: Process evaluation by means of toxicity tests and studies on operating parameters, *Journal of Hazardous Materials*, 227-228 (2012) 204-210.
- [3] J. Yuan, J. Duan, C.P. Saint, D. Mulcahy, Removal of glyphosate and aminomethylphosphonic acid from synthetic water by nanofiltration, *Environmental Technology*, 39 (2018) 1384-1392.
- [4] A. Loi-Brügger, S. Panglisch, G. Hoffmann, P. Buchta, R. Gimbel, C.J. Nacke, Removal of trace organic substances from river bank filtrate – performance study of RO and NF membranes, *Water Supply*, 8 (2008) 85-92.
- [5] P. Xu, J.E. Drewes, C. Bellona, G. Amy, T.U. Kim, M. Adam, T. Heberer, Rejection of emerging organic micropollutants in nanofiltration-reverse osmosis membrane applications, *Water environment research : a research publication of the Water Environment Federation*, 77 (2005) 40-48.
- [6] K. Kimura, S. Toshima, G. Amy, Y. Watanabe, Rejection of neutral endocrine disrupting compounds (EDCs) and pharmaceutical active compounds (PhACs) by RO membranes, *Journal of Membrane Science*, 245 (2004) 71-78.
- [7] A.H.C. Van Bruggen, M.M. He, K. Shin, V. Mai, K.C. Jeong, M.R. Finckh, J.G. Morris, Environmental and health effects of the herbicide glyphosate, *Science of The Total Environment*, 616-617 (2018) 255-268.
- [8] S.-Y. Pan, A.Z. Haddad, A. Kumar, S.-W. Wang, Brackish water desalination using reverse osmosis and capacitive deionization at the water-energy nexus, *Water Research*, 183 (2020) 116064.
- [9] B. Cyna, G. Chagneau, G. Bablon, N. Tanghe, Two years of nanofiltration at the Méry-sur-Oise plant, France, *Desalination*, 147 (2002) 69-75.
- [10] K. Majamaa, J. Warczok, M. Lehtinen, Recent operational experiences of FILMTEC™ NF270 membrane in Europe, *Water Science and Technology*, 64 (2011) 228-232.
- [11] R.-y. Fu, T. Zhang, X.-m. Wang, Rigorous determination of pore size non-uniformity for nanofiltration membranes by incorporating the effects on mass transport, *Desalination*, 549 (2023) 116318.
- [12] V. Freger, Nanoscale heterogeneity of polyamide membranes formed by interfacial polymerization, *Langmuir : the ACS journal of surfaces and colloids*, 19 (2003) 4791-4797.
- [13] X. Song, B. Gan, S. Qi, H. Guo, C.Y. Tang, Y. Zhou, C. Gao, Intrinsic nanoscale structure of thin film composite polyamide membranes: Connectivity, defects, and structure–property correlation, *Environmental Science & Technology*, 54 (2020) 3559-3569.
- [14] L.D. Nghiem, A.I. Schäfer, M. Elimelech, Removal of natural hormones by nanofiltration membranes: Measurement, modeling, and mechanisms, *Environmental Science & Technology*, 38 (2004) 1888-1896.
- [15] A.J.C. Semião, A.I. Schäfer, Removal of adsorbing estrogenic micropollutants by nanofiltration membranes. Part A—Experimental evidence, *Journal of Membrane Science*, 431 (2013) 244-256.
- [16] A. Imbrogno, A.I. Schäfer, Micropollutants breakthrough curve phenomena in nanofiltration: Impact of operational parameters, *Separation and Purification Technology*, 267 (2021) 118406.

- [17] S. Castaño Osorio, P.M. Biesheuvel, E. Spruijt, J.E. Dykstra, A. van der Wal, Modeling micropollutant removal by nanofiltration and reverse osmosis membranes: Considerations and challenges, *Water Research*, 225 (2022) 119130.
- [18] N.S. Schwindt, R. Epsztein, A.P. Straub, S. Yue, M.R. Shirts, Molecular details of ion de-coordination at elevated salinity and pressure and consequences for membrane separations, *Journal of Membrane Science*, 734 (2025) 124358.
- [19] H. Ohshima, T. Kondo, Electrostatic repulsion of ion penetrable charged membranes: Role of Donnan potential, *Journal of Theoretical Biology*, 128 (1987) 187-194.
- [20] P. Aydogan Gokturk, R. Sujanani, J. Qian, Y. Wang, L.E. Katz, B.D. Freeman, E.J. Crumlin, The Donnan potential revealed, *Nature Communications*, 13 (2022) 5880.
- [21] L. Wang, J. He, M. Heiranian, H. Fan, L. Song, Y. Li, M. Elimelech, Water transport in reverse osmosis membranes is governed by pore flow, not a solution-diffusion mechanism, *Science Advances*, 9 (2023) eadf8488.
- [22] H. Fan, M. Heiranian, M. Elimelech, The solution-diffusion model for water transport in reverse osmosis: What went wrong?, *Desalination*, 580 (2024) 117575.
- [23] S.H. Kim, S.-Y. Kwak, T. Suzuki, Positron annihilation spectroscopic evidence to demonstrate the flux-enhancement mechanism in morphology-controlled thin-film-composite (TFC) membrane, *Environmental Science & Technology*, 39 (2005) 1764-1770.
- [24] T.E. Culp, B. Khara, K.P. Brickey, M. Geitner, T.J. Zimudzi, J.D. Wilbur, S.D. Jons, A. Roy, M. Paul, B. Ganapathysubramanian, A.L. Zydney, M. Kumar, E.D. Gomez, Nanoscale control of internal inhomogeneity enhances water transport in desalination membranes, *Science*, 371 (2021) 72-75.
- [25] L. Wang, T. Cao, J.E. Dykstra, S. Porada, P.M. Biesheuvel, M. Elimelech, Salt and water transport in reverse osmosis membranes: Beyond the solution-diffusion model, *Environmental Science & Technology*, 55 (2021) 16665-16675.
- [26] R.W. Baker, *Membrane technology and applications*, John Wiley & Sons, 2023.
- [27] V.H. Hegde, M.F. Doherty, T.M. Squires, A two-phase model that unifies and extends the classical models of membrane transport, *Science*, 377 (2022) 186-191.
- [28] V. Freger, G.Z. Ramon, The solution-diffusion model: "Rumors of my death have been exaggerated", *Journal of Membrane Science Letters*, 4 (2024) 100084.
- [29] H. Yan, X. Miao, J. Xu, G. Pan, Y. Zhang, Y. Shi, M. Guo, Y. Liu, The porous structure of the fully-aromatic polyamide film in reverse osmosis membranes, *Journal of Membrane Science*, 475 (2015) 504-510.
- [30] Y. Yoon, P. Westerhoff, S.A. Snyder, E.C. Wert, Nanofiltration and ultrafiltration of endocrine disrupting compounds, pharmaceuticals and personal care products, *Journal of Membrane Science*, 270 (2006) 88-100.
- [31] L.D. Nghiem, A.I. Schäfer, M. Elimelech, Pharmaceutical retention mechanisms by nanofiltration membranes, *Environmental Science & Technology*, 39 (2005) 7698-7705.
- [32] Y. Cheng, H. Ding, Y. Liu, D. He, L.E. Peng, H. Matsuyama, M. Hu, X. Li, Fabrication of polyethersulfone/sulfonated polysulfone loose nanofiltration membranes for enhanced selectivity of pharmaceuticals and personal care products and minerals, *Separation and Purification Technology*, 337 (2024) 126466.
- [33] E.E. Chang, C.-H. Liang, C.-P. Huang, P.-C. Chiang, A simplified method for elucidating the effect of size exclusion on nanofiltration membranes, *Separation and Purification Technology*, 85 (2012) 1-7.
- [34] M. Zocchi, R. Sommaruga, Microplastics modify the toxicity of glyphosate on *Daphnia magna*, *Science of The Total Environment*, 697 (2019) 134194.
- [35] A.I. Schäfer, I. Akanyeti, A.J.C. Semião, Micropollutant sorption to membrane polymers: A review of mechanisms for estrogens, *Advances in Colloid and Interface Science*, 164 (2011) 100-117.
- [36] K. Boussu, C. Vandecasteele, B. Van der Bruggen, Relation between membrane characteristics and performance in nanofiltration, *Journal of Membrane Science*, 310 (2008) 51-65.
- [37] A.E. Yaroshchuk, Non-steric mechanisms of nanofiltration: superposition of Donnan and dielectric exclusion, *Separation and Purification Technology*, 22-23 (2001) 143-158.

- [38] R. Epsztein, E. Shaulsky, N. Dizge, D.M. Warsinger, M. Elimelech, Role of ionic charge density in Donnan exclusion of monovalent anions by nanofiltration, *Environmental Science & Technology*, 52 (2018) 4108-4116.
- [39] V. Freger, Dielectric exclusion, an éminence grise, *Advances in Colloid and Interface Science*, 319 (2023) 102972.
- [40] K. Kimura, G. Amy, J. Drewes, Y. Watanabe, Adsorption of hydrophobic compounds onto NF/RO membranes: an artifact leading to overestimation of rejection, *Journal of Membrane Science*, 221 (2003) 89-101.
- [41] B. Van der Bruggen, J. Schaep, D. Wilms, C. Vandecasteele, Influence of molecular size, polarity and charge on the retention of organic molecules by nanofiltration, *Journal of Membrane Science*, 156 (1999) 29-41.
- [42] G. Hurwitz, G.R. Guillen, E.M.V. Hoek, Probing polyamide membrane surface charge, zeta potential, wettability, and hydrophilicity with contact angle measurements, *Journal of Membrane Science*, 349 (2010) 349-357.
- [43] L.D. Nghiem, A.I. Schäfer, M. Elimelech, Role of electrostatic interactions in the retention of pharmaceutically active contaminants by a loose nanofiltration membrane, *Journal of Membrane Science*, 286 (2006) 52-59.
- [44] W.R. Bowen, J.S. Welfoot, Modelling the performance of membrane nanofiltration—critical assessment and model development, *Chemical Engineering Science*, 57 (2002) 1121-1137.
- [45] R.S. Roth, L. Birnhack, M. Avidar, E.A. Hjelvik, A.P. Straub, R. Epsztein, Effect of solution ions on the charge and performance of nanofiltration membranes, *npj Clean Water*, 7 (2024) 25.
- [46] D.L. Oatley, L. Llenas, R. Pérez, P.M. Williams, X. Martínez-Lladó, M. Rovira, Review of the dielectric properties of nanofiltration membranes and verification of the single oriented layer approximation, *Advances in Colloid and Interface Science*, 173 (2012) 1-11.
- [47] S. You, J. Lu, C.Y. Tang, X. Wang, Rejection of heavy metals in acidic wastewater by a novel thin-film inorganic forward osmosis membrane, *Chemical Engineering Journal*, 320 (2017) 532-538.
- [48] C.P. Koutsou, A.J. Karabelas, Shear stresses and mass transfer at the base of a stirred filtration cell and corresponding conditions in narrow channels with spacers, *Journal of Membrane Science*, 399-400 (2012) 60-72.
- [49] B. Tansel, J. Sager, T. Rector, J. Garland, R.F. Strayer, L. Levine, M. Roberts, M. Hummerick, J. Bauer, Significance of hydrated radius and hydration shells on ionic permeability during nanofiltration in dead end and cross flow modes, *Separation and Purification Technology*, 51 (2006) 40-47.
- [50] A. Imbrogno, A.I. Schäfer, Comparative study of nanofiltration membrane characterization devices of different dimension and configuration (cross flow and dead end), *Journal of Membrane Science*, 585 (2019) 67-80.
- [51] E. Worch, Eine neue Gleichung zur Berechnung von Diffusionskoeffizienten gelöster Stoffe, *Vom Wasser*, 81 (1993) 289-297.
- [52] Y.-H. Cai, A.I. Schäfer, Renewable energy powered membrane technology: Impact of solar irradiance fluctuation on direct osmotic backwash, *Journal of Membrane Science*, 598 (2020) 117666.
- [53] M. Andreev, J.J. de Pablo, A. Chremos, J.F. Douglas, Influence of ion solvation on the properties of electrolyte solutions, *The Journal of Physical Chemistry B*, 122 (2018) 4029-4034.
- [54] V. Wieser, L.L.E. Mears, R.D. Barker, H.-W. Cheng, M. Valtiner, Hydration forces dominate surface charge dependent lipid bilayer interactions under physiological conditions, *The Journal of Physical Chemistry Letters*, 12 (2021) 9248-9252.
- [55] V. Pavluchkov, I. Shefer, O. Peer-Haim, J. Blotevogel, R. Epsztein, Indications of ion dehydration in diffusion-only and pressure-driven nanofiltration, *Journal of Membrane Science*, 648 (2022) 120358.
- [56] R. Epsztein, R.M. DuChanois, C.L. Ritt, A. Noy, M. Elimelech, Towards single-species selectivity of membranes with subnanometre pores, *Nature Nanotechnology*, 15 (2020) 426-436.
- [57] C. Song, B. Corry, Intrinsic ion selectivity of narrow hydrophobic pores, *The Journal of Physical Chemistry B*, 113 (2009) 7642-7649.
- [58] H. Liu, C.J. Jameson, S. Murad, Molecular dynamics simulation of ion selectivity process in nanopores, *Molecular Simulation*, 34 (2008) 169-175.

- [59] L.A. Richards, B.S. Richards, B. Corry, A.I. Schäfer, Experimental energy barriers to anions transporting through nanofiltration membranes, *Environmental Science & Technology*, 47 (2013) 1968-1976.
- [60] Y.-A. Boussouga, T. Okkali, T. Luxbacher, A.I. Schäfer, Chromium (III) and chromium (VI) removal and organic matter interaction with nanofiltration, *Science of The Total Environment*, 885 (2023) 163695.
- [61] E. Idil Mouhoumed, A. Szymczyk, A. Schäfer, L. Paugam, Y.H. La, Physico-chemical characterization of polyamide NF/RO membranes: Insight from streaming current measurements, *Journal of Membrane Science*, 461 (2014) 130-138.
- [62] Y. Shi, R. Zhang, Z. Zhu, X. Wu, J. Tian, J. Zhu, C. Tang, Molecularly welded "cellulose-like" membrane with extensive H-bond networks enabling robust and tunable nanofiltration, *Advanced Functional Materials*, n/a (2025) e20590.
- [63] A.K. Singh, S. Prakash, V. Kulshrestha, V.K. Shahi, Cross-linked hybrid nanofiltration membrane with antibiofouling properties and self-assembled layered morphology, *ACS Applied Materials & Interfaces*, 4 (2012) 1683-1692.
- [64] K. Boussu, Y. Zhang, J. Cocquyt, P. Van der Meeren, A. Volodin, C. Van Haesendonck, J.A. Martens, B. Van der Bruggen, Characterization of polymeric nanofiltration membranes for systematic analysis of membrane performance, *Journal of Membrane Science*, 278 (2006) 418-427.
- [65] A. Imbrogno, J.I. Calvo, M. Breida, R. Schwaiger, A.I. Schäfer, Molecular weight cut off (MWCO) determination in ultra- and nanofiltration: Review of methods and implications on organic matter removal, *Separation and Purification Technology*, 354 (2025) 128612.
- [66] F. Maggi, D. la Cecilia, F.H.M. Tang, A. McBratney, The global environmental hazard of glyphosate use, *Science of The Total Environment*, 717 (2020) 137167.
- [67] F. Maggi, F.H.M. Tang, F.N. Tubiello, Agricultural pesticide land budget and river discharge to oceans, *Nature*, 620 (2023) 1013-1017.
- [68] Agropages, 2025, Glyphosate prices rise, boosting industry leaders' performance, accessed on September 2 2025, https://news.agropages.com/News/NewsDetail---54367.htm?utm_source=chatgpt.com.
- [69] T. Luxbacher, *The ZETA guide: Principles of the streaming potential technique*, Anton Paar GmbH: Graz, Austria, (2014).
- [70] Y.-A. Boussouga, H. Than, A.I. Schäfer, Selenium species removal by nanofiltration: Determination of retention mechanisms, *Science of The Total Environment*, 829 (2022) 154287.
- [71] Z.W. Windom, M. Datta, M.M. Huda, M.A. Sabuj, N. Rai, Understanding speciation and solvation of glyphosate from first principles simulations, *Journal of Molecular Liquids*, 365 (2022) 120154.
- [72] J.P.A. de Jesus, F.d.A. La Porta, Elucidating the spectroscopic and physicochemical properties for a pH-dependent glyphosate structure from a computational perspective, *Computational and Theoretical Chemistry*, 1226 (2023) 114209.
- [73] O. Fliss, K. Essalah, A. Ben Fredj, Stabilization of glyphosate zwitterions and conformational/tautomerism mechanism in aqueous solution: insights from ab initio and density functional theory-continuum model calculations, *Physical Chemistry Chemical Physics*, 23 (2021) 26306-26323.
- [74] A.M. Stingel, C. Calabrese, P.B. Petersen, Strong intermolecular vibrational coupling through cyclic hydrogen-bonded structures revealed by ultrafast continuum MID-IR spectroscopy, *The Journal of Physical Chemistry B*, 117 (2013) 15714-15719.
- [75] B.L. Van Hoozen, Jr., P.B. Petersen, Vibrational tug-of-war: The pKa dependence of the broad vibrational features of strongly hydrogen-bonded carboxylic acids, *The Journal of chemical physics*, 148 (2018) 134309.
- [76] V.B. Kartha, R. Norman Jones, R.E. Robertson, Hydrogen bonding and solvolysis of alkyl sulphonate esters and related compounds, *Proceedings of the Indian Academy of Sciences - Section A*, 58 (1963) 216-228.

REVIEWER COMMENTS

Reviewer #1 (Remarks to the Author):

I have carefully read the authors' response and appreciate their efforts to address my comments. However, the essence of my original criticism remains. While the study is interesting and relevant, with some novel aspects, I do not find it sufficiently strong in scientific merit, conceptual depth, or overall impact to meet the standards of Nature Communications.

Although the authors provided detailed replies, the actual revisions to the manuscript are relatively minor and do not meaningfully enhance its scientific strength or clarity. Therefore, even with improved writing (which is still needed), the manuscript would still not reach the level of originality and significance expected for this journal.

Thank you for your assessment. While some reviewers were satisfied with revisions, this reviewer vis clearly against publication. We have addressed the specific comments and trust that revisions are able to convince. In our view the methods and the science are novel and warrant publication in Nature Communications.

Below are specific concerns regarding the authors' responses (numbers correspond to those in their rebuttal letter):

- General response: The fact that hydration of micropollutants has not been widely studied (which is debatable) does not, by itself, make the work sufficiently novel or significant for Nature Communications.

If there is any work on hydration of micropollutants then we would like to know about this. Please provide the relevant references as our thorough searches did not yield such work. Hydration is well studied for ionic contaminants. The quantification of the hydration of organic contaminants requires suitable tools, according to our knowledge and this is the very challenging task that we have tackled through a very interdisciplinary collaboration.

- Response 1: The title remains overly general and non-informative, resembling that of a review article rather than an original research paper. It does not convey the specific insight or contribution of the study. While this is partly stylistic, I still recommend making it more focused and descriptive.

Thank you for your comment, but we do not really understand the concern of the reviewer. The title of this manuscript is "*Removal of glyphosate and aminomethylphosphonic acid (AMPA) by nanofiltration membranes: The role of hydration*", which encompasses a specific technology (Nanofiltration), the work concerns the removal of two specific micropollutants (*glyphosate and aminomethylphosphonic acid (AMPA)*) and the novelty lies in the attempt to discern possible hydration effects from steric, Donnan and dielectric exclusions. The challenge is the estimation of the strength of hydration from spectroscopic measurements and MD modelling. In our opinion, the title is apt for the original research that we carried out and does not give any impression of a review. It is indeed IAMT policy to review literature very carefully, which is – in our view of particular importance when tools are used that are rarely within the expertise of all readers. In this case the spectroscopy and the MD simulations were new to us and membrane water process engineers – it is this a service to the readers from various fields to provide the necessary background information.

In order to accommodate the concerns of the reviewer, we have decided to change the title to "The role of hydration in the removal of glyphosate (GLY) and aminomethylphosphonic acid (AMPA) by nanofiltration membranes".

- Response 2: The Nature Communications guidelines explicitly prohibit subheadings in the Introduction and emphasize brevity. The current Introduction is long, subdivided, and includes excessive background (e.g., two full paragraphs on contaminant exposure and abundance). Such information could be summarized in one or two sentences. Similar verbosity appears throughout the manuscript, resulting in a presentation more suited to a thesis than to a concise, high-impact research paper.

Thank you for your emphasis that the Introduction is too long. This time we have made the following changes to shorten the texts. The word count has decreased from 2420 to 1520 words, and the sub-headings have been removed. While we believe that this shortening is not improving the manuscript, but rather removes valuable information (see above comment in this regard), we have carried out this request to satisfy the reviewer.

The section related to GLY/AMPA occurrence in the water environment is significantly shortened. It now reads:

“Pollution of water resources by pesticides and herbicides is an emerging concern worldwide [1-3] and reportedly cause an annual economic loss of 78 million euros in Europe [4]. Organophosphate herbicides are widely used in agriculture [5-7], despite their well-established toxicity to aquatic organisms and human health [8]. Exposure to elevated concentrations of glyphosate (N-(phosphonomethyl) glycine) (GLY) and aminomethylphosphonic acid (AMPA) are linked to increased risks of kidney, dermatological, and respiratory issues, mental disorders, miscarriages, and cancer [9-14]. In some countries, these pesticides are not regulated and sold in commodity shops (*e.g.* Roundup) without any regulations [15]. GLY accounts for 30% of total herbicide use worldwide [16-18] and its market is still expanding, projected to reach 17 billion Euro by 2031 [7]. In 2023, the European Commission authorized the GLY market for 10 more years until 2033 [19]. GLY can be broken down by microbial organisms into shorter and potentially harmful degradation products [20, 21]. AMPA is formed from GLY *via* a N–C bond breaking [20] and exhibits similar toxicity to humans [22, 23], although AMPA is more persistent in soil than GLY [24]. Another source of AMPA are antiscalants for RO membranes, contributing to AMPA leaching from RO that can be detected in the water treatment plant effluents [25]. In ground- and surface water, GLY and AMPA generally occur at concentrations between 0.2 and 370 µg/L [7, 26-28], although GLY has also been reported in milligrams-per-liter concentrations [29]. In Germany, GLY and AMPA concentrations were detected at around 1 µg/L [30, 31].

In 2022, the European Commission proposes to regulate the herbicide concentrations in groundwater and inland surface water to be below 100 ng/L for each type of herbicide, and 500 ng/L for total herbicides [32]. The same thresholds are regulated for drinking water [33]. Balancing economic benefit and water environmental protection is the key to sustainable development [34], which emphasizes the importance of advanced water treatment technologies to remove GLY and its metabolites (such as AMPA) from water sources [35, 36].”

The section on exclusion mechanisms is shortened by 30%, with the textbook knowledge on the zeta potential shortened. It now reads:

“The incomplete removal of GLY/AMPA can be explained by the structural imperfections of NF/RO membranes, which contains sub-nanometer pores (or sometimes referred to as regions of lower polymer densities) forming transport channels of water and certain solutes [37, 38]. Micropollutants such as GLY/AMPA are excluded *via* i) size, ii) Donnan, and iii) dielectric exclusions [39] (see Figure 1). Size exclusion is based on the relative sizes of the membrane pores and target molecules [40-42]; removal by size exclusion thus strongly depends on NF/RO membrane pores [43], which are non-uniform in sizes [44, 45], and the molecular characteristics such as shape and orientation [46]. When micropollutants penetrate the pores, adsorption on membrane surface materials (such as polyamide) occurs and subsequently, the micropollutants are transported through the membrane by a combination of desorption, diffusion, and convection in the pores [39, 47] (Figure 1D). Charged and hydrophilic micropollutants such as GLY/AMPA have a weaker interaction with polyamide than uncharged and hydrophobic micropollutants [48]; hence, adsorption of these to membrane material is low [49].

The weak attraction of charged micropollutants to the membrane polymer is also related to two other exclusion mechanisms: Donnan and dielectric exclusions [50-52]. Charge (Donnan) exclusion indicates the retention due to electrostatic repulsion between the (negatively) charged ions or micropollutants and the

charged membrane surface [53]. The charge at the membrane surface (or the pore section close to the membrane surface) is characterized by the zeta potential (ζ) [54] and the double layer thickness at the membrane surface (the Debye length, κ^{-1}) [55, 56]. The relative scale of pore radius and the Debye length controls the transport of ions and charged species that are smaller than membrane pores [55, 56]. The surface charge (or zeta potential) will determine the electrostatic interaction ‘strength’, which will decrease significantly (by around 2.8 times, or 1 e at the Debye length distance [55-57]). If the Debye length is greater than the pore radius, charged species at the pore entrance likely repel the charged membrane surface or pore walls and encounter resistance depending on the (pore) surface electrostatic potential [58], increasing the likelihood of removal (Figure S1). Naturally, pores are tortuous, which complicates the application of this conceptual model. The strength of electrostatic interactions depends on the charges of the membrane surface and GLY/AMPA [59]. Both micropollutants are polar and hydrophilic with amine, phosphate, and carboxylate groups [60, 61], which makes the charge strongly pH-dependent (Figure S2).”

The discussion about dielectric exclusion has also been shortened, which now reads:

“Dielectric exclusion (*i.e.*, retention of the hydrated solutes) depends on the combined size of the molecule (or ion) plus the hydration layer being larger than the pore [52, 62], and the resistance of ions/micropollutants entering the pores due to an energy barrier associated with dehydration [63, 64]. Dielectric exclusion is directly independent of the charges of micropollutants and membrane (although the hydration properties may be influenced by the charges) and always occurs [65]. Many studies have looked into dielectric exclusion for ion transport through NF/RO pores [51, 66], but research on this mechanism for charged organic compounds (that are larger than ions but may possess multiple charged sites) is sparse. Molecular dynamics (MD) [67], density functional theory (DFT) simulations [68], and quantum mechanics (*ab initio*) calculations [69] suggest that zwitterionic GLY adopts a linearized structure in water and dominantly forms strong hydrogen bonds with water at the phosphate and carboxylate ends. Larger hydration shell sizes can be induced with the increase of molecular charge corresponding to the change in phosphate and carboxylate group charges [67]. Ions or molecules with high charge density mean more energy is required to shed the hydration layer [70, 71]. Such shredding is possible if the kinetic energy (for example, by applied pressure in pressure-driven membrane filtration) surpasses the activation energy of partial hydration, allowing charged ions and molecules to enter the NF/RO pores [72, 73]. This dehydration effect has been reported for solvated ions (and potentially charged molecules) from computational [74, 75] and experimental studies [72, 76].”

The description of spectroscopic analysis has also been shortened to avoid duplicate contents. This reads:

“It is possible to characterize the strength of the hydration layer of charged pollutants. Infrared solvation shell spectroscopy [77] and Raman spectroscopy, paired with multivariate-curve-resolution [78, 79] (IR-SSS and Raman-MCR) can extract the vibrational spectrum of a probe molecule with its solvation shell from the total spectrum. The total spectrum of a solution is considered to contain two contributions: the spectrum of (i) the bulk background solvent and that of (ii) the solute molecule with its solvation shell, *i.e.*, the part of the solvent that is perturbed by the solute due to the interactions between the solute and solvent [77, 80]. The spectra of the probe molecule solution and that of the pure background solvent are measured individually [81], and their difference (following a mathematical MCR routine [82]) results in the solute-correlated spectrum [83, 84]. The obtained solute-correlated spectrum is analyzed for the spectral shifts of the OH stretch in the solvation shell relative to that of the bulk. The OH stretch vibration is very sensitive to the local hydrogen-bonded network and thus an excellent probe of the changes in the hydrogen-bonding strengths between the solute and solvent compared to that of the solvent-solvent interactions [85]. Here, stronger hydrogen bonding weakens the OH bond, resulting in a red-shift of the OH stretch vibration. The OH stretch spectrum thus provides a map of the hydrogen-bond strengths within the sample [77].”

While the new sections have been highlighted, the removal of deleted sections will probably not show the major revision that the reviewer expects. Please take this fact in to consideration.

- Response 8: Although dead-end filtration is experimentally simpler, it is less representative of real membrane systems and more prone to concentration polarization-an important factor in transport studies. The claim that dead-end configuration promotes ion dehydration more than crossflow remains weak and insufficiently supported, even if previously mentioned in ref. 49.

Thank you very much again for your insightful comment. The choice of filtration system (IAMT has a vast number of both stirred cell and crossflow systems) depends on the project and in this case a stirred cell was deemed most suitable. The mass transfer of all systems has been characterized at IAMT and indeed a stirred cell is not so dissimilar to a spiral wound module as in such a crossflow system the concentration increases significantly over the length of a module, which is not what a laboratory crossflow module – with an inherently small membrane area – can adequately represent. The comment that this is less representative of a real membrane is this incorrect.

Inevitably each choice brings pros and cons and this is carefully evaluated when making choices in environmental design. Indeed, when designing the experiments with the stirred cell system, the purpose was to elevate the shear force and in the process the effect of concentration polarization was mistakenly overlooked. This was corrected in the first revision. As the shear force and concentration polarization are coupled together, the decrease in concentration had been attributed to both the shredding of the hydration layer and the concentration effect. In the last round of revisions, we had done additional calculations for the NF membranes to i) determine the mass transfer coefficients at each flux, ii) estimate the concentration on the membrane surface, iii) determine the “real” removal, and finally iv) attribute whether the trend in “real” removal can be solely attributed to CP (“real” removal is uniform) or attributed to a combination of CP and hydration layer shredding. While our results for NF270 and NF90 implies strong effect of CP (decreased observed removal but uniform real removal) at different fluxes, our results for HY50 membrane imply that the effect of CP cannot alone explain the trend in real removal.

To avoid misinterpretation, we have removed the sentence claiming that dead-end was selected to enhance the hydration layer shedding in page 21 lines 507–509, as this depends on experimental design and operating parameters, rather than the system configuration alone.

- Response 12: I do not dispute the occurrence of dehydration; rather, I question whether the pressures typical of RO/NF systems (5–70 bar) are sufficient to induce it. The cited studies (refs. 55 and 59) assert this effect but do not demonstrate it quantitatively. The authors were asked to perform simple estimations to evaluate whether such pressures could realistically compress hydration shells, but this was not done. The distinction they draw between pressure effects in bulk solution and within the membrane also remains unclear.

Thank you very much for your comment. We have looked at the information closely and performed some estimations. Firstly, we would like to clarify that in this round, the reviewer comments that we need to “evaluate whether such pressures could realistically compress hydration shells”. This comment is clearer than the original comment in revision 1, in that now we know that the review expects the hydration shell is compressed for GLY/AMPA to enter the pore.

In the bulk solution, the hydration layer is complete, as can be seen in the following figure:

Figure 1. Hydration shell of charge-neutral (a) AMPA, and (b) GLY in the bulk water solution (snapshot from MD simulations, hydrogen bonds are indicated by the dashed red lines)

To estimate the compression, we can regard the solvated species as the spherical particles of approximate radius 0.5 nm. Application of 70 bar pressure would shrink the hydrogen bonds between the species and the water molecules by $\sim 0.01 \text{ \AA}$, using the typical hydrogen-bond force constant value of 1 eV/\AA^2 (or $96 \text{ kJ/mol per \AA}^2$). Indeed, this compression is negligible in the bulk solution.

However, when the species enters the membrane pores, it interacts with polar groups that naturally replace some of the hydration-shell water molecules. This is what dehydration means: some water molecules move away from the species' proximity shell and towards the 'bulk' (*i.e.* far away from the species). This process is spontaneous, and the species tends to stick to the pore surface and form clusters as they move through it. That is shown in the following snapshots from the molecular dynamics simulations:

Figure 2. Hydration shell of charge-neutral (a) AMPA, and (b) GLY inside the piperazine-polyamide membrane pores (snapshot from molecular dynamics simulations, pore surface around the polyamide is drawn)

In this regard, the transport of ion/species does not depend on the compression of the hydration shell, but the reordering or shuffling of the hydrogen bonding network. In the bulk phase, the energy of this rearrangement is similar to the hydrogen bond strength and in the order of 20 kJ/mol [86]. In the membrane system, this energy is higher but within the same order.

Epsztein *et al.* summarized that the 'dehydration energy' for ion permeation in NF270 and NF90 membranes derived from experiments are below 65 kJ/mol [73]. This energy can be explained as the required energy when some water molecules shred hydrogen bonds with each other in the hydration layer subtracted by the released energy when they form hydrogen bonds in the bulk phase. The pressure is therefore not required for changing the volume of the system *i.e.* compression, but to contribute to the molecular kinetics (facilitating solution flow through the membrane pores and helps the species to get into contact with the pore surface) and speed up the hydrogen bond network reordering.

- Response 13: The statement “discern concentration polarization from dehydration” is conceptually unclear—these are distinct and non-comparable phenomena. In addition, describing concentration polarization as “evaluated via concentration at the membrane surface” is incorrect: elevated surface concentration is a symptom, not the cause, of polarization.

Thank you very much for your insightful comment. As we described above, we were not trying to discern the two effects, although there is an implication that the dehydration may play a more pronounced role in the retention by the HY50 membrane compared to the NF270 and NF90 membranes.

The sentences in page 12, lines 316–318 has been revised from:

“In an attempt to discern a potential role of hydration, the driving force was varied in experiments with different fluxes.”

To now read:

“In an attempt to indicate a clearer contribution of hydration to the overall GLY/AMPA removal by NF membranes, the driving force was varied in experiments with different fluxes.”

We trust that these thorough discussions will help clarify the concerns of the reviewer and enable publication of this manuscript. Thank you for the time given to review our work.

Reviewer #2 (Remarks to the Author):

My comments were addressed.

Thank you very much.

Reviewer #3 (Remarks to the Author):

After re-evaluation, I have the following additional comments

1. The title is rather narrow focusing on GLY and AMPA, whereas the topic of the paper is more generic. However, there is a lot of attention on GLY and AMPA in the public domain, so I appreciate that authors mention these compounds explicitly in their title.

Thank you very much for your assessment (which seems opposite to Reviewer #1’s Comment #1). It is astonishing that reviewer#1 deemed the title as too unspecific. The title has been changed and the compounds were maintained in the title, indeed this is a very timely and somewhat controversial issues and finding treatment solution is an important aspect of this discussion. While the hydration effect that we address can be valid to other charged micropollutants, variability may arise from specific chemical properties and indeed we only discuss only these two types of micropollutants (GLY and AMPA), and focus on careful design of filtration experiments, spectroscopy and MD modelling for a more in-depth analysis. Hopefully we have found a compromise that satisfies all reviewers.

2. The readability of the paper has been improved compared to the earlier version. Also the introduction is clearer and somewhat more concise than the previous version, although still somewhat lengthy, e.g. L88-101 provides a lot of detail on performances of different/specific membranes, which can be summarized in a couple of sentences.

Thank you very much for your comment. Since Reviewer #1 has similar concerns, we have revised the introduction and significantly reduced the word count. Please refer to Reviewer #1's Comment #2 for the detailed changes in the main text.

3. In addition, the description of figure 1 can be more concise, e.g. the explanation on the role of zeta potential (L127-152) can be found in standard text books, and does not require such extensive explanation in this paper. L178-207, can also be more compact focusing less on the details of the methodology and more on what information can be obtained with these techniques and how this relates to the interpretation of dielectric exclusion.

Thank you, we have already revised the Introduction following yours and Reviewer #1's comments. Please see our response to Reviewer #1's Comment #2.

4. The authors have chosen a dead-end filtration set up to study the retention of GLY/AMP, because more shear force can be generated. Authors claim that the hydrodynamic conditions in the dead-end module are similar to those in cross-flow with reference to previous work that they have done. No further details are given in this paper. Nevertheless, it is well known that the conditions in a dead-end filtration system whereby the shear force is generated by a magnetic stirrer on top of the membrane are significantly different from cross flow filtration, which is used in practice. Therefore, there remains uncertainty in how far the results from one system (dead-end) can be transferred to that of another system (cross-flow).

Thank you very much for your insightful comments. A small note must be added here in that our in-house designed stirred called actually are not the 'typical' magnetic stirrer that operated to mix the solution, but rather a very well-designed stirrer that operates in the vicinity of the membrane and this creates very good mass transfer. This system has been both carefully designed and evaluated in terms of mass transfer. To ensure comparability between dead-end and cross flow configuration, Schäfer's group previously demonstrated that, with carefully selected operating parameters, the hydrodynamics of a lab-scale dead-end stirred cell can be matched to those of a crossflow system (see that are very important in mass transfer and a hindrance when explaining transport mechanisms. At IAMT normally no spacers are used.

Table 1). Experimental results for salt and organic tracer removal in these systems were consistent, supporting that dead-end configuration can be considered for this study. It is important to note that most laboratory experiments bear draw-backs and that in crossflow filtration is associated with the use (or omission) of spacers that are very important in mass transfer and a hindrance when explaining transport mechanisms. At IAMT normally no spacers are used.

Table 1. Configuration of filtration systems [87]

	Micro-crossflow system	Macro-crossflow system	Stirred cell system
Membrane area (cm ²)	2.0	47.5	38.5
Stirring speed (rpm)	-	-	100-400
Crossflow velocity (m/s)	0.1-0.6	0.1-0.8	-
Reynolds number	100-800	100-1000	1-5 · 10 ⁴
Sherwood number	30-80	15-40	400-1700
Schmidt number	600-1500	600-1500	600-1500
Mass transfer coefficients (10 ⁻⁵ m/s)*			
For NaCl	6.1	2.8	5.7
For MgSO ₄	4.0	1.9	3.8
For glucose	3.4	1.6	3.2
For xylose	3.7	1.7	3.4
For erythritol	3.9	1.8	3.7

For dioxane	4.4	2.0	4.1
-------------	-----	-----	-----

*** At stirred stirring 400 rpm and crossflow velocity in the other two systems 0.4 m/s.**

5. Authors claim that the higher shear force in a dead-end filtration system can lead to more shredding of the hydration shell and therefore less retention. However, the higher shear force will also lead to less concentration polarization and hence higher permeation of ions. I wonder how, with the current experimental setup, these two mechanisms can be decoupled. Also, Donnan and Dielectric effects cannot be fully decoupled up to an extend that one can quantify what part of the rejection is due to steric and what part due to Donnan and dielectric exclusion.

Thank you very much for your insightful comment. Decoupling different mechanisms in membrane filtration is indeed typically near impossible due to the complex interplay of so many mechanisms. This comment is similar to Reviewer #1, Comment #8. Please kindly refer to our response to that comment.

Additionally, in theory, the Donnan effect, which depends on the charges of both the pollutant and membrane surface can be negligible if the membrane surface has no charge, leaving behind only the dielectric exclusion as potential mechanism. However, an uncharged NF membrane is not a possibility because such membrane does firstly not exist, and secondly it would be highly prone to fouling. The HY membranes have some negative surface charge indicated from zeta potential measurements, but this is much weaker than the surface charge of polyamide membranes (NF270, NF90 and BW30). We can attribute that the dielectric exclusion may play a bigger role in the exclusion by the HY membranes than by the polyamide membranes, but indeed the size and charge exclusion cannot be fully decoupled as there is simply no membrane that allows an investigation with only one parameter changing. This is a field where '2D Materials' that are hailed as the next generation membranes (a promise that we disagree with) may offer the possibility of fundamental studies through carefully designed pore size and charge. To date this possibility has not been achieved and in the meantime, we are limited to what is commercially available. Nevertheless, such work offers significant progress, not in membrane properties, but in the tools of spectroscopy and MD that can be applied to answer such questions.

I believe this paper is worth publishing, despite the many remaining questions and uncertainties, as it provides new insights, although not conclusive, on the importance of Donnan and dielectric exclusion in addition to size exclusion of relatively small molecules at different (nominal) MWCO's. Also the chosen molecules (GLY/AMPA) are highly relevant, e.g. for aquatic systems toxicity as well as (drinking) water quality. However, if the paper doesn't meet the requirements of Nature Communications, then I would recommend publication in e.g. J. Membrane Science or J. Water Process Engineering.

Thank you for the positive assessment and I trust that Nature Communications will value the work. The previous papers of this project were published in Water Research, J Haz Materials, Advanced Functional Materials and it seems to us that this – more fundamental work compared to the previous – is indeed well suited to Nature Communications. It is early days, as these publications were only published in the last two years, but it appears that they are well received given the relevance of the topic, the analysis at environmentally relevant concentrations and the work with relevant commercial membranes, rather than some irrelevant hype material.

--

** See Nature Portfolio's author and referees' website at www.nature.com/authors for information about policies, services and author benefits.

This email has been sent through the Springer Nature Tracking System NY-610A-NPG&MTS

Confidentiality Statement:

This e-mail is confidential and subject to copyright. Any unauthorised use or disclosure of its contents is prohibited. If you have received this email in error please notify our Manuscript Tracking System Helpdesk team at <http://platformsupport.nature.com>.

Details of the confidentiality and pre-publicity policy may be found here

<http://www.nature.com/authors/policies/confidentiality.html>

Privacy Policy | Update Profile

References

- [1] L. Yang, X. He, S. Ru, Y. Zhang, Herbicide leakage into seawater impacts primary productivity and zooplankton globally, *Nature Communications*, 15 (2024) 1783.
- [2] F. Maggi, F.H.M. Tang, F.N. Tubiello, Agricultural pesticide land budget and river discharge to oceans, *Nature*, 620 (2023) 1013-1017.
- [3] S. Stehle, S. Bub, R. Schulz, Compilation and analysis of global surface water concentrations for individual insecticide compounds, *Science of The Total Environment*, 639 (2018) 516-525.
- [4] P. Fantke, R. Friedrich, O. Jolliet, Health impact and damage cost assessment of pesticides in Europe, *Environment International*, 49 (2012) 9-17.
- [5] A.C. Schomberg, S. Bringezu, A.W.H. Beusen, Water quality footprint of agricultural emissions of nitrogen, phosphorus and glyphosate associated with German bioeconomy, *Communications Earth & Environment*, 4 (2023) 404.
- [6] A. Sharma, A. Shukla, K. Attri, M. Kumar, P. Kumar, A. Suttee, G. Singh, R.P. Barnwal, N. Singla, Global trends in pesticides: A looming threat and viable alternatives, *Ecotoxicology and Environmental Safety*, 201 (2020) 110812.
- [7] J.P. Muñoz, E. Silva-Pavez, D. Carrillo-Beltrán, G.M. Calaf, Occurrence and exposure assessment of glyphosate in the environment and its impact on human beings, *Environmental Research*, 231 (2023) 116201.
- [8] H. Mali, C. Shah, B.H. Raghunandan, A.S. Prajapati, D.H. Patel, U. Trivedi, R.B. Subramanian, Organophosphate pesticides an emerging environmental contaminant: Pollution, toxicity, bioremediation progress, and remaining challenges, *Journal of Environmental Sciences*, 127 (2023) 234-250.
- [9] European Environmental Agency, How pesticides impact human health and ecosystems in Europe, in: European commission (Ed.), 2023.
- [10] V.J. Koller, M. Fürhacker, A. Nersesyan, M. Mišák, M. Eisenbauer, S. Knasmueller, Cytotoxic and DNA-damaging properties of glyphosate and Roundup in human-derived buccal epithelial cells, *Archives of toxicology*, 86 (2012) 805-813.
- [11] K.Z. Guyton, D. Loomis, Y. Grosse, F. El Ghissassi, L. Benbrahim-Tallaa, N. Guha, C. Scocianti, H. Mattock, K. Straif, Carcinogenicity of tetrachlorvinphos, parathion, malathion, diazinon, and glyphosate, *The Lancet Oncology*, 16 (2015) 490-491.
- [12] J.P. Myers, M.N. Antoniou, B. Blumberg, L. Carroll, T. Colborn, L.G. Everett, M. Hansen, P.J. Landrigan, B.P. Lanphear, R. Mesnage, L.N. Vandenberg, F.S. vom Saal, W.V. Welshons, C.M. Benbrook, Concerns over use of glyphosate-based herbicides and risks associated with exposures: a consensus statement, *Environmental Health*, 15 (2016) 19.
- [13] C. Nerozzi, S. Recuero, G. Galeati, D. Bucci, M. Spinaci, M. Yeste, Effects of Roundup and its main component, glyphosate, upon mammalian sperm function and survival, *Scientific Reports*, 10 (2020) 11026.
- [14] N. Lemke, A. Murawski, M.I.H. Schmied-Tobies, E. Rucic, H.-W. Hoppe, A. Conrad, M. Kolossa-Gehring, Glyphosate and aminomethylphosphonic acid (AMPA) in urine of children and adolescents in Germany –

Human biomonitoring results of the German Environmental Survey 2014–2017 (GerES V), *Environment International*, 156 (2021) 106769.

[15] S. Haggblade, A. Diarra, A. Traoré, Regulating agricultural intensification: Lessons from West Africa's rapidly growing pesticide markets, *Development Policy Review*, 40 (2022) e12545.

[16] C. Gillezeau, M. van Gerwen, R.M. Shaffer, I. Rana, L. Zhang, L. Sheppard, E. Taioli, The evidence of human exposure to glyphosate: A review, *Environmental health*, 18 (2019) 2-2.

[17] C.M. Benbrook, Trends in glyphosate herbicide use in the United States and globally, *Environmental sciences Europe*, 28 (2016) 3-3.

[18] F. Maggi, D. la Cecilia, F.H.M. Tang, A. McBratney, The global environmental hazard of glyphosate use, *Science of The Total Environment*, 717 (2020) 137167.

[19] B. Casassus, EU allows use of controversial weedkiller glyphosate for 10 more years, *Nature*, (2023).

[20] A.V. Sviridov, T.V. Shushkova, I.T. Ermakova, E.V. Ivanova, D.O. Epiktetov, A.A. Leontievsky, Microbial degradation of glyphosate herbicides (Review), *Applied Biochemistry and Microbiology*, 51 (2015) 188-195.

[21] B. Hove-Jensen, D.L. Zechel, B. Jochimsen, Utilization of glyphosate as phosphate source: biochemistry and genetics of bacterial carbon-phosphorus lyase, *Microbiology and Molecular Biology Review*, 78 (2014) 176-197.

[22] K. Zouaoui, S. Dulaurent, J.M. Gaulier, C. Moesch, G. Lachâtre, Determination of glyphosate and AMPA in blood and urine from humans: About 13 cases of acute intoxication, *Forensic Science International*, 226 (2013) e20-e25.

[23] H.M. Schluter, H. Bariami, H.L. Park, Potential role of glyphosate, glyphosate-based herbicides, and AMPA in breast cancer development: A review of human and human cell-based studies, *Int J Environ Res Public Health*, 21 (2024).

[24] C.P.M. Bento, X. Yang, G. Gort, S. Xue, R. van Dam, P. Zomer, H.G.J. Mol, C.J. Ritsema, V. Geissen, Persistence of glyphosate and aminomethylphosphonic acid in loess soil under different combinations of temperature, soil moisture and light/darkness, *Science of The Total Environment*, 572 (2016) 301-311.

[25] D. Armbruster, U. Müller, O. Happel, Characterization of phosphonate-based antiscalants used in drinking water treatment plants by anion-exchange chromatography coupled to electrospray ionization time-of-flight mass spectrometry and inductively coupled plasma mass spectrometry, *Journal of Chromatography A*, 1601 (2019) 189-204.

[26] N. Suciu, E. Russo, M. Calliera, G.P. Luciani, M. Trevisan, E. Capri, Glyphosate, glufosinate ammonium, and AMPA occurrences and sources in groundwater of hilly vineyards, *Science of The Total Environment*, 866 (2023) 161171.

[27] M. Feltracco, E. Barbaro, E. Morabito, R. Zangrando, R. Piazza, C. Barbante, A. Gambaro, Assessing glyphosate in water, marine particulate matter, and sediments in the Lagoon of Venice, *Environmental Science and Pollution Research*, 29 (2022) 16383-16391.

[28] I. Navarro, A. de la Torre, P. Sanz, N. Abrantes, I. Campos, A. Alaoui, F. Christ, F. Alcon, J. Contreras, M. Glavan, I. Pasković, M.P. Pasković, T. Nørgaard, D. Mandrioli, D. Sgargi, J. Hofman, V. Aparicio, I. Baldi, M. Bureau, A. Vested, P. Harkes, E. Huerta-Lwanga, H. Mol, V. Geissen, V. Silva, M.Á. Martínez, Assessing pesticide residues occurrence and risks in water systems: A Pan-European and Argentina perspective, *Water Research*, 254 (2024) 121419.

[29] I.B. Lima, I.G. Boëchat, M.D. Fernandes, J.A.F. Monteiro, L. Rivaroli, B. Gücker, Glyphosate pollution of surface runoff, stream water, and drinking water resources in Southeast Brazil, *Environmental Science and Pollution Research*, 30 (2023) 27030-27040.

[30] N. Tauchnitz, F. Kurzius, H. Rupp, G. Schmidt, B. Hauser, M. Schrödter, R. Meissner, Assessment of pesticide inputs into surface waters by agricultural and urban sources - A case study in the Querne/Weida catchment, central Germany, *Environmental Pollution*, 267 (2020) 115186.

[31] V. Silva, L. Montanarella, A. Jones, O. Fernández-Ugalde, H.G.J. Mol, C.J. Ritsema, V. Geissen, Distribution of glyphosate and aminomethylphosphonic acid (AMPA) in agricultural topsoils of the European Union, *Science of The Total Environment*, 621 (2018) 1352-1359.

[32] European Commission, Proposal for a Directive of the European Parliament and of the Council amending Directive 2000/60/EC establishing a framework for community action in the field of water policy, *Directive*

2006/118/EC on the protection of groundwater against pollution and deterioration and Directive 2008/105/EC on environmental quality standards in the field of water policy, in, 2022.

[33] The European Parliament and the Council of European Union, Directive (EU) 2020/2184 of the European parliament and of the council of 16 December 2020 on the quality of water intended for human consumption, in: Directive - 2020/2184, Official Journal of the European Union, 2020, pp. 37-38.

[34] United Nation, Goal 6: Clean water and sanitation, in, The Sustainable Development Goals, 2015.

[35] M. Garrido-Baserba, D.L. Sedlak, M. Molinos-Senante, I. Barnosell, O. Schraa, D. Rosso, M. Verdaguer, M. Poch, Using water and wastewater decentralization to enhance the resilience and sustainability of cities, *Nature Water*, 2 (2024) 953-974.

[36] K. Obaideen, N. Shehata, E.T. Sayed, M.A. Abdelkareem, M.S. Mahmoud, A.G. Olabi, The role of wastewater treatment in achieving sustainable development goals (SDGs) and sustainability guideline, *Energy Nexus*, 7 (2022) 100112.

[37] T.E. Culp, B. Khara, K.P. Brickey, M. Geitner, T.J. Zimudzi, J.D. Wilbur, S.D. Jons, A. Roy, M. Paul, B. Ganapathysubramanian, A.L. Zydney, M. Kumar, E.D. Gomez, Nanoscale control of internal inhomogeneity enhances water transport in desalination membranes, *Science*, 371 (2021) 72-75.

[38] H. Yan, X. Miao, J. Xu, G. Pan, Y. Zhang, Y. Shi, M. Guo, Y. Liu, The porous structure of the fully-aromatic polyamide film in reverse osmosis membranes, *Journal of Membrane Science*, 475 (2015) 504-510.

[39] S. Castaño Osorio, P.M. Biesheuvel, E. Spruijt, J.E. Dykstra, A. van der Wal, Modeling micropollutant removal by nanofiltration and reverse osmosis membranes: Considerations and challenges, *Water Research*, 225 (2022) 119130.

[40] Y. Yoon, P. Westerhoff, S.A. Snyder, E.C. Wert, Nanofiltration and ultrafiltration of endocrine disrupting compounds, pharmaceuticals and personal care products, *Journal of Membrane Science*, 270 (2006) 88-100.

[41] L.D. Nghiem, A.I. Schäfer, M. Elimelech, Pharmaceutical retention mechanisms by nanofiltration membranes, *Environmental Science & Technology*, 39 (2005) 7698-7705.

[42] Y. Cheng, H. Ding, Y. Liu, D. He, L.E. Peng, H. Matsuyama, M. Hu, X. Li, Fabrication of polyethersulfone/sulfonated polysulfone loose nanofiltration membranes for enhanced selectivity of pharmaceuticals and personal care products and minerals, *Separation and Purification Technology*, 337 (2024) 126466.

[43] R.-y. Fu, T. Zhang, X.-m. Wang, Rigorous determination of pore size non-uniformity for nanofiltration membranes by incorporating the effects on mass transport, *Desalination*, 549 (2023) 116318.

[44] V. Freger, Nanoscale heterogeneity of polyamide membranes formed by interfacial polymerization, *Langmuir : the ACS journal of surfaces and colloids*, 19 (2003) 4791-4797.

[45] X. Song, B. Gan, S. Qi, H. Guo, C.Y. Tang, Y. Zhou, C. Gao, Intrinsic nanoscale structure of thin film composite polyamide membranes: Connectivity, defects, and structure–property correlation, *Environmental Science & Technology*, 54 (2020) 3559-3569.

[46] E.E. Chang, C.-H. Liang, C.-P. Huang, P.-C. Chiang, A simplified method for elucidating the effect of size exclusion on nanofiltration membranes, *Separation and Purification Technology*, 85 (2012) 1-7.

[47] A.I. Schäfer, I. Akanyeti, A.J.C. Semião, Micropollutant sorption to membrane polymers: A review of mechanisms for estrogens, *Advances in Colloid and Interface Science*, 164 (2011) 100-117.

[48] K. Boussu, C. Vandecasteele, B. Van der Bruggen, Relation between membrane characteristics and performance in nanofiltration, *Journal of Membrane Science*, 310 (2008) 51-65.

[49] J. Yuan, J. Duan, C.P. Saint, D. Mulcahy, Removal of glyphosate and aminomethylphosphonic acid from synthetic water by nanofiltration, *Environmental Technology*, 39 (2018) 1384-1392.

[50] A.E. Yaroshchuk, Non-steric mechanisms of nanofiltration: superposition of Donnan and dielectric exclusion, *Separation and Purification Technology*, 22-23 (2001) 143-158.

[51] R. Epsztein, E. Shaulsky, N. Dizge, D.M. Warsinger, M. Elimelech, Role of ionic charge density in Donnan exclusion of monovalent anions by nanofiltration, *Environmental Science & Technology*, 52 (2018) 4108-4116.

[52] V. Freger, Dielectric exclusion, an éminence grise, *Advances in Colloid and Interface Science*, 319 (2023) 102972.

- [53] B. Van der Bruggen, J. Schaep, D. Wilms, C. Vandecasteele, Influence of molecular size, polarity and charge on the retention of organic molecules by nanofiltration, *Journal of Membrane Science*, 156 (1999) 29-41.
- [54] G. Hurwitz, G.R. Guillen, E.M.V. Hoek, Probing polyamide membrane surface charge, zeta potential, wettability, and hydrophilicity with contact angle measurements, *Journal of Membrane Science*, 349 (2010) 349-357.
- [55] L.D. Nghiem, A.I. Schäfer, M. Elimelech, Role of electrostatic interactions in the retention of pharmaceutically active contaminants by a loose nanofiltration membrane, *Journal of Membrane Science*, 286 (2006) 52-59.
- [56] W.R. Bowen, J.S. Welfoot, Modelling the performance of membrane nanofiltration—critical assessment and model development, *Chemical Engineering Science*, 57 (2002) 1121-1137.
- [57] S. Bhattacharjee, DLS and zeta potential – What they are and what they are not?, *Journal of Controlled Release*, 235 (2016) 337-351.
- [58] S. You, J. Lu, C.Y. Tang, X. Wang, Rejection of heavy metals in acidic wastewater by a novel thin-film inorganic forward osmosis membrane, *Chemical Engineering Journal*, 320 (2017) 532-538.
- [59] H. Saitúa, F. Giannini, A.P. Padilla, Drinking water obtaining by nanofiltration from waters contaminated with glyphosate formulations: Process evaluation by means of toxicity tests and studies on operating parameters, *Journal of Hazardous Materials*, 227-228 (2012) 204-210.
- [60] S. Paul, W.F. Meggitt, P. Donald, Adsorption, mobility, and microbial degradation of glyphosate in the soil, *Weed Science*, 23 (1975) 229-234.
- [61] H.A. Pereira, P.R.T. Hernandez, M.S. Netto, G.D. Reske, V. Vieceli, L.F.S. Oliveira, G.L. Dotto, Adsorbents for glyphosate removal in contaminated waters: a review, *Environmental Chemistry Letters*, 19 (2021) 1525-1543.
- [62] R. Wang, S. Lin, Pore model for nanofiltration: History, theoretical framework, key predictions, limitations, and prospects, *Journal of Membrane Science*, 620 (2021) 118809.
- [63] D.L. Oatley, L. Llenas, R. Pérez, P.M. Williams, X. Martínez-Lladó, M. Rovira, Review of the dielectric properties of nanofiltration membranes and verification of the single oriented layer approximation, *Advances in Colloid and Interface Science*, 173 (2012) 1-11.
- [64] A.E. Yaroshchuk, Dielectric exclusion of ions from membranes, *Advances in Colloid and Interface Science*, 85 (2000) 193-230.
- [65] S. Bandini, D. Vezzani, Nanofiltration modeling: the role of dielectric exclusion in membrane characterization, *Chemical Engineering Science*, 58 (2003) 3303-3326.
- [66] L.A. Richards, A.I. Schäfer, B.S. Richards, B. Corry, The importance of dehydration in determining ion transport in narrow pores, *Small*, 8 (2012) 1701-1709.
- [67] Z.W. Windom, M. Datta, M.M. Huda, M.A. Sabuj, N. Rai, Understanding speciation and solvation of glyphosate from first principles simulations, *Journal of Molecular Liquids*, 365 (2022) 120154.
- [68] J.P.A. de Jesus, F.d.A. La Porta, Elucidating the spectroscopic and physicochemical properties for a pH-dependent glyphosate structure from a computational perspective, *Computational and Theoretical Chemistry*, 1226 (2023) 114209.
- [69] O. Fliss, K. Essalah, A. Ben Fredj, Stabilization of glyphosate zwitterions and conformational/tautomerism mechanism in aqueous solution: insights from ab initio and density functional theory-continuum model calculations, *Physical Chemistry Chemical Physics*, 23 (2021) 26306-26323.
- [70] M. Andreev, J.J. de Pablo, A. Chremos, J.F. Douglas, Influence of ion solvation on the properties of electrolyte solutions, *The Journal of Physical Chemistry B*, 122 (2018) 4029-4034.
- [71] V. Wieser, L.L.E. Mears, R.D. Barker, H.-W. Cheng, M. Valtiner, Hydration forces dominate surface charge dependent lipid bilayer interactions under physiological conditions, *The Journal of Physical Chemistry Letters*, 12 (2021) 9248-9252.
- [72] V. Pavluchkov, I. Shefer, O. Peer-Haim, J. Blotevogel, R. Epsztein, Indications of ion dehydration in diffusion-only and pressure-driven nanofiltration, *Journal of Membrane Science*, 648 (2022) 120358.
- [73] R. Epsztein, R.M. DuChanois, C.L. Ritt, A. Noy, M. Elimelech, Towards single-species selectivity of membranes with subnanometre pores, *Nature Nanotechnology*, 15 (2020) 426-436.

- [74] C. Song, B. Corry, Intrinsic ion selectivity of narrow hydrophobic pores, *The Journal of Physical Chemistry B*, 113 (2009) 7642-7649.
- [75] H. Liu, C.J. Jameson, S. Murad, Molecular dynamics simulation of ion selectivity process in nanopores, *Molecular Simulation*, 34 (2008) 169-175.
- [76] L.A. Richards, B.S. Richards, B. Corry, A.I. Schäfer, Experimental energy barriers to anions transporting through nanofiltration membranes, *Environmental Science & Technology*, 47 (2013) 1968-1976.
- [77] Y. Sun, P.B. Petersen, Solvation shell structure of small molecules and proteins by IR-MCR spectroscopy, *The Journal of Physical Chemistry Letters*, 8 (2017) 611-614.
- [78] D. Ben-Amotz, Hydration shell vibrational spectroscopy, *Journal of the American Chemical Society*, 141 (2019) 10569-10580.
- [79] P. Perera, M. Wyche, Y. Loethen, D. Ben-Amotz, Solute-induced perturbations of solvent-shell molecules observed using multivariate Raman curve resolution, *Journal of the American Chemical Society*, 130 (2008) 4576-4577.
- [80] C. Bottari, L. Almásy, B. Rossi, B. Bracco, M. Paolantoni, A. Mele, Interfacial water and microheterogeneity in aqueous solutions of ionic liquids, *The Journal of Physical Chemistry B*, 126 (2022) 4299-4308.
- [81] C. Ruckebusch, L. Blanchet, Multivariate curve resolution: A review of advanced and tailored applications and challenges, *Analytica Chimica Acta*, 765 (2013) 28-36.
- [82] M. Garrido, F.X. Rius, M.S. Larrechi, Multivariate curve resolution–alternating least squares (MCR-ALS) applied to spectroscopic data from monitoring chemical reactions processes, *Analytical and Bioanalytical Chemistry*, 390 (2008) 2059-2066.
- [83] J.G. Davis, K.P. Gierszal, P. Wang, D. Ben-Amotz, Water structural transformation at molecular hydrophobic interfaces, *Nature*, 491 (2012) 582-585.
- [84] D.S. Wilcox, B.M. Rankin, D. Ben-Amotz, Distinguishing aggregation from random mixing in aqueous t-butyl alcohol solutions, *Faraday Discussions*, 167 (2013) 177-190.
- [85] H.-P. Cheng, Water clusters: Fascinating hydrogen-bonding networks, solvation shell structures, and proton motion, *The Journal of Physical Chemistry A*, 102 (1998) 6201-6204.
- [86] M.W. Feyereisen, D. Feller, D.A. Dixon, Hydrogen bond energy of the water dimer, *The Journal of Physical Chemistry*, 100 (1996) 2993-2997.
- [87] A. Imbrogno, A.I. Schäfer, Comparative study of nanofiltration membrane characterization devices of different dimension and configuration (cross flow and dead end), *Journal of Membrane Science*, 585 (2019) 67-80.

REVIEWERS' COMMENTS

Reviewer #1 (Remarks to the Author):

The authors have made additional (and important) changes to the manuscript. As my core concerns regarding the novelty remained, I'm leaving the decision on publication to the editors and wishing the authors best of luck.

Thank you very much for your comments. Through our revisions and efforts to address your concerns, we have significantly improved our analyses and the quality of this paper. As the editors have agreed to our submission, we hope that our data are well-presented and can provide a basis for further engaging discussions and exchanges on this topic.

Reviewer #3 (Remarks to the Author):

I have once more read the submitted paper and I believe the quality has improved compared quite a bit compared to the two previous versions. Also, the introduction is more concise and focused. This paper contributes in obtaining a better understanding (although not conclusive) of the transport mechanisms in nanofiltration membranes and are worth publishing.

We appreciate very much your comment.

Removal of glyphosate and aminomethylphosphonic acid (AMPA) by nanofiltration membranes: the role of hydration

In this paper Trinh et al. study the retention of glyphosate of AMPA by a range of nanofiltration membranes with different sizes and different separate layer (chemistries) for the membranes.

The main message of the paper is that besides size exclusion, Donnan exclusion and dielectric exclusion are both important parameters for membranes with larger pore sizes.

Regarding the paper, I have the following comments.

L 76: it is strange to base the amount of tonnage of organophosphate chemicals in agriculture in 2025 on a prediction from 2020. Is there no current data available of how much organophosphates have been used more recently, e.g. in 2024 or even better the first half of 2025.

L 132: It mentions that the slipping plane is the distance from the membrane where the electrolyte becomes mobile. This is not correct, beyond the slipping plane the water molecules become mobile. Electrolyte ions can also be mobile behind the slipping plane due to surface conductivity.

Instead of characterizing the surface charge by zeta potential it would be better to determine the (total) charge of the membrane. Also the zetapotential is a function of the electrolyte concentration, the lower the concentration the higher the potential. It remains unclear at what concentrations the zeta potential has been measured and also with what method the surface charge has been calculated/derived from the zeta potential.

L135 etc.: I would rather prefer to see an explanation in terms of Donnan potential, instead of electrical resistance due to the Debye length. The main parameter is the relative concentration of the ions in the membrane relative to that in the bulk.

L147: Equalizing Ψ_d with the ξ potential is not correct. Normally the ξ potential is lower than the Stern potential, especially for polymeric surfaces as membranes or rough surfaces. It is important to note that even at distances larger than the Debye length, the potential is still not zero. At the Debye length the potential has dropped by $1/\kappa$.

I would recommend to make reference to standard books about double layer theory. There is no need to explain this in detail in this paper and in any the case the given explanation is too simplistic.

Fig. 2 and L 249-253: The adsorbed amount of GLY and AMPA is calculated from the mass balance (aa given in table 2), under dynamic conditions. However, I would rather prefer to see independent adsorption isotherms, whereby the adsorbed amount is measured for a range of equilibrium concentrations.

In figure 3 and also fig 4 it is puzzling to relate the datapoints to the actual membranes. The text mentions membrane (names) and the graphs shows MWCO. This makes it all very puzzling. For example in fig 3 at the 3 low MWCO, which datapoint corresponds with which membrane?

For the tighter membranes it was apparently not possible to obtain fluxes of 50 L/m².h., the flux was only 30 L/m².h. This shortcoming makes it difficult to compare the results for the different membranes, especially as the authors want to study the effect of the membrane flux on the extension of the hydration layer, and hence retention.

The pore diameter of the tighter membranes (BW 30, NF 90 and NF 270) varies from 0.46nm to 0.68nm, which is more than 45% increase. Nevertheless, the results in fig. 3 hardly show any

difference in performance. This is highly puzzling. Also the hydraulic diameter of GLY and AMPA (resp. 0.62nm and 0.49nm) is larger than the pore diameter of BW30. I therefore would expect the highest retention for this membrane, which should be close to 100%.

L277: why would the molecular orientation have an effect on the retention? I assume that if an omp is small enough it will pass the membrane anyway independent of orientation.

L297-303: see previous comment, I would rather prefer to explain this in terms of Donnan potential, which determines the ion concentrations in the pores. The way it is described now remains very qualitative. Also L302 states that 'the strength of electrostatic interactions will depend on the potential in the pore space'. Which interactions are meant here. Is it the interaction between the omp and the membranes, or is this all about the ion concentration in the pore which is determined by the Donnan potential?

Fig 4: B is hard to read because of the size of the figure. Also I am puzzled by what the insert should demonstrate.

L317: the presented zeta potentials are meaningless, as long as it is not clear how they are measured and at what salt concentrations.

L337: states that 'No adsorption is measured at pH 2', however, fig. 5 I, does show low adsorption at pH 2.

Fig 5 shows a strong effect of membrane surface chemistry (polyamide vs polysulfone) on adsorption and removal. In l338-344 an attempt is made to explain this observation, but this remains rather opportunistic and superficial. I miss a more in depth analysis, e.g. what is the difference in surface groups, what is the density of these groups and how can they interact with the AMPA/GLY, how strong are these interactions, etc.

L365: Why is the higher removal of GLY (compared to AMPA) due to the higher negative charge of GLY. Would the removal not be higher because of the higher Mw of GLY?

L399-401: Here it is stated that more molecules could pass through the membrane pores because of shredding of the hydration layer. I wonder if this is true and without quantifying concentration polarization this statement is hard to substantiate.

L423: states that 'the different removal between GLY and AMPA cannot be explained by hydrogen bonding dielectric exclusion'. However, L 425-426 states that both Donnan exclusion and dielectric exclusion play a role. So what part can be explained by Donnan exclusion and what part by dielectric exclusion?

L439-443: My understanding is that at lower pH the hydration shell is more structured and hence smaller. If this is indeed the analysis, then this should be described clearer.

L457: apart from the hydration shell also the Mw of GLY is higher than that of AMPA, which should have an effect on the retention.

L506: In the conclusion paragraph I find it confusing to read that the flux was expected to play an important role in the transport ... The conclusion paragraph should be conclusive what the role of the flux is, in other words plays the flux an important role or not.

Some smaller comments

L 107: remove `and`

L141: remove `the`

L148: remove `then`

L 183: replace difference with `different`.

L260: remove `to the` and add `and`

L 292: add `and`

L344: Should read `sulphonated`

L 376: replace `differing` with `different`